# Yes, Q-learning Helps Offline In-Context RL

## Abstract

Existing offline in-context reinforcement learning (ICRL) methods have predominantly relied on supervised training objectives, which are known to have limitations in offline RL settings. In this study, we explore the integration of RL objectives within an offline ICRL framework. Through experiments on more than 150 Grid-World and MuJoCo environment-derived datasets, we demonstrate that optimizing RL objectives directly improves performance by approximately 30% on average compared to widely adopted Algorithm Distillation (AD), across various dataset coverages, structures, expertise levels, and environmental complexities. Furthermore, in the challenging XLand-MiniGrid environment, RL objectives doubled the performance of AD. Our results also reveal that the addition of conservatism during value learning brings additional improvements in almost all settings tested. Our findings emphasize the importance of aligning ICRL learning objectives with the RL reward-maximization goal, and demonstrate that offline RL is a promising direction for advancing ICRL.

## 1 Introduction

The advent of sequence generation models, particularly those based on the Transformer architecture (Vaswani, 2017), has revolutionized many fields by enabling models to generalize beyond their training domain. In particular, large language models can perform novel tasks by processing a textual description and a few examples provided as input without any parameter updates, a phenomenon known as in-context (IC) learning (Brown et al., 2020). This capability is highly desirable in meta-reinforcement learning (Beck et al., 2023), where the goal is to produce an agent that is able to adapt efficiently to unseen tasks. Recently appeared in-context RL (Moeini et al., 2025) aims to produce general meta-RL models with scalable architectures analogous to Large Language Models. However, training such models online is not feasible and may be unsafe. Offline pre-training increases the applicability of ICRL by eliminating potentially costly or dangerous online interactions, as seen in domains such as robotics, autonomous driving, and healthcare. However, current offline ICRL methods face critical limitations.

Established approaches such as Algorithm Distillation (AD) (Laskin et al., 2022) and Decision-Pretrained Transformer (DPT) (Lee et al., 2024), along with their variants, have shown promise in offline ICRL. However, none of these methods explicitly optimize for the RL objective - maximizitaion of cumulative reward. This oversight poses significant challenges when dealing with offline RL tasks, where leveraging RL to achieve optimal behavior is crucial (Kumar et al., 2022). Although recent work (Grigsby et al., 2023; Elawady et al., 2024) has explored online ICRL, these methods rely on numerous heuristics for effective performance and remain untested in offline setting, which is inherently more challenging (Levine et al., 2020). See Appendix A for an extended related work.

Our main goal is to investigate whether methods that optimize RL objective can achieve significantly better results in offline ICRL and whether this improvements are universal across various axis. In particular, we aim to address the following questions: 1) Does explicit optimization of the RL objective improve performance in offline ICRL? 2) How does the effectiveness of this optimization depend on the coverage and quality of offline datasets? 3) Do we need specialized RL techniques from the offline RL family for effective offline ICRL? 4) How would algorithms behave if we do not have access to learning histories but rather a random set of trajectories? 5) Does RL better handle mixture of dynamics and out-of-distribution dynamics?

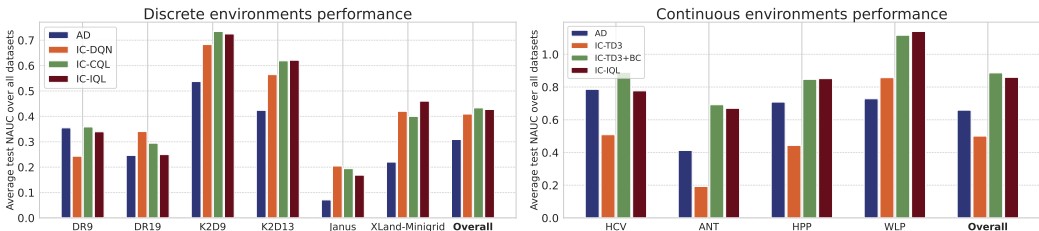

Figure 1: Mean test NAUC scores across environments averaged over all constructed datasets. NAUC is a normalized AUC of the test-time performance curve, see Section 3.3 for details.

To answer these questions, we performed an empirical study using more than 150 datasets derived from the widely used GridWorld and MuJoCo (Todorov et al., 2012) tasks. We compared several RL-based approaches with Algorithm Distillation, a strong and widely adopted supervised baseline, to evaluate the impact of explicitly optimizing for reward in offline ICRL. To our knowledge, this is the first study to explicitly optimize the RL objective in an offline ICRL setting using a scalable Transformer architecture.

In summary, we adapt a Transformer-based AD backbone to incorporate explicit RL objectives and evaluate them across diverse offline datasets. Our key findings are that (i) optimizing RL objectives substantially improves in-context performance compared to supervised AD approach, (ii) offline-regularized algorithms consistently outperform unconstrained counterparts, and (iii) these gains persist across dataset coverage and quality levels. These results suggest practitioners should prefer offline-constrained RL objectives when training large-scale ICRL agents.

## 2 PRELIMINARIES

### 2.1 OFFLINE IN-CONTEXT REINFORCEMENT LEARNING

Reinforcement Learning (RL) is commonly formulated as a Partially Observable Markov Decision Process (POMDP) defined by the tuple $(\mathcal{S}, \mathcal{A}, \mathcal{O}, P, R, \Omega, \gamma)$. At each timestep, an agent receives an observation $o \in \mathcal{O}$, selects an action $a \in \mathcal{A}$ according to its policy $\pi(a \mid o)$, and receives a reward $r = R(s, a)$. The goal is to learn an optimal policy $\pi^*$ that maximizes the expected cumulative discounted reward: $J(\pi) = \mathbb{E}\left[\sum_{t=0}^{\infty} \gamma^t r_t \mid \pi\right]$, where $r_t = R(s_t, a_t)$. Offline RL (Levine et al., 2020) aims to learn such a policy solely from a fixed dataset $\mathcal{D} = \{(o_i, a_i, r_i, o'_i)\}_{i=1}^{N}$ without further interaction with the environment. In-Context RL (ICRL) (Moeini et al., 2025) aims to enable adaptation to new tasks purely through contextual learning, without explicit parameter updates. Offline ICRL specifically trains models on pre-collected data to infer and adapt to novel tasks at deployment.

### 2.2 ALGORITHM DISTILLATION

Algorithm Distillation (AD) (Laskin et al., 2022) is an offline ICRL method that serves as a strong baseline for subsequent works (Lee et al., 2024; Elawady et al., 2024). AD distills a policy improvement operator from a dataset

$$\mathcal{D} = \left\{\left(\tau_1^{\mathcal{G}_i}, \ldots, \tau_n^{\mathcal{G}_i}\right) \sim \mathcal{A}_{\mathcal{G}_i} \mid \mathcal{G}_i \in p(\mathcal{G})\right\}_{i=1}^{N},$$

where $\mathcal{A}_{\mathcal{G}_i}$ is an RL algorithm trained in environment $\mathcal{G}_i$, and each trajectory $\tau_j^{\mathcal{G}_i} = \left(o_1^j, a_1^j, r_1^j, \ldots, o_T^j, a_T^j, r_T^j\right)$ represents interactions collected during the base algorithm training. The ordered sequence of these trajectories is referred to as a learning history. AD trains an autoregressive Transformer $M_\theta$ to predict the next action given a segment of learning history and the current observation $o_k^{j+C}$:

$$\hat{a}_k^{j+C} = M_\theta\left(o_{T-l}^j, a_{T-l}^j, r_{T-l}^j, \ldots, o_T^j, a_T^j, r_T^j, o_1^{j+1}, \ldots, o_{k-1}^{j+C}, a_{k-1}^{j+C}, r_{k-1}^{j+C}, o_k^{j+C}\right).$$

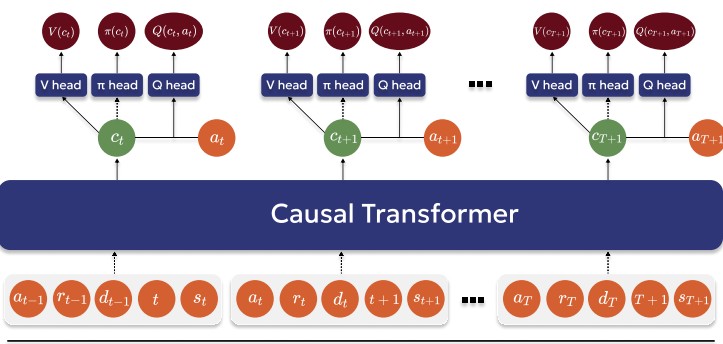

Figure 2: Overview of the proposed approach. As the input, our model takes a sequence of trajectories (without hard requirements on their structure) where each transition is represented with a tuple consisting of previous action, previous reward, previous episode's done flag, current episode timestep and other sequence elements marked by different timestep subscripts ($t$ and $T$) to indicate their potential origin from distinct trajectories. Then the resulting context embedding $c_t$ is used to predict both value functions and the policy output $\pi$. The V-head is employed only in IC-IQL, while the $\pi$ head is used exclusively for continuous actions. Dashed arrows denote the absence of gradient flow.

It is crucial that segments of learning histories sampled for training span multiple episodes for the in-context learning property to emerge. After pretraining, $M_\theta$ can solve unseen tasks in-context without requiring parameter updates.

## 3 METHODOLOGY

### 3.1 RL INCORPORATION

In this study, we use Algorithm Distillation (AD) (Laskin et al., 2022), a transformer-based (Vaswani, 2017) architecture, as our baseline. AD's objective is to predict the next action given the improving learning history as context, where each step is represented as a tuple (*state*, *previous action*, *previous reward*). In our implementation these tuples are encoded into a single token through concatenation.

We retain the same Transformer backbone as AD but introduce several modifications to incorporate RL objectives: inspired by Grigsby et al. (2023), input tuples are augmented with *previous done* flags to indicate episode termination and current episode *step* (we provide ablation on this design choice in Appendix D); the next-action prediction head is replaced with value-function heads trained using corresponding RL loss functions. For continuous problems, we also add the policy head. The illustration can be found in Figure 2.

We adopt three RL methods for discrete environments. Twin Deep Q Network (DQN) (Mnih, 2013): A simple RL method without offline-specific components. Conservative Q-Learning (CQL) (Kumar et al., 2020): A widely used offline RL approach that incorporates value-function pessimism. And Implicit Q-Learning (IQL) (Kostrikov et al., 2021): A popular offline RL method based on implicit regularization, known for its strong performance across diverse tasks. Inspired by the adaptation of IQL for the NLP tasks with ILQL (Snell et al., 2022) we add the CQL term to the IQL loss in discrete environments. We also run tests with continuous environments where we adopt TD3 (Fujimoto et al., 2018) as online baseline, it's minimalist TD3+BC (Fujimoto & Gu, 2021) offline modification and continuous IQL.

Each RL objective (DQN, CQL, IQL, TD3(+BC)) is applied on top of the shared Transformer backbone (exact loss formulas are moved to the Appendix B due to the volume limits). Q- and V-values are used in the usual temporal-difference (TD) fashion for the TD loss (i.e., next-value targets are used as in classical TD learning) and the transformer predicts these next values conditioned on the full offline context window. Target value networks are implemented as separate lightweight heads attached to the backbone to save memory and speed up training; this follows the AMAGO-inspired design. Gradients from Q- and V-head losses are backpropagated into the backbone (so the backbone

is trained from value losses), whereas when a policy head is present its gradients are detached and do not update the backbone for improved stability.

During inference, discrete approaches predict actions using the $\arg\max$ operator while continuous use deterministic policy output.

We refer to all of the RL modifications upon AD with IC- (In-Context) prefix, i.e., IC-DQN, IC-CQL, IC-IQL, IC-TD3 and IC-TD3+BC.

## 3.2 ENVIRONMENTS AND DATASETS

For most of our experiments we utilize two environments used by Laskin et al. (2022): Dark Room (DR) and Dark Key-to-Door (K2D). DR is a discrete Markov Decision Process (MDP), while K2D is a partially observable MDP (POMDP). Both environments involve a 2D grid where the agent can move up, down, left, right, or remain stationary, observing only its current position at each step. These are popular environments that allow us to scale our experiments under the limited computational budget for obtaining trustworthy conclusions. In Appendix C we introduce modified DR environment for a separate set of experiments. We also run tests using popular continuous MuJoCo environments widely used in meta RL research (Rakelly et al., 2019): HalfCheetahVel (HCV), AntDir (ANT), HopperParams (HPP) and Walker2DParams (WLP).

In DR, the agent starts at the center of the grid and must navigate to an unknown target goal to receive a reward of 1. In K2D, the agent starts at a random location and must first find a "key" (reward: 1) and then reach a "door" (reward: 1), both of which have unknown locations. In both environments episodes terminate either upon task completion or after a fixed number of steps. In HCV agent must run with a fixed unknown velocity, in ANT agent has to navigate to the unknown point, and in HPP and WLP agent must move as fast as possible avoiding falling but in different instances of environments system parameters (e.g. gravity or masses) are randomized.

For DR we consider 9x9 version with episode length of 20 and 19x19 version with episode length of 100. For K2D we test 9x9 version with 50 steps per episode and 13x13 version with 100 steps per episode. This allows us to test performance across different levels of complexity. MuJoCo environments are truncated after 200 steps.

For discrete environments, we collect training histories using the Q-learning (Watkins & Dayan, 1992) algorithm, varying the number of histories per target goal (1 or 5). For DR 9x9 we form datasets for 70, 40 and 20 train targets, for DR 19x19 we collect histories with 300, 150 and 75 train targets, and for both K2D versions we created datasets with 1000, 500 and 250 train goals. For continuous environments we collect learning histories from 100, 50 and 25 environments instances using Soft Actor-Critic (SAC) algorithm (Haarnoja et al., 2018). To analyze the impact of data quality, we partition the trajectory datasets into three expertise levels `early`, `mid` and `late` by dividing original datasets into three equal parts trajectory-wise.

We name datasets using the following convention: {*environment name*}{{*grid size*}}-{*num training targets*}-{*num histories per target*}[-{*expertise level*}]. Absence of the expertise level in the name indicates the full (complete) dataset. For example, "DR9-70-5" refers to a complete dataset from the 9x9 DR environment with 70 training targets and 5 histories per target. Additional dataset details are provided in Appendix G.

## 3.3 EVALUATION

To assess the performance of trained policies, we roll out each policy over 100 successive episodes for all discrete environments and track performance after 25, 50, and 100 (1, 2, 4) episodes. For continuous environments we roll out over 4 episodes and track performance after 1, 2 and 4 episodes.[1] Additionally, we compute the Normalized Area Under the Curve (NAUC)[2] of episode performance. We also report metrics from the rliable library (Agarwal et al., 2021) above the tracked ones for the reliability.

---

[1]Note that the optimal meta policy should be able to solve all the considered discrete tasks within four episodes and continuous tasks within one or two episodes.

[2]The AUC value is divided by the expert policy AUC.

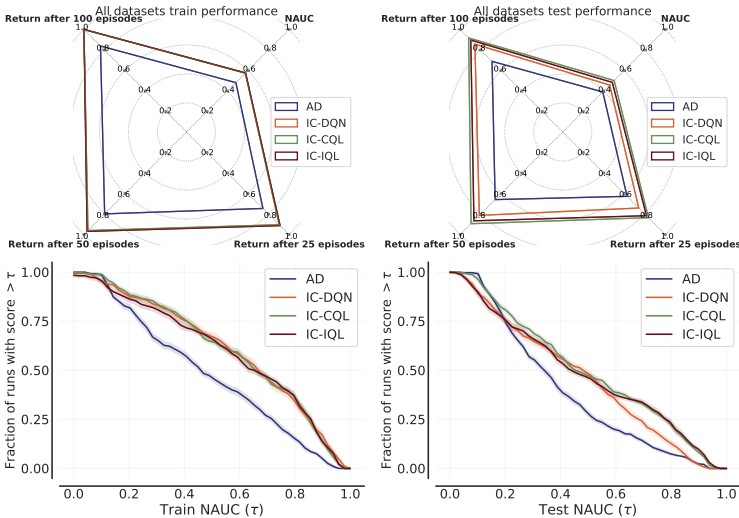

Figure 3: Top: algorithms performance across tracked metrics averaged over all discrete datasets. Bottom: rliable performance profiles of NAUC. Left: train targets. Right: test targets.

The NAUC provides a single numerical value for comparing agents, as it captures the progression of performance over episodes while being more robust to noise than fixed-episode evaluations. Tracking performance at fixed episodes helps identify convergence rates and potential degradation during rollouts. NAUC is used for selecting the best hyperparameters (details in Appendix F).

For the DR environments, we evaluate on all target goals that are excluded from the training set. For all other environments, we use a fixed set of 100 random test targets or configurations that are not part of the training data. To ensure robustness, we follow the evaluation protocol from Tarasov et al. (2024b), using different random seeds for hyperparameter search and final evaluation.

## 4 EXPERIMENTAL RESULTS

We begin this section by comparing the overall performance of the selected methods using discrete environments, demonstrating the suitability of the newly introduced NAUC metric for evaluation. Subsequently, we analyze the performance of these methods across critical offline RL dimensions, including data quality and coverage. Further, we investigate the impact of removing the assumption of access to learning histories, which may not always hold in real-world scenarios (Zisman et al., 2023). In addition, we conduct the experiment in a challenging XLand-MiniGrid (Nikulin et al., 2023) environment. Finally, we show that the benefits from using RL extend to continuous environments. In Appendix C, we also test the ability to learn in a mixture of dynamics along with the handling of out-of-distribution (OOD) dynamics.

### 4.1 OVERALL PERFORMANCE

For this analysis, we utilized all available discrete datasets. The top graphs in Figure 3 show the averaged metrics for both test and train targets. Across the Dark Room (DR) and Dark Key-to-Door (K2D) environments, we observe that the tested methods maintain stable performance throughout rollouts, confirming that NAUC is a reliable metric for comparative analysis.

The bottom plots in Figure 3 display performance profiles based on NAUC. From these results, several key observations can be made. First, RL-based approaches consistently outperform Algorithm Distillation (AD) on average. Second, while all RL methods show similar performance on train targets, there are notable differences on test targets. CQL achieves the best performance on test targets (with a 28.8% average improvement compared to AD), IQL follows closely behind (23.7% improvement) and DQN exhibits the weakest performance among the RL methods on average (16.6% improvement). Tabular results for each dataset are provided in Appendix J, and additional rliable

metrics are included in Appendix I.1. These metrics, including the Interquartile Mean (IQM) and mean values for both NAUC and final scores, statistically validate the superior performance of CQL over other methods.

These findings support our core hypothesis that optimizing the RL objective is crucial for solving ICRL problems. Moreover, the performance gap between the offline RL objectives and the online DQN highlights the advantages of offline RL algorithms in this setup. It is important to note that we have very limited hyperparameters tuning of RL approaches compared to AD tuning, and potential gains might be even higher with equal tuning budgets (see Appendix F).

## 4.2 VARIOUS COVERAGE

In this part of the analysis, we explore the impact of dataset coverage, a critical property in the offline RL setup (Schweighofer et al., 2021). Coverage is examined along two axes: the number of unique training targets and the number of learning histories per target. As our experiments below demonstrate, AD requires multiple histories per task collected from different agents to perform well. However, such multihistory coverage may be impractical in real-world settings. To investigate these aspects, we use the complete datasets across all environments.

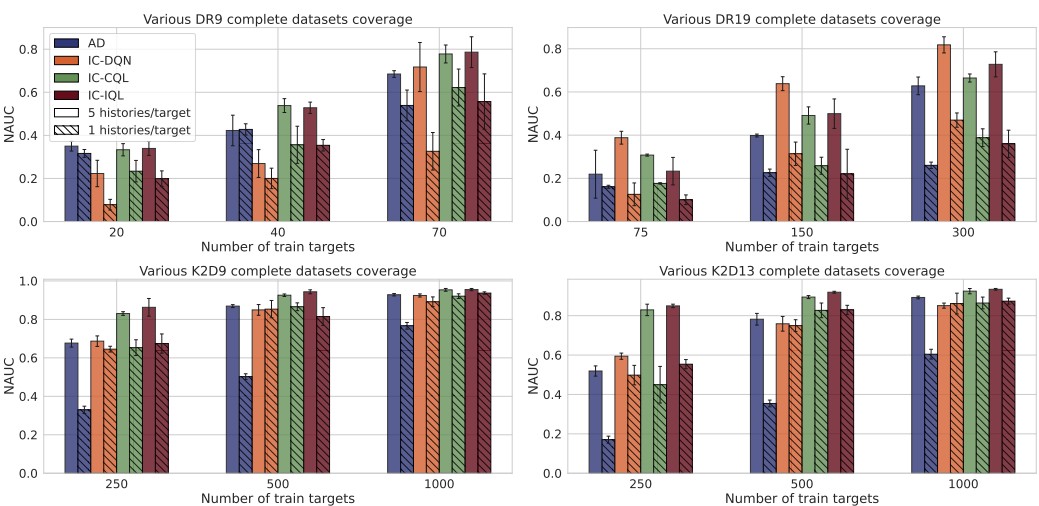

Figure 4: NAUC score comparison between considered approaches for various dataset coverage in terms of number of train targets and histories per target. Averaged over 4 test random seeds. Confidence intervals depict std across seeds.

The NAUC scores for each dataset are shown in Figure 4. As expected, all methods perform better with increased target coverage and more histories per target. However, the results highlight notable differences in performance across methods. Offline RL approaches outperform AD across most configurations. The exception is the DR9 datasets, where AD slightly exceeds offline RL in scenarios with 20 and 40 targets when using only one history per target. DQN also outperforms AD on average and, surprisingly, shows superior performance over offline RL approaches on DR19 tasks. In the more complex K2D environments, RL-based methods are significantly more robust to the absence of multiple histories per target. For K2D, RL approaches achieve close performance with the five histories per target setup once a sufficient target coverage level is reached. In contrast, AD experiences a notable drop in performance without repeated targets, underscoring its reliance on repetitive learning histories.

The key takeaway is that RL-based approaches are more data-efficient than AD and demonstrate greater tolerance for limited target repetition in learning histories. This robustness makes RL methods more suitable for real-world applications, where complete coverage and extensive learning histories are often unattainable.

## 4.3 VARIOUS EXPERTISE

Dataset expertise is another critical factor in offline RL (Schweighofer et al., 2021). In this part of the analysis, we evaluate the performance of different methods across discrete datasets of varying expertise levels. The "complete" datasets represent full learning histories, which were originally proposed for AD. The `mid` datasets include interpolation between low-quality and near-convergence trajectories, resembling truncated versions of the complete datasets. In contrast, `early` and `late` datasets reflect real-world scenarios: the former consists of low-quality data that is relatively easy to gather, while the latter comprises near-optimal examples of problem solutions. Detailed statistics for these datasets can be found in Appendix G.

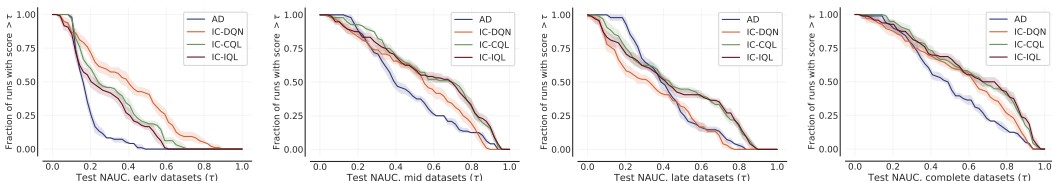

Figure 5: rliable performance profiles of NAUC for various discrete datasets expertise. From left to right: `early`, `mid`, `late`, complete datasets.

Figure 5 shows the test NAUC scores for the various dataset types. AD performs notably poorly on `early` datasets, failing to produce policies with NAUC scores higher than 0.4. In contrast, all RL approaches achieve significantly higher scores. DQN emerges as the best-performing method in this setup, likely because low-quality datasets require less regularization, allowing the agent to pursue higher rewards. CQL, which is implemented with cross-entropy loss in discrete cases, can be viewed as an interpolation between DQN and AD. Its performance is closer to AD when the regularization coefficient is high, which limits its potential on low-quality datasets. Reducing CQL's pessimism might yield better results than DQN, but we did not test it due to computational constraints.

As anticipated, the `mid` and complete datasets exhibit similar performance trends, with AD remaining the weakest approach and CQL leading, closely followed by IQL. Surprisingly, AD performs competitively on `late` datasets with high coverage, despite the lower data diversity in these datasets. In contrast, DQN's performance on `late` datasets is significantly weaker than the offline RL methods, further underscoring the importance of offline regularization. Additional rliable metrics and final performance scores in Appendix I.2 statistically validate these observations.

In summary, the experiments highlight that RL-based methods outperform AD across datasets of varying expertise levels. However, the results also suggest that the hyperparameters of offline RL methods need to be carefully tuned to match the quality of the available data, an aspect not fully explored in this iteration of the study.

## 4.4 ABSENCE OF THE LEARNING HISTORY STRUCTURE

Algorithm Distillation relies on the availability of progressing learning histories, with multiple behavior policies collecting data for each task. In practice, however, such structured data is rarely available. To address this, we conducted experiments to evaluate how all considered approaches perform when the inherent ordering of learning histories is absent. First, we tested the algorithms using a randomly shuffled dataset, which disrupts the sequential improvement that AD is designed to distill. To counteract this, we also investigated an approach to build some order: after randomly sampling trajectories, we sort them based on their discounted return values. Although this sorting method may be sensitive to the choice of discount factor, we found it to work effectively for some tasks.

For these experiments, we used complete datasets from the DR19 and K2D13 environments with one learning history per target, as this represents a more realistic scenario. As illustrated in Figure 6, on average, RL approaches outperform AD when the data is randomly ordered, with the sole exception of CQL on the DR19-300-1 dataset. Under random ordering, there is little difference among the RL methods on K2D13, while on DR19 DQN exhibits notable superiority – a result that is consistent with observations on highly sub-optimal (`early`) datasets. When the unordered

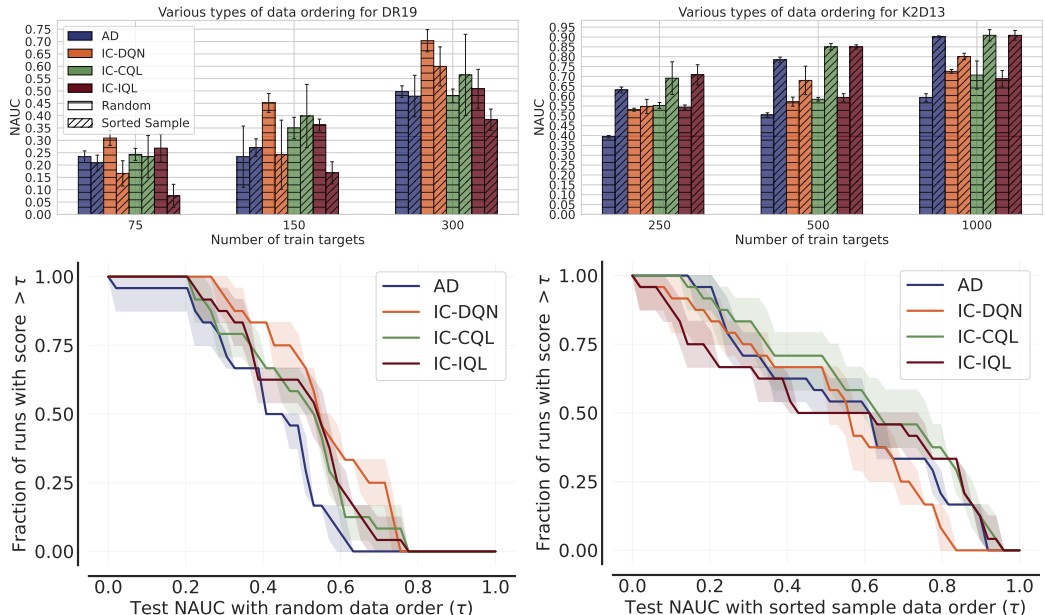

Figure 6: NAUC test scores on DR19 and K2D13 datasets with different data structures. Top: performance splitted across environments and dataset coverages. Bottom: rliable performance profiles for random data order (left) and sorted samples (right).

data is sorted by discounted return, offline RL methods consistently outperform both AD and DQN in K2D environment. In the DR19 environment, however, only CQL (which can be seen as an interpolation between AD and DQN) maintains its performance lead, while IQL shows diminished results. Surprisingly, on DR19, both DQN and IQL perform better with randomly ordered data than with sorted samples. We do not yet have a plausible explanation for this phenomenon or why it appears exclusively in the simpler DR19 environment.

In summary, CQL demonstrates the best performance across different environments and dataset coverages without learning histories access. Additional metrics presented in Appendix I.3 further support this finding.

## 4.5 EVALUATION ON THE XLAND-MINIGRID ENVIRONMENT

Table 1: XLand-Minigrid trivial `tiny` dataset test tasks scores. Statistics are averaged over 4 random seeds.

| Metric | AD | IC-DQN | IC-CQL | IC-IQL |
|---|---|---|---|---|
| NAUC | $0.22 \pm 0.03$ | $0.42 \pm 0.03$ | $0.40 \pm 0.04$ | $0.46 \pm 0.03$ |
| Last episode mean return | $0.21 \pm 0.02$ | $0.43 \pm 0.05$ | $0.39 \pm 0.03$ | $0.45 \pm 0.05$ |

To further validate our approach in more challenging settings, we evaluate our methods on the XLand-Minigrid trivial environment (Nikulin et al., 2023) using datasets provided by Nikulin et al. (2024). In order to keep the experiments tractable, we reduce the dataset size by selecting only one learning history per rule set and retaining only the first third of transitions from each history. This subsampling results in a dataset that is just 1% of the original size, which we refer to as the `tiny` dataset.

Table 1 presents the performance results on this dataset, reporting both the NAUC and the mean return from the last episode. The results show that RL-based approaches significantly outperform AD: NAUC and mean return scores for RL methods are approximately twice as high as those for AD. Among the RL approaches, DQN slightly outperforms CQL, while IQL achieves marginally better performance than DQN.

It is noteworthy that in the original work (Nikulin et al., 2024), AD achieved a mean performance of approximately 0.4 **using 100 times more data and three times more rollout episodes** (500 episodes compared to 150 in our experiments). These findings clearly demonstrate that the benefits of explicitly optimizing RL objectives extend to more complex and data-sparse environments.

## 4.6 CONTINUOUS STATE AND ACTION SPACES

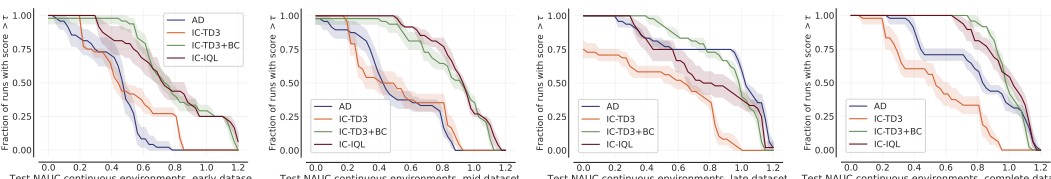

Figure 7: rliable performance profiles of NAUC for various continuous datasets expertise. From left to right: `early`, `mid`, `late` complete datasets.

Thus far, our experiments have focused on discrete environments, which offer a controlled setting to analyze the behavior of RL-based ICRL methods. However, many real-world applications – ranging from robotics and autonomous driving to control tasks – operate in continuous state and action spaces. In this subsection, we extend our experimental analysis to continuous environments to determine how explicit RL objective optimization performs when faced with the additional challenges posed by infinite state and action spaces. This exploration aims to bridge the gap between our current discrete experiments and the demands of real-world applications, ultimately paving the way for broader adoption and further refinement of offline ICRL methods.

In Figure 7, we present performance profiles for AD, the online RL method TD3 (Fujimoto et al., 2018), its lightweight offline RL counterpart TD3+BC (Fujimoto & Gu, 2021) and IQL. Consistent with our findings in Section 4.3, the offline RL approaches (TD3+BC and IQL) significantly outperform AD across most setups, with AD matching performance only on `late` (near-expert behavior) datasets. IQL performs slightly worse than TD3+BC on average. Notably, a key difference emerges between continuous and discrete environments: the online RL method (TD3) demonstrates lower performance than AD in continuous domains, likely due to the more severe challenges posed by out-of-distribution states and actions. These results underscore not only that ICRL methods that incorporate offline RL components achieve better performance, but also highlight the critical importance of the offline component in continuous settings. See Appendix I and Appendix J for more results and metrics.

## 5 CONCLUSION AND LIMITATIONS

In this work, we have demonstrated that explicitly optimizing RL objectives is highly beneficial for offline ICRL. Our experiments reveal that incorporating RL optimization leads to improved performance across a variety of environments, dataset coverage levels, and dataset expertise and structure conditions. In particular, even under much smaller hyperparameters tunning budget offline RL approaches consistently outperform Algorithm Distillation and are usually more effective than online methods, highlighting the advantages of offline-specific regularizations and methodologies in many ICRL scenarios.

Although our approach improves several aspects of ICRL, it still relies on large model capacities and diverse training data to enable in-context capabilities to emerge even in relatively simple environments. Although we observe better performance in out-of-distribution (OOD) test environments, our method does not fully resolve this challenge (see Appendix C). Furthermore, we do not introduce any mechanisms specifically designed to handle adaptation to novel environments or OOD observations, a limitation shared by the offline RL methods employed in this work. Potential directions for addressing these challenges are discussed in Appendix E.

Overall, our results underscore the importance of aligning learning objectives with the intrinsic goals of Reinforcement Learning, setting the stage for more robust and efficient offline ICRL methods.

## 6 REPRODUCIBILITY STATEMENT

The source code which includes environments, datasests collection and all algorithms implementation is provided as an attachment to our submission. The detailed description of our methodology, environments, evaluation and experimental details is provided in section 3, Appendix F and Appendix G. The exact hyperparameters for each approach are provided in Appendix H.

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

## A    RELATED WORK

### A.1    OFFLINE REINFORCEMENT LEARNING

Offline RL aims to train agents that maximize reward using pre-collected datasets without interacting with the environment. This setup introduces unique challenges, particularly in handling out-of-distribution (OOD) state-action pairs (Levine et al., 2020). Over the years, this field has witnessed rapid development, with various methods proposed to address these challenges (Kumar et al., 2020; An et al., 2021; Kostrikov et al., 2021; Fujimoto & Gu, 2021). In our study we test widely adopted offline RL baselines for offline ICRL setting in order to demonstrate benefits they bring as reward maximization algorithms. For discrete environments we used Conservative Q-learning (CQL) (Kumar et al., 2020) and Implicit Q-learning (IQL) (Kostrikov et al., 2021). Based on findings from Tarasov et al. (2024c), for continuous environments we used IQL and simple yet effective (Tarasov et al., 2024b) TD3+BC (Fujimoto & Gu, 2021) approach.

A prominent direction in offline RL involves modeling trajectories with Transformers through supervised learning, as first introduced by Decision Transformer (DT) (Chen et al., 2021). However, subsequent studies (Yamagata et al., 2023; Hu et al., 2024; Zhuang et al., 2024) demonstrated that supervised approaches, which lack explicit reward maximization, often fail with low-quality datasets or datasets which do not contain problem solving trajectories. They struggle to "stitch" suboptimal trajectories into optimal policies – a limitation that can be addressed by methods that directly optimize RL objectives. Our intuition tells that in the context of offline ICRL similar issues might arise when reward is not maximized which is confirmed by our experiments.

### A.2    IN-CONTEXT REINFORCEMENT LEARNING

Algorithm Distillation (AD) (Laskin et al., 2022) marked a significant step towards scalable In-Context RL (ICRL) by leveraging Transformer architectures to learn an "improvement" operator. It does so by distilling information from the training histories of single-task agents across various environments. AD assumes access to complete training histories, which may not always be available. In this work we demonstrate that RL-based approaches can levarage datasets more efficiently (especially datasets with low-quality demonstrations) and are able to handle unstructured data better.

Decision-Pretrained Transformer (DPT) (Lee et al., 2024) introduced another approach, focusing on predicting optimal actions from historical data and a given state. However, this method assumes access to an oracle for optimal action sampling, which is often impractical. RL-based methods that we test in this work do not require access to the oracle.

Neither AD, DPT, nor their follow-up modifications (Sinii et al., 2023; Schmied et al., 2024; Dai et al., 2024; Huang et al., 2024; Son et al., 2024; Zisman et al., 2024) optimize RL objectives during offline training. This omission can result in suboptimal policies, as these methods essentially adapt supervised learning techniques like DT to the offline ICRL setting, without addressing the fundamental reward maximization goal of RL.

Recent works such as AMAGO (Grigsby et al., 2023) and ReLIC (Elawady et al., 2024) have explored scalable In-Context RL by incorporating off-policy RL techniques. These methods outperform AD and DT in online RL setups but have yet to be tested in offline environments. Offline RL presents distinct challenges—such as the inability to interact with the environment—that make direct application of online approaches less effective (Fujimoto et al., 2019; Levine et al., 2020). This gap underscores the need for offline-specific methods that explicitly optimize RL objectives. Moreover, AMAGO and ReLIC rely on many implementation details and in this work we demonstrate that solid performance can be achieved without complex modifications. ReLIC is a purely online and based on PPO which can not be applied in offline setup correctly. And AMAGO is extremely over-engineered approach with a big amount of hyperparameters which require extensive tuning. We do not have enough resources to run fair comparison in this study.

## B    RL OBJECTIVES

Here we list the precise loss/objective functions used for the RL algorithms we integrate in our transformer-based in-context setting.

**DQN (Mnih, 2013)**    We optimize the Q-value function via the standard TD-error (Bellman) loss:

$$L_{\text{DQN}}(\theta) = \mathbb{E}_{(s,a,r,s')\sim\mathcal{D}} \left[ \left( Q_\theta(s,a) - (r + \gamma \max_{a'} Q_{\theta'}(s',a')) \right)^2 \right],$$

where $\theta'$, here and further, denotes the parameters of the target network.

**Conservative Q-Learning (CQL) (Kumar et al., 2020)**    The loss is composed of a Bellman (TD) part plus a pessimism-promoting regularizer:

$$L_{\text{CQL}}(\theta) = \mathbb{E}_{(s,a,r,s')\sim\mathcal{D}} \left[ \left( Q_\theta(s,a) - (r + \gamma \max_{a'} Q_{\theta'}(s',a')) \right)^2 \right]$$

$$+ \ \alpha \left( \mathbb{E}_{s\sim\mathcal{D}} \left[ \log\left( \frac{1}{N} \sum_{i=1}^{N} \exp(Q_\theta(s,\tilde{a}_i)) \right) \right] - \mathbb{E}_{(s,a)\sim\mathcal{D}} \left[ Q_\theta(s,a) \right] \right),$$

where $\{\tilde{a}_i\}$ are actions sampled from some action distribution (e.g. uniform or behavior-policy), and $\alpha > 0$ is the pessimism (regularization) hyperparameter.

**Implicit Q-Learning (IQL) (Kostrikov et al., 2021)**    IQL uses two critic heads (value $V$ and Q) plus an expectile-based value loss, and a policy extraction (or policy loss if policy head is used):

$$L_V(\phi) = \mathbb{E}_{(s,a)\sim\mathcal{D}} \left[ \rho_\tau \big( Q_\theta(s,a) - V_\phi(s) \big) \right],$$
$$L_Q(\theta) = \mathbb{E}_{(s,a,r,s')\sim\mathcal{D}} \left[ \left( Q_\theta(s,a) - (r + \gamma V_\phi(s')) \right)^2 \right],$$

where $\rho_\tau(u)$ is the expectile (asymmetric squared) loss defined by

$$\rho_\tau(u) = |\tau - \mathbf{1}_{\{u<0\}}| \, u^2,$$

and $\tau \in (0,1)$ is the expectile hyperparameter.

If a policy head $\pi_\omega$ is used (continuous problems), the policy loss is:

$$L_\pi(\omega) = \mathbb{E}_{(s,a)\sim\mathcal{D}} \left[ \exp\big( \beta \left( Q_\theta(s,a) - V_\phi(s) \right) \big) \left( -\log \pi_\omega(a \mid s) \right) \right],$$

where $\beta$ controls trade-off between behaving like the dataset versus emphasizing higher-advantage actions.

**TD3 + BC (Fujimoto & Gu, 2021)**    We have two components: Q-function (critic) loss (using clipped double Q) and a policy loss that includes a behavior cloning penalty.

$$L_{\text{TD}}(\theta_i) = \mathbb{E}_{(s,a,r,s')\sim\mathcal{D}} \left[ \left( Q_{\theta_i}(s,a) - (r + \gamma \min_{j=1,2} Q_{\theta'_j}(s', \pi(s') + \epsilon)) \right)^2 \right]$$

where, $\epsilon \sim \text{clip}(\mathcal{N}(0,\sigma), -c, c)$, $\theta_i$ indexes one of the two Q-functions.

Policy objective:

$$L_{\pi_{\text{new}}}(\omega) = -\mathbb{E}_{(s,a)\sim\mathcal{D}} \left[ Q(s, \pi_\omega(s)) - \alpha \left\| \pi_\omega(s) - a \right\|^2 \right],$$

where $\alpha$ is the trade-off coefficient between maximizing expected return and staying close to behavior policy. Setting $\alpha = 0$ recovers standard TD3 policy loss (Fujimoto et al., 2018).

In all cases, the backbone (transformer + context) is shared. The Q- and V-head losses are backpropagated into the backbone. When a policy head is present, its gradient is detached from the backbone to improve stability.

## C  ADDITIONAL EXPERIMENTS WITH MIXTURE OF DYNAMICS

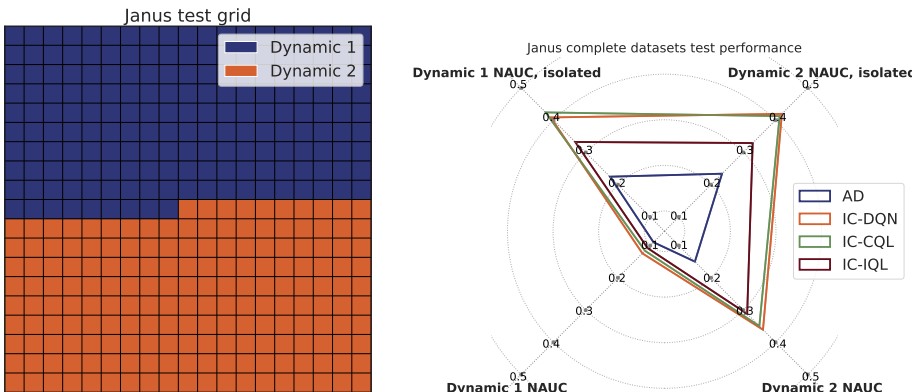

Figure 8: Left: Visualization of the Janus test environment. Right: NAUC scores averaged over all Janus datasets, with metrics reported separately for targets located in the first and second zones of the Janus environment. The top values show performance when trained agents are deployed in the standard DR19 environment, while the bottom values reflect performance when deployed in the Janus grid.

In this experiment, we investigate how algorithms perform when trained on environments with different dynamics and subsequently deployed into an environment featuring OOD dynamics. To this end, we introduce a modified version of the DR19 environment, called Janus[3]. In the Janus setup, learning histories are independently collected from two distinct instances of DR19, each governed by a different dynamic function (for example, actions in the second instance may map to inverted directions). Consequently, the training dataset includes examples of behavior under both dynamics, yet no single history contains a mixture of these dynamics. After training the agent on this combined dataset, we deploy it into a grid where the first half exhibits one dynamic and the second half the other, as illustrated in the left graph of Figure 8. The complete datasets are collected using the same configuration as for DR19, with the only modification being that for each learning history, the underlying environment dynamic is uniformly selected at random.

This experimental setup allows us to assess how effectively different approaches learn and generalize across multiple dynamics, as well as how they cope with an environment that blends these dynamics in a single deployment. Given the increased task complexity, we extended the number of rollout episodes to 200. The results, presented in the right plot of Figure 8, indicate that when an agent is trained on both dynamics and deployed into an environment featuring only one of them, its performance remains comparable across dynamics. However, when tested in an environment that combines both dynamics, performance decreases for both, with a more pronounced drop in one of the dynamics. This discrepancy is likely influenced by the asymmetry of the test environment, where the central starting position falls within the second dynamic. Ideally, a robust agent should perform uniformly well under both dynamics.

Across all conditions, RL-based approaches continue to demonstrate superiority over AD. In isolated tests, IQL outperforms AD by approximately 50%, while DQN and CQL achieve roughly twice the performance of AD. In the Janus environment, RL methods better preserve performance under one dynamic (as evidenced by the slopes of the performance lines) and exhibit slightly improved performance under the other dynamic. It is not surprising that offline RL counterparts do not provide benefits due to the fact that offline algorithms are supposed to avoid the OOD state-action pairs which are unavoidable in the Janus setup and considered offline approaches do not provide guarantees for this case. Tabular scores can be found in Appendix J.8.

---

[3]Ancient Roman two-faced god of duality.

## D  RL Tuple Ablation

In this section we provide an ablation on the addition of the *step* and *done* elements to the timestep tuples using K2D9 datasets with various expertise for three discrete RL algorithms we used in our study, i.e. IC-DQN (Table 2), IC-CQL (Table 3) and IC-IQL (Table 4). As we can see from the results, removing both augmentations leads to the worst performance (except for a couple of cases). Adding both is the best solution on average. Interestingly, with more data adding only steps is more beneficial than adding only dones and vise-versa with smaller datasets.

Table 2: NAUC scores on test targets with with and without *done* and *step* elements in the tuple for DQN. The best performance is marked with **bold** and 2-nd best with underline.

| Dataset | IC-DQN, w/o done, w/o step | IC-DQN, w/o done | IC-DQN, w/o step | IC-DQN |
|---|---|---|---|---|
| K2D9-250-1-early | $\underline{0.51} \pm 0.02$ | $0.30 \pm 0.07$ | $\mathbf{0.53} \pm 0.00$ | $0.41 \pm 0.06$ |
| K2D9-250-1-mid | $0.47 \pm 0.03$ | $\underline{0.53} \pm 0.02$ | $0.47 \pm 0.04$ | $\mathbf{0.62} \pm 0.01$ |
| K2D9-250-1-late | $0.07 \pm 0.00$ | $\underline{0.08} \pm 0.01$ | $\underline{0.08} \pm 0.01$ | $\mathbf{0.11} \pm 0.04$ |
| K2D9-500-1-early | $0.58 \pm 0.00$ | $0.59 \pm 0.02$ | $\underline{0.60} \pm 0.01$ | $\mathbf{0.61} \pm 0.01$ |
| K2D9-500-1-mid | $0.53 \pm 0.04$ | $\underline{0.74} \pm 0.06$ | $0.68 \pm 0.03$ | $\mathbf{0.78} \pm 0.03$ |
| K2D9-500-1-late | $0.14 \pm 0.01$ | $\underline{0.47} \pm 0.03$ | $0.18 \pm 0.01$ | $\mathbf{0.54} \pm 0.09$ |
| K2D9-1000-1-early | $0.58 \pm 0.01$ | $\mathbf{0.65} \pm 0.04$ | $0.61 \pm 0.00$ | $\underline{0.63} \pm 0.03$ |
| K2D9-1000-1-mid | $0.71 \pm 0.05$ | $\mathbf{0.83} \pm 0.01$ | $\underline{0.82} \pm 0.03$ | $0.81 \pm 0.04$ |
| K2D9-1000-1-late | $0.21 \pm 0.01$ | $\underline{0.48} \pm 0.00$ | $0.36 \pm 0.03$ | $\mathbf{0.61} \pm 0.06$ |
| Average | $0.32$ | $\underline{0.39}$ | $0.36$ | $\mathbf{0.43}$ |

Table 3: NAUC scores on test targets with with and without *done* and *step* elements in the tuple for CQL. The best performance is marked with **bold** and 2-nd best with underline.

| Dataset | IC-CQL, w/o done, w/o step | IC-CQL, w/o done | IC-CQL, w/o step | IC-CQL |
|---|---|---|---|---|
| K2D9-250-1-early | $\underline{0.23} \pm 0.03$ | $0.14 \pm 0.00$ | $\mathbf{0.27} \pm 0.03$ | $0.15 \pm 0.01$ |
| K2D9-250-1-mid | $0.60 \pm 0.01$ | $\underline{0.65} \pm 0.02$ | $0.64 \pm 0.02$ | $\mathbf{0.69} \pm 0.03$ |
| K2D9-250-1-late | $0.34 \pm 0.02$ | $0.29 \pm 0.02$ | $\mathbf{0.40} \pm 0.04$ | $\underline{0.35} \pm 0.01$ |
| K2D9-500-1-early | $\mathbf{0.52} \pm 0.01$ | $0.37 \pm 0.03$ | $\mathbf{0.52} \pm 0.01$ | $\underline{0.38} \pm 0.06$ |
| K2D9-500-1-mid | $0.77 \pm 0.01$ | $\underline{0.80} \pm 0.02$ | $\underline{0.80} \pm 0.02$ | $\mathbf{0.82} \pm 0.02$ |
| K2D9-500-1-late | $0.54 \pm 0.02$ | $\mathbf{0.71} \pm 0.05$ | $0.63 \pm 0.02$ | $\underline{0.70} \pm 0.03$ |
| K2D9-1000-1-early | $0.48 \pm 0.03$ | $\mathbf{0.62} \pm 0.05$ | $0.50 \pm 0.02$ | $\underline{0.58} \pm 0.02$ |
| K2D9-1000-1-mid | $0.80 \pm 0.02$ | $\mathbf{0.92} \pm 0.02$ | $0.86 \pm 0.02$ | $\underline{0.90} \pm 0.01$ |
| K2D9-1000-1-late | $0.72 \pm 0.02$ | $\mathbf{0.80} \pm 0.02$ | $0.71 \pm 0.02$ | $\underline{0.79} \pm 0.03$ |
| Average | $0.42$ | $\underline{0.44}$ | $\underline{0.44}$ | $\mathbf{0.45}$ |

Table 4: NAUC scores on test targets with with and without *done* and *step* elements in the tuple for IQL. The best performance is marked with **bold** and 2-nd best with underline.

| Dataset | IC-IQL, w/o done, w/o step | IC-IQL, w/o done | IC-IQL, w/o step | IC-IQL |
|---|---|---|---|---|
| K2D9-250-1-early | $\mathbf{0.20} \pm 0.01$ | $\underline{0.14} \pm 0.00$ | $\mathbf{0.20} \pm 0.00$ | $\underline{0.14} \pm 0.00$ |
| K2D9-250-1-mid | $0.52 \pm 0.01$ | $\underline{0.68} \pm 0.01$ | $0.59 \pm 0.02$ | $\mathbf{0.72} \pm 0.02$ |
| K2D9-250-1-late | $0.32 \pm 0.02$ | $0.32 \pm 0.01$ | $\underline{0.37} \pm 0.02$ | $\mathbf{0.39} \pm 0.01$ |
| K2D9-500-1-early | $\underline{0.39} \pm 0.03$ | $0.20 \pm 0.02$ | $\mathbf{0.46} \pm 0.03$ | $0.19 \pm 0.04$ |
| K2D9-500-1-mid | $0.73 \pm 0.03$ | $\underline{0.82} \pm 0.01$ | $0.76 \pm 0.01$ | $\mathbf{0.86} \pm 0.02$ |
| K2D9-500-1-late | $0.52 \pm 0.05$ | $\underline{0.69} \pm 0.02$ | $0.62 \pm 0.03$ | $\mathbf{0.71} \pm 0.01$ |
| K2D9-1000-1-early | $0.42 \pm 0.04$ | $\underline{0.48} \pm 0.04$ | $0.43 \pm 0.01$ | $\mathbf{0.50} \pm 0.03$ |
| K2D9-1000-1-mid | $0.82 \pm 0.02$ | $\mathbf{0.91} \pm 0.01$ | $\underline{0.86} \pm 0.01$ | $\mathbf{0.91} \pm 0.01$ |
| K2D9-1000-1-late | $0.72 \pm 0.01$ | $\underline{0.76} \pm 0.01$ | $0.73 \pm 0.02$ | $\mathbf{0.77} \pm 0.05$ |
| Average | $0.39$ | $\underline{0.42}$ | $\underline{0.42}$ | $\mathbf{0.43}$ |

## E  Future Work

Future work should extend this study by evaluating more complex environments such as NetHack (Küttler et al., 2020; Kurenkov et al., 2024) or more setups of XLand-MiniGrid (Nikulin et al.,

2023; 2024). Additionally, it is essential to explore ICRL in settings with visual observations, e.g. Meta-World (Yu et al., 2020), to further validate and generalize our findings.

Our approach can be further enhanced by incorporating several modifications that have proven effective for Transformer-based RL solutions. For example, integrating adjustments from methods like AMAGO and ReLIC (Grigsby et al., 2023; Elawady et al., 2024), employing N-gram heads (Akyürek et al., 2024; Zisman et al., 2024), or adopting mixture-of-experts (MoE) architectures (Shazeer et al., 2017; Obando-Ceron et al., 2024) could all contribute to improved performance. Additionally, replacing the regression loss used for value functions with a classification objective has demonstrated promising results (Farebrother et al., 2024) but should be carefully integrated in offline setup (Tarasov et al., 2024a). These potential enhancements underscore the flexibility of our framework and point to exciting avenues for future research.

It is also important to investigate usefulness of RL approaches in creating generalist models which are trained to operate in various environments which was recently done for Algorithm Distillation (Polubarov et al., 2025) or generalization to completely new environments as it was done by Raparthy et al. (2023).

Another promising direction for future work is to investigate the application of offline In-Context RL methods in an offline-to-online setting (Nair et al., 2020; Lee et al., 2022), where model weights are updated during rollouts. While this approach does not offer benefits when using the supervised objectives, the explicit RL objectives we optimize have the potential to further improve performance.

## F ADDITIONAL EXPERIMENTAL DETAILS

All experiments were conducted using NVIDIA H100 GPUs.

### F.1 IMPLEMENTATION DETAILS

Our implementation of Algorithm Distillation (AD) is based on the Decision Transformer (DT) codebase from Tarasov et al. (2024c). In our version, we remove the return-to-go input and merge state, action, and reward into a single token. This tokenization strategy reduces the overall Transformer sequence length, thereby decreasing both computation time and memory usage. When solving XLand-Minigrid we use similar implementation from Nikulin et al. (2024).

When adapting AD for Reinforcement Learning, we add value function and policy heads (for continuous problems) on top of the original AD backbone. The value heads heads are implemented as two-layer multilayer perceptrons (MLPs) with a Leaky ReLU activation function between layers. Policy heads are three-layer MLPs and analagous to AMAGO (Grigsby et al., 2023) we do not pass gradients to the Transformer backbone from these heads to improve training stability. There are also standard for RL target value function heads. To provide richer input information, we merge the additional *previous done* flag and step number with the (*state*, *previous action*, *previous reward*) token. In continuous environments Q value heads get the current *action* as additional input. For continuous IQL we also had to add LayerNorm (Ba et al., 2016) into the heads in order to stabilize learning process.

### F.2 HYPERPARAMETERS CHOICE

For hyperparameter tuning, we use the NAUC metric to select the best model configuration. The tuning is performed on the largest complete dataset, and the best hyperparameters are then applied across other datasets, even though this may result in suboptimal performance, especialy for offline RL approaches.

For AD, we tuned parameters that strongly influence performance, including attention dropout, embedding dropout, and residual dropout (each varied over values $\{0.1, 0.3, 0.5\}$), label smoothing for discrete environments (tested with values $\{0.1, 0.3\}$) and Transformer sequence length (tested over $\{100, 200\}$). This tuning resulted in 54 candidate hyperparameter sets. The best values, along with other general hyperparameter settings, are documented in Appendix H.

Due to computational constraints, the hyperparameters identified for AD were reused for the RL approaches. In the case of CQL, we tuned the discount factor $\gamma$ over values $\{0.8, 0.9, 0.95\}$ for

XLand-Minigrid and $\{0.7, 0.8, 0.9\}$ for other environments, and adjusted the CQL weight to be within $0.1, 0.3, 0.5$ for DR environments, within $0.3, 0.5, 1.0$ for Dark K2D environments, within $0.01, 0.05, 0.1$ for Janus, and within $0.01, 0.1, 0.5$ for XLand-Minigrid. For DQN, we simply adopted the $\gamma$ values found for CQL without additional tuning. Discrete IQL was tuned over a discount factor $\gamma$ with the same configuration as for CQL, an IQL parameter $\tau$ over $\{0.5, 0.7, 0.9\}$, and used a CQL weight of either $0.0$ or the best value found for CQL. For TD3+BC we tuned discount factor over $\{0.9, 0.95, 0.99\}$ and BC weight over $\{0.1, 0.3, 1.0\}$, for TD3 we reuse TD3+BC best discount factor and set BC weight to zero. For continuous IQL, we ran discount factor search over the same values as for TD3+BC, IQL $\tau$ over $\{0.5, 0.7, 0.9\}$ and IQL $\beta$ over $\{1, 3, 10\}$. This tuning yielded 9 candidate configurations for CQL and TD3+BC, 18 for discrete IQL, substantially fewer than those evaluated for AD. For continuous IQL, it resulted in 27 candidates, which is twice less than AD search space.

Each approach was trained over a fixed number of epochs to account for varying dataset sizes: 30 epochs for DR9, HPP and WLP, 15 epochs for HCV, 10 epochs for DR19, K2D9 and ANT, 6 epochs for K2D13. We track metrics after each epoch and report the mean value across multiple seeds for the epoch according to the best NAUC value. Notably, we observed that AD exhibits greater instability during training compared to the RL approaches, meaning that our design choices in the experimental protocol tend to favor AD, yet its performance remains inferior.

Preliminary experiments indicated that an important hyperparameter controlling AD subsampling is best set to 4 for DR and continuous environments and 8 for K2D. When evaluating on incomplete datasets, we reduced these values to 1 for DR and continuous environments and to 2 for K2D to maintain a consistent number of trajectories. In Section 4.4 the subsample parameter was set to 1, as it does not have any motivation there and would just discard a large number of trajectories.

For a complete list of hyperparameter values, please refer to Appendix H.

## G  DATASETS DETAILS

In this section, we describe the data collection process and provide detailed statistics for each of the obtained datasets. We do not provide data for Janus datasets as they are very similar to DR19.

### G.1  DATA COLLECTION

**Discrete Environments.** To construct the complete discrete datasets, we employed a tabular Q-learning algorithm (Watkins & Dayan, 1992) with a linearly decayed $\epsilon$-greedy exploration strategy. For the K2D environments, which are originally formulated as POMDPs and require memory to solve, we doubled the state space by mapping each grid position to two distinct states: one corresponding to the scenario where the key has not been collected and the other where it has. This transformation effectively converts K2D into a fully observable MDP. The hyperparameter values used for Q-learning across all dataset collections are provided in Table 5.

Table 5: Q-learning hyperparameters for DR and K2D data collection.

| Hyperparameter | Value |
|---|---|
| Learning rate | 0.9933 |
| Discount factor ($\gamma$) | 0.9 |
| Number of episodes | 200 |

**Continous Environments.** For collecting the learning histories in continuous environments we used SAC implementation from Clean RL (Huang et al., 2022). We kept most of the SAC hyperparameters default and we present the varied subset in Table 6.

Datasets representing various levels of expertise are derived by segmenting the complete learning histories into three equal parts on a trajectory-wise basis.

Table 6: SAC hyperparameters for HCV, ANT, HPP and WLP data collection.

| Hyperparameter | Value |
|---|---|
| Critic learning rate | 3e-4 for HCV and ANT |
| | 1e-4 for HPP and WLP |
| Actor learning rate | 3e-4 for HCV and ANT |
| | 1e-4 for HPP and WLP |
| Discount factor ($\gamma$) | 0.99 |
| Number of timesteps | 100000 for HCV |
| | 400000 for ANT |
| | 10000 for HPP and WLP |
| Warm-up timesteps | 2000 |

## G.2 LEARNING CURVES

The learning curves presented in the following graphs illustrate the average returns for all datasets as a function of the episode number within the learning history. By concatenating the `early`, `mid`, and `late` segments, we obtain the complete dataset curves. We do not provide curves for the HPP and WLP due to the variable amount of episodes for each learning history.

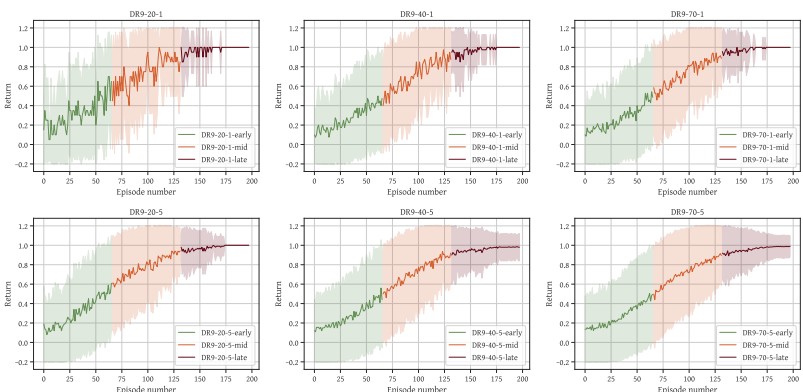

Figure 9: Q-learning DR9 learning curves.

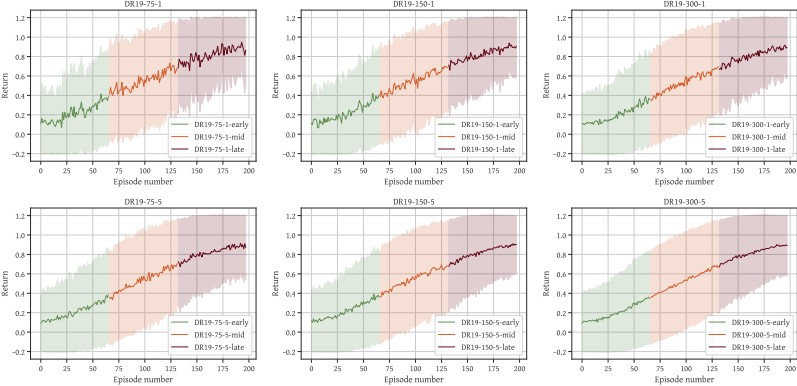

Figure 10: Q-learning DR19 learning curves.

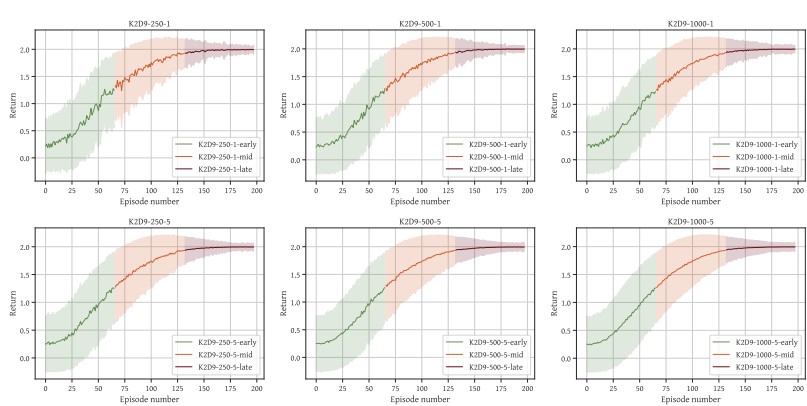

Figure 11: Q-learning K2D9 learning curves.

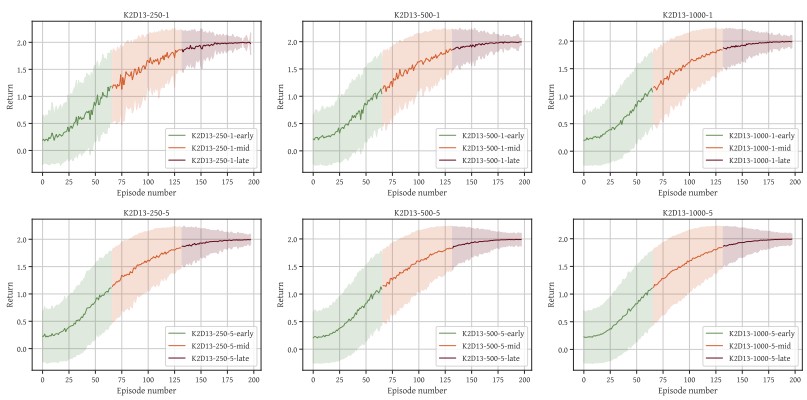

Figure 12: Q-learning DR13 learning curves.

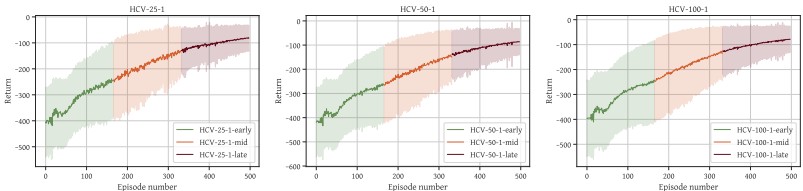

Figure 13: SAC HCV learning curves.

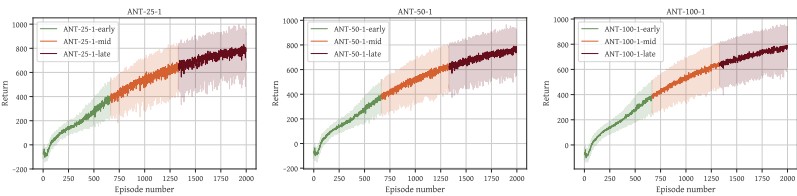

Figure 14: SAC ANT learning curves.

### G.3 DATASETS STATISTICS

Table 7: XLand-MiniGrid trivial `tiny` statistics.

| Dataset | Size | Mean trajectory length | Mean trajectory return | Success rate |
|---|---|---|---|---|
| XLand-MiniGrid trivial tiny | 203090000 | 129.77 | 0.48 | 0.58 |

Table 8: DR9 datasets statistics.

| Dataset | Size | Mean trajectory length | Mean trajectory return | Success rate |
|---|---|---|---|---|
| DR9-20-1-early | 21218 | 16.07 | 0.32 | 0.32 |
| DR9-20-1-mid | 13155 | 9.97 | 0.74 | 0.74 |
| DR9-20-1-late | 6497 | 4.92 | 0.98 | 0.98 |
| DR9-20-1 | 41126 | 10.28 | 0.69 | 0.69 |
| DR9-20-5-early | 107351 | 16.27 | 0.31 | 0.31 |
| DR9-20-5-mid | 62630 | 9.49 | 0.78 | 0.78 |
| DR9-20-5-late | 31339 | 4.75 | 0.98 | 0.98 |
| DR9-20-5 | 202655 | 10.13 | 0.69 | 0.69 |
| DR9-40-1-early | 44329 | 16.79 | 0.27 | 0.27 |
| DR9-40-1-mid | 28585 | 10.83 | 0.72 | 0.72 |
| DR9-40-1-late | 14164 | 5.37 | 0.98 | 0.98 |
| DR9-40-1 | 87673 | 10.96 | 0.66 | 0.66 |
| DR9-40-5-early | 221731 | 16.80 | 0.27 | 0.27 |
| DR9-40-5-mid | 139820 | 10.59 | 0.73 | 0.73 |
| DR9-40-5-late | 72883 | 5.52 | 0.96 | 0.96 |
| DR9-40-5 | 437283 | 10.93 | 0.65 | 0.65 |
| DR9-70-1-early | 77758 | 16.83 | 0.27 | 0.27 |
| DR9-70-1-mid | 48926 | 10.59 | 0.74 | 0.74 |
| DR9-70-1-late | 24940 | 5.40 | 0.98 | 0.98 |
| DR9-70-1 | 152609 | 10.90 | 0.66 | 0.66 |
| DR9-70-5-early | 388911 | 16.84 | 0.27 | 0.27 |
| DR9-70-5-mid | 246344 | 10.66 | 0.73 | 0.73 |
| DR9-70-5-late | 127897 | 5.54 | 0.96 | 0.96 |
| DR9-70-5 | 768396 | 10.98 | 0.65 | 0.65 |

Table 9: DR19 datasets statistics.

| Dataset | Size | Mean trajectory length | Mean trajectory return | Success rate |
|---|---|---|---|---|
| DR19-75-1-early | 420339 | 84.92 | 0.22 | 0.22 |
| DR19-75-1-mid | 280925 | 56.75 | 0.55 | 0.55 |
| DR19-75-1-late | 146170 | 29.53 | 0.82 | 0.82 |
| DR19-75-1 | 853890 | 56.93 | 0.53 | 0.53 |
| DR19-75-5-early | 2131932 | 86.14 | 0.21 | 0.21 |
| DR19-75-5-mid | 1441194 | 58.23 | 0.54 | 0.54 |
| DR19-75-5-late | 733739 | 29.65 | 0.82 | 0.82 |
| DR19-75-5 | 4338440 | 57.85 | 0.52 | 0.52 |
| DR19-150-1-early | 844293 | 85.28 | 0.22 | 0.22 |
| DR19-150-1-mid | 562681 | 56.84 | 0.55 | 0.55 |
| DR19-150-1-late | 289073 | 29.20 | 0.82 | 0.82 |
| DR19-150-1 | 1708314 | 56.94 | 0.53 | 0.53 |
| DR19-150-5-early | 4225426 | 85.36 | 0.22 | 0.22 |
| DR19-150-5-mid | 2819202 | 56.95 | 0.55 | 0.55 |
| DR19-150-5-late | 1455768 | 29.41 | 0.82 | 0.82 |
| DR19-150-5 | 8563461 | 57.09 | 0.53 | 0.53 |
| DR19-300-1-early | 1719628 | 86.85 | 0.20 | 0.20 |
| DR19-300-1-mid | 1161116 | 58.64 | 0.53 | 0.53 |
| DR19-300-1-late | 604069 | 30.51 | 0.81 | 0.81 |
| DR19-300-1 | 3511094 | 58.52 | 0.52 | 0.52 |
| DR19-300-5-early | 8584583 | 86.71 | 0.20 | 0.20 |
| DR19-300-5-mid | 5870842 | 59.30 | 0.53 | 0.53 |
| DR19-300-5-late | 2993142 | 30.23 | 0.81 | 0.81 |
| DR19-300-5 | 17580304 | 58.60 | 0.52 | 0.52 |

Table 10: K2D9 datasets statistics.

| Dataset | Size | Mean trajectory length | Mean trajectory return | Success rate |
|---|---|---|---|---|
| K2D9-250-1-early | 776325 | 47.05 | 0.65 | 0.14 |
| K2D9-250-1-mid | 505820 | 30.66 | 1.68 | 0.70 |
| K2D9-250-1-late | 244008 | 14.79 | 1.98 | 0.98 |
| K2D9-250-1 | 1536153 | 30.72 | 1.44 | 0.61 |
| K2D9-250-5-early | 3887625 | 47.12 | 0.65 | 0.14 |
| K2D9-250-5-mid | 2535825 | 30.74 | 1.68 | 0.70 |
| K2D9-250-5-late | 1227843 | 14.88 | 1.98 | 0.98 |
| K2D9-250-5 | 7701588 | 30.81 | 1.44 | 0.61 |
| K2D9-500-1-early | 1560936 | 47.30 | 0.64 | 0.13 |
| K2D9-500-1-mid | 1021017 | 30.94 | 1.67 | 0.70 |
| K2D9-500-1-late | 494081 | 14.97 | 1.98 | 0.98 |
| K2D9-500-1 | 3095911 | 30.96 | 1.44 | 0.61 |
| K2D9-500-5-early | 7800316 | 47.27 | 0.64 | 0.13 |
| K2D9-500-5-mid | 5066158 | 30.70 | 1.68 | 0.70 |
| K2D9-500-5-late | 2478549 | 15.02 | 1.98 | 0.98 |
| K2D9-500-5 | 15446515 | 30.89 | 1.44 | 0.61 |
| K2D9-1000-1-early | 3127972 | 47.39 | 0.64 | 0.12 |
| K2D9-1000-1-mid | 2046177 | 31.00 | 1.68 | 0.70 |
| K2D9-1000-1-late | 994116 | 15.06 | 1.98 | 0.98 |
| K2D9-1000-1 | 6208094 | 31.04 | 1.44 | 0.60 |
| K2D9-1000-5-early | 15621614 | 47.34 | 0.64 | 0.13 |
| K2D9-1000-5-mid | 10132408 | 30.70 | 1.69 | 0.70 |
| K2D9-1000-5-late | 4941080 | 14.97 | 1.98 | 0.98 |
| K2D9-1000-5 | 30894006 | 30.89 | 1.44 | 0.61 |

Table 11: K2D13 datasets statistics.

| Dataset | Size | Mean trajectory length | Mean trajectory return | Success rate |
|---|---|---|---|---|
| K2D13-250-1-early | 1574100 | 95.40 | 0.57 | 0.10 |
| K2D13-250-1-mid | 1052247 | 63.77 | 1.55 | 0.60 |
| K2D13-250-1-late | 487589 | 29.55 | 1.95 | 0.95 |
| K2D13-250-1 | 3137407 | 62.75 | 1.36 | 0.55 |
| K2D13-250-5-early | 7869792 | 95.39 | 0.57 | 0.10 |
| K2D13-250-5-mid | 5223538 | 63.32 | 1.56 | 0.61 |
| K2D13-250-5-late | 2391714 | 28.99 | 1.95 | 0.95 |
| K2D13-250-5 | 15593413 | 62.37 | 1.36 | 0.56 |
| K2D13-500-1-early | 3151770 | 95.51 | 0.56 | 0.10 |
| K2D13-500-1-mid | 2125016 | 64.39 | 1.54 | 0.59 |
| K2D13-500-1-late | 986467 | 29.89 | 1.95 | 0.95 |
| K2D13-500-1 | 6309472 | 63.09 | 1.36 | 0.55 |
| K2D13-500-5-early | 15777391 | 95.62 | 0.56 | 0.10 |
| K2D13-500-5-mid | 10642766 | 64.50 | 1.54 | 0.59 |
| K2D13-500-5-late | 4846718 | 29.37 | 1.95 | 0.95 |
| K2D13-500-5 | 31478474 | 62.96 | 1.36 | 0.55 |
| K2D13-1000-1-early | 6301009 | 95.47 | 0.56 | 0.10 |
| K2D13-1000-1-mid | 4188247 | 63.46 | 1.55 | 0.61 |
| K2D13-1000-1-late | 1930904 | 29.26 | 1.95 | 0.95 |
| K2D13-1000-1 | 12508282 | 62.54 | 1.36 | 0.56 |
| K2D13-1000-5-early | 31547429 | 95.60 | 0.56 | 0.10 |
| K2D13-1000-5-mid | 21152907 | 64.10 | 1.55 | 0.60 |
| K2D13-1000-5-late | 9596737 | 29.08 | 1.95 | 0.95 |
| K2D13-1000-5 | 62722139 | 62.72 | 1.36 | 0.55 |

Table 12: HCV datasets statistics.

| Dataset | Size | Mean trajectory length | Mean trajectory return |
|---|---|---|---|
| HCV-25-1-early | 830000 | 200.00 | -317.43 |
| HCV-25-1-mid | 830000 | 200.00 | -183.93 |
| HCV-25-1-late | 830000 | 200.00 | -102.40 |
| HCV-25-1 | 2500000 | 200.00 | -200.97 |
| HCV-50-1-early | 1660000 | 200.00 | -332.00 |
| HCV-50-1-mid | 1660000 | 200.00 | -198.24 |
| HCV-50-1-late | 1660000 | 200.00 | -109.20 |
| HCV-50-1 | 5000000 | 200.00 | -212.86 |
| HCV-100-1-early | 3320000 | 200.00 | -311.78 |
| HCV-100-1-mid | 3320000 | 200.00 | -182.78 |
| HCV-100-1-late | 3320000 | 200.00 | -99.38 |
| HCV-100-1 | 10000000 | 200.00 | -197.70 |

Table 13: ANT datasets statistics.

| Dataset | Size | Mean trajectory length | Mean trajectory return |
|---|---|---|---|
| ANT-25-1-early | 3330000 | 200.00 | 173.81 |
| ANT-25-1-mid | 3330000 | 200.00 | 530.98 |
| ANT-25-1-late | 3330000 | 200.00 | 725.87 |
| ANT-25-1 | 10000000 | 200.00 | 477.06 |
| ANT-50-1-early | 6660000 | 200.00 | 174.28 |
| ANT-50-1-mid | 6660000 | 200.00 | 512.84 |
| ANT-50-1-late | 6660000 | 200.00 | 699.77 |
| ANT-50-1 | 20000000 | 200.00 | 462.46 |
| ANT-100-1-early | 13320000 | 200.00 | 176.09 |
| ANT-100-1-mid | 13320000 | 200.00 | 525.65 |
| ANT-100-1-late | 13320000 | 200.00 | 714.36 |
| ANT-100-1 | 40000000 | 200.00 | 472.20 |

Table 14: HPP datasets statistics.

| Dataset | Size | Mean trajectory length | Mean trajectory return |
|---|---|---|---|
| HPP-25-1-early | 37549 | 21.35 | 17.11 |
| HPP-25-1-mid | 49040 | 27.88 | 33.49 |
| HPP-25-1-late | 160671 | 91.34 | 164.30 |
| HPP-25-1 | 248832 | 46.89 | 71.72 |
| HPP-50-1-early | 75071 | 21.70 | 17.64 |
| HPP-50-1-mid | 97406 | 28.15 | 33.86 |
| HPP-50-1-late | 322563 | 93.23 | 166.08 |
| HPP-50-1 | 497329 | 47.69 | 72.56 |
| HPP-100-1-early | 151372 | 21.90 | 17.73 |
| HPP-100-1-mid | 196313 | 28.40 | 34.34 |
| HPP-100-1-late | 642446 | 92.93 | 169.42 |
| HPP-100-1 | 994458 | 47.73 | 73.85 |

Table 15: WLP datasets statistics.

| Dataset | Size | Mean trajectory length | Mean trajectory return |
|---|---|---|---|
| WLP-25-1-early | 32114 | 20.92 | 3.30 |
| WLP-25-1-mid | 35679 | 23.24 | 6.51 |
| WLP-25-1-late | 179708 | 117.07 | 122.93 |
| WLP-25-1 | 248319 | 53.67 | 44.19 |
| WLP-50-1-early | 64537 | 21.08 | 3.61 |
| WLP-50-1-mid | 72339 | 23.62 | 7.14 |
| WLP-50-1-late | 357409 | 116.72 | 120.60 |
| WLP-50-1 | 496424 | 53.74 | 43.67 |
| WLP-100-1-early | 131731 | 21.05 | 4.10 |
| WLP-100-1-mid | 150639 | 24.08 | 9.28 |
| WLP-100-1-late | 706669 | 112.94 | 121.11 |
| WLP-100-1 | 992983 | 52.61 | 44.73 |

# H HYPERPARAMETERS

Table 16: AD general hyperparameters. The approximate model size is 25,000,000 parameters for XLand-MiniGrid and 12,500,000 parameters for other environments.

| | Hyperparameter | Value |
|---|---|---|
| AD hyperparameters | Optimizer | Adam (Kingma, 2014) |
| | Batch size | 512 |
| | Sequence length | 100, HCV, ANT |
| | | 120, DR9 |
| | | 200, K2D9, HPP, WLP |
| | | 400, DR19 and K2D19 |
| | | 512, XLand-MiniGrid |
| | Learning rate | 0.0003 |
| | Learning schedule | Constant |
| | Adam betas | (0.9, 0.99) |
| | Clip grad norm | 1.0 |
| | Weight decay | 0.0 |
| | Subsample | 4, for complete ANT and DR datasets |
| | | 1, for incomplete DR and continuous datasets |
| | | 8, for complete K2D datasets |
| | | 2, for incomplete K2D and complete continuous (except ANT) datasets |
| | | 0.5, for XLand-MiniGrid (XLand-MiniGrid AD implementation uses different subsample strategy.) |
| Architecture | Number of layers | 8, for XLand-MiniGrid |
| | | 4, otherwise |
| | Number of attention heads | 8, for XLand-MiniGrid |
| | | 4, otherwise |
| | Embedding dimension | 512 |
| | Activation function | GELU |

Table 17: AD per-environment tuned parameters.

| Environment | Attention Dropout | Embedding Dropout | Residual Dropout | Label Smoothing |
|---|---|---|---|---|
| DR9 | 0.5 | 0.1 | 0.1 | 0.3 |
| DR19 | 0.5 | 0.5 | 0.3 | 0.1 |
| K2D9 | 0.1 | 0.5 | 0.1 | 0.3 |
| K2D19 | 0.1 | 0.5 | 0.1 | 0.3 |
| Janus | 0.5 | 0.5 | 0.1 | 0.3 |
| XLand-MiniGrid trivial tiny | 0.1 | 0.5 | 0.1 | 0.3 |
| HCV | 0.1 | 0.1 | 0.5 | - |
| ANT | 0.3 | 0.1 | 0.1 | - |
| HPP | 0.3 | 0.3 | 0.3 | - |
| WLP | 0.1 | 0.5 | 0.5 | - |
| MW-DR9 | 0.5 | 0.1 | 0.3 | 0.3 |

Table 18: CQL per-environment tuned parameters.

| Environment | Discount Factor $\gamma$ | CQL weight |
|---|---|---|
| DR9 | 0.7 | 0.3 |
| DR19 | 0.8 | 0.3 |
| K2D9 | 0.7 | 1.0 |
| K2D19 | 0.7 | 1.0 |
| Janus | 0.8 | 0.01 |
| XLand-MiniGrid trivial tiny | 0.9 | 0.01 |
| MW-DR9 | 0.9 | 0.5 |

Table 19: Discrete IQL per-environment tuned parameters.

| Environment | Discount Factor $\gamma$ | CQL weight | IQL $\tau$ |
|---|---|---|---|
| DR9 | 0.7 | 0.3 | 0.9 |
| DR19 | 0.8 | 0.0 | 0.7 |
| K2D9 | 0.7 | 1.0 | 0.5 |
| K2D19 | 0.7 | 1.0 | 0.7 |
| Janus | 0.8 | 0.01 | 0.7 |
| XLand-MiniGrid trivial tiny | 0.9 | 0.0 | 0.9 |
| MW-DR9 | 0.9 | 0.5 | 0.7 |

Table 20: TD3+BC per-environment tuned parameters.

| Environment | Discount Factor $\gamma$ | BC weight |
|---|---|---|
| HCV | 0.9 | 0.3 |
| ANT | 0.9 | 1.0 |
| HPP | 0.99 | 1.0 |
| WLP | 0.99 | 1.0 |

# I    ADDITIONAL PLOTS AND METRICS

## I.1    OVERALL PERFORMANCE

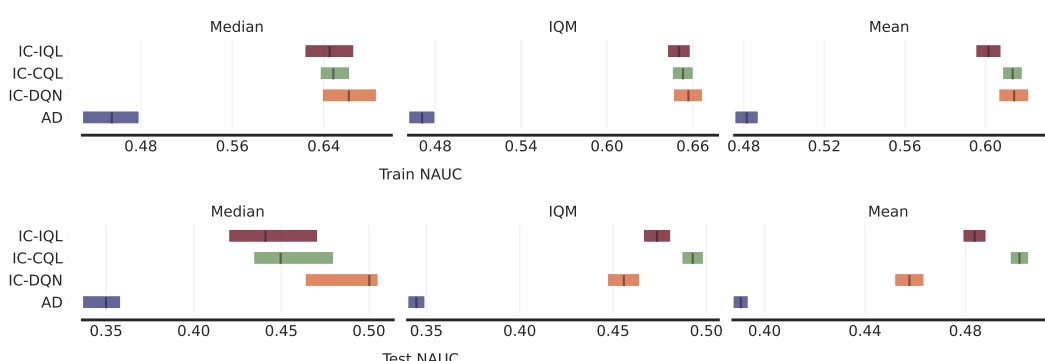

Figure 15: Median, IQM and mean of NAUC computed with rliable approach across all available discrete datasets. Top: train targets. Bottom: test targets.

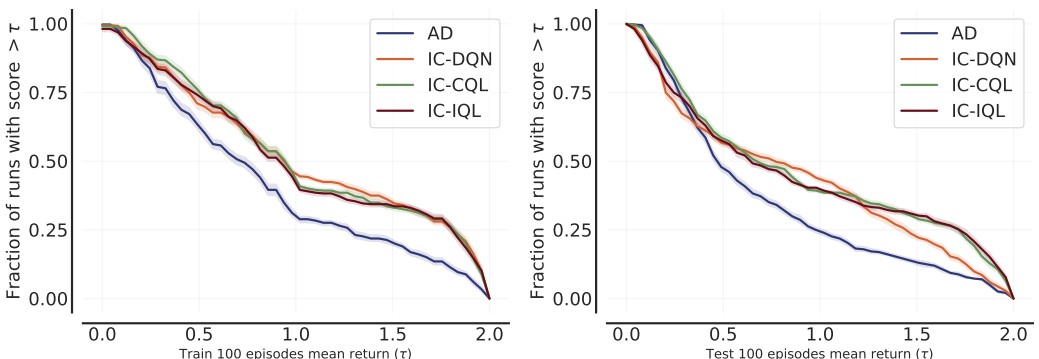

Figure 16: rliable performance profiles of the 100th episode scores across all available discrete datasets. Left: train targets. Right: test targets.

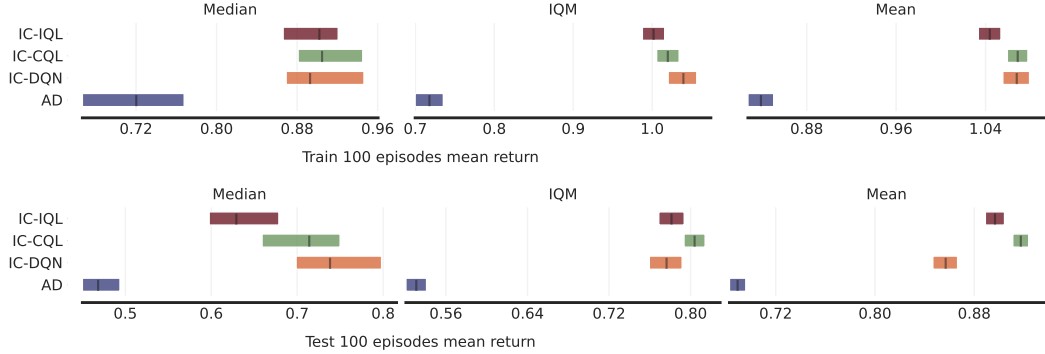

Figure 17: Median, IQM and mean of 100th episode scores computed with rliable approach across all available discrete datasets. Top: train targets. Bottom: test targets.

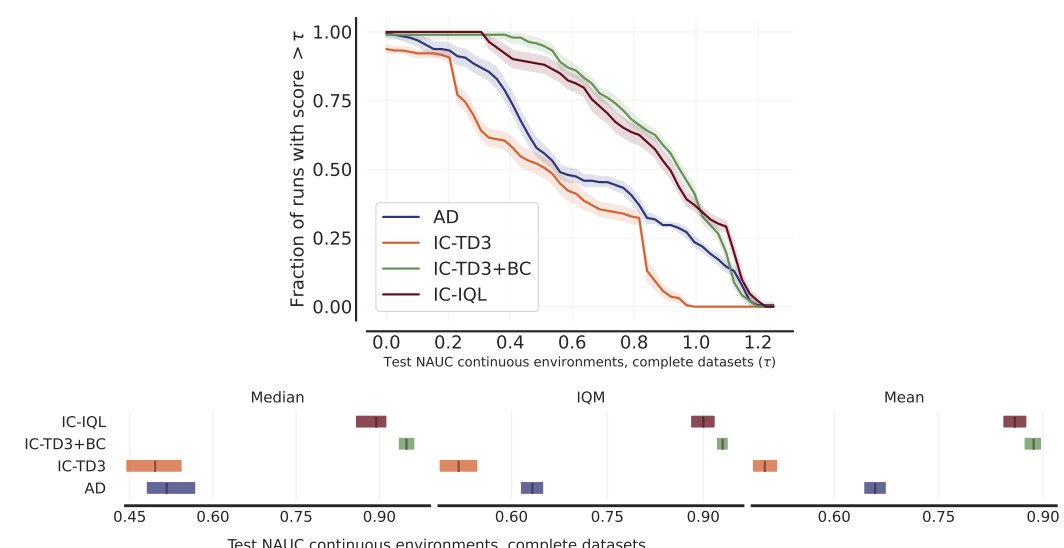

Figure 18: rliable metrics of the NAUC across all available continuous datasets. Top: performance profiles. Bottom: median, IQM and mean.

## I.2 VARIOUS EXPERTISE PERFORMANCE

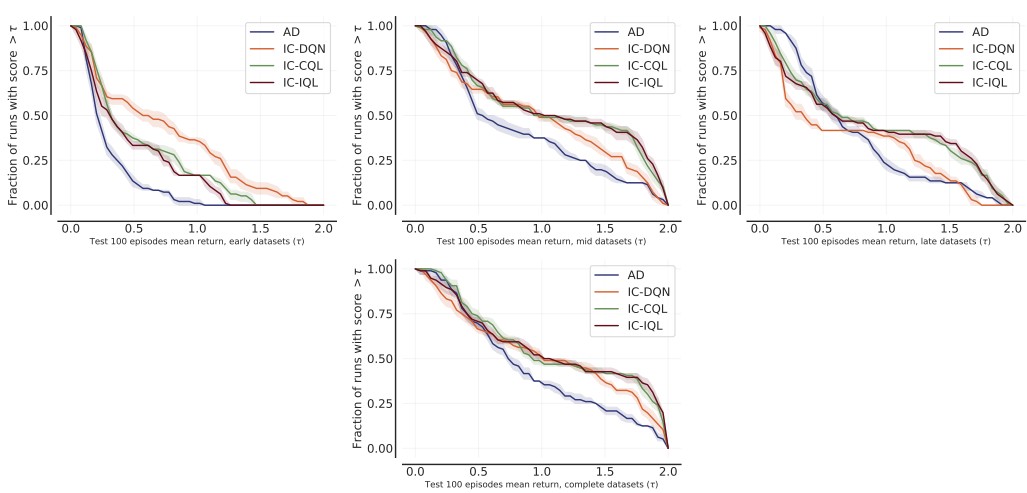

Figure 19: rliable performance profiles of 100th episode scores for various discrete datasets expertise. Top, from left to right: `early`, `mid`, `late` datasets. Bottom: complete learning histories.

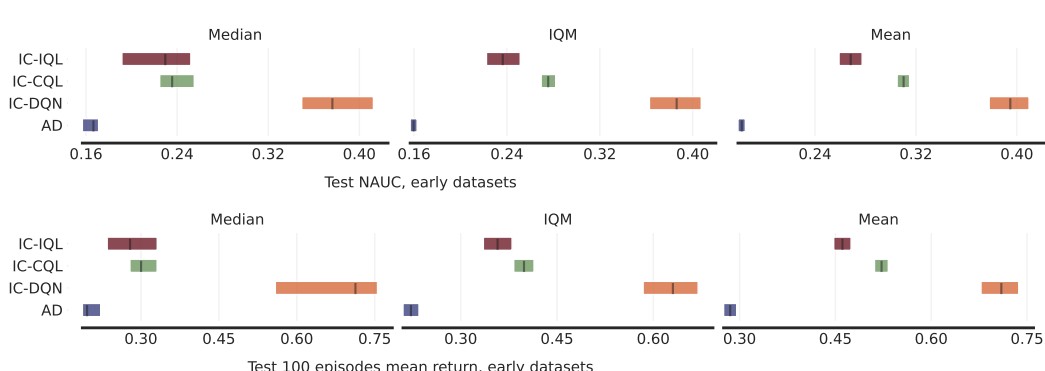

Figure 20: Median, IQM and mean of NAUC and 100th episode scores computed with rliable approach across `early` discrete datasets for test targets. Top: NAUC. Bottom: 100th episode performance.

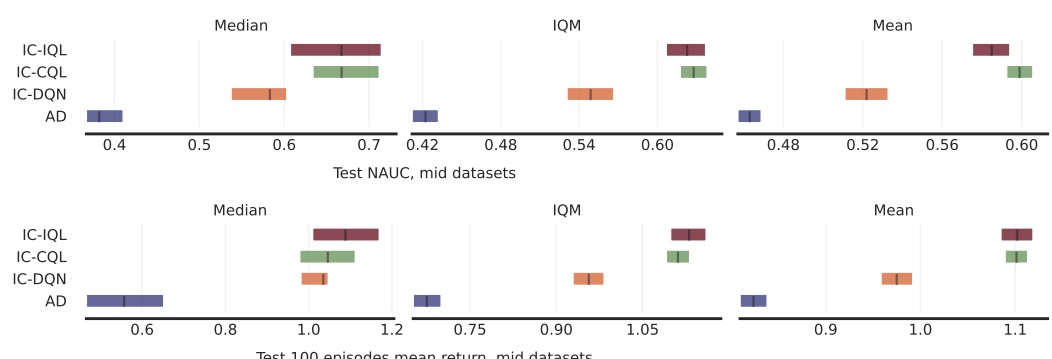

Figure 21: Median, IQM and mean of NAUC and 100th episode scores computed with rliable approach across `mid` discrete datasets for test targets. Top: NAUC. Bottom: 100th episode performance.

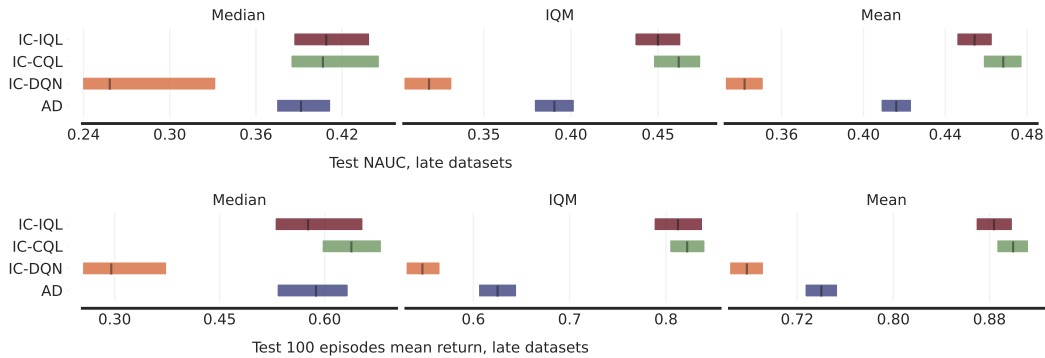

Figure 22: Median, IQM and mean of NAUC and 100th episode scores computed with rliable approach across `late` discrete datasets for test targets. Top: NAUC. Bottom: 100th episode performance.

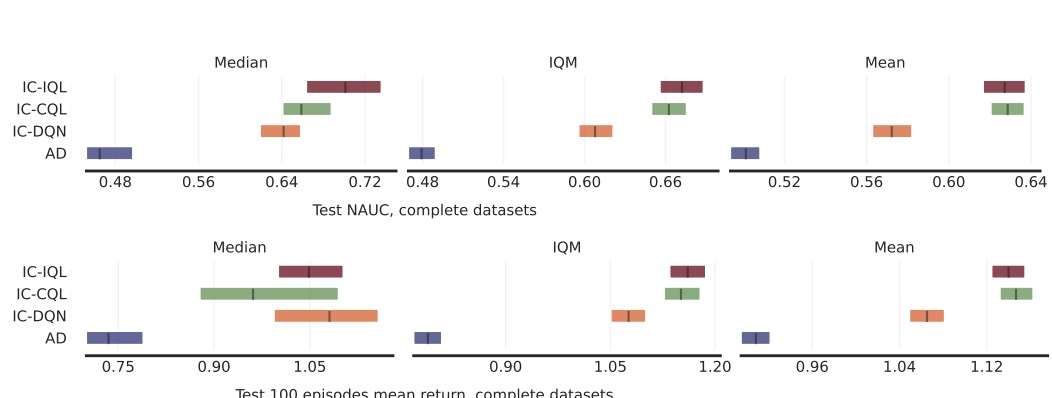

Figure 23: Median, IQM and mean of NAUC and 100th episode scores computed with rliable approach across complete discrete datasets for test targets. Top: NAUC. Bottom: 100th episode performance.

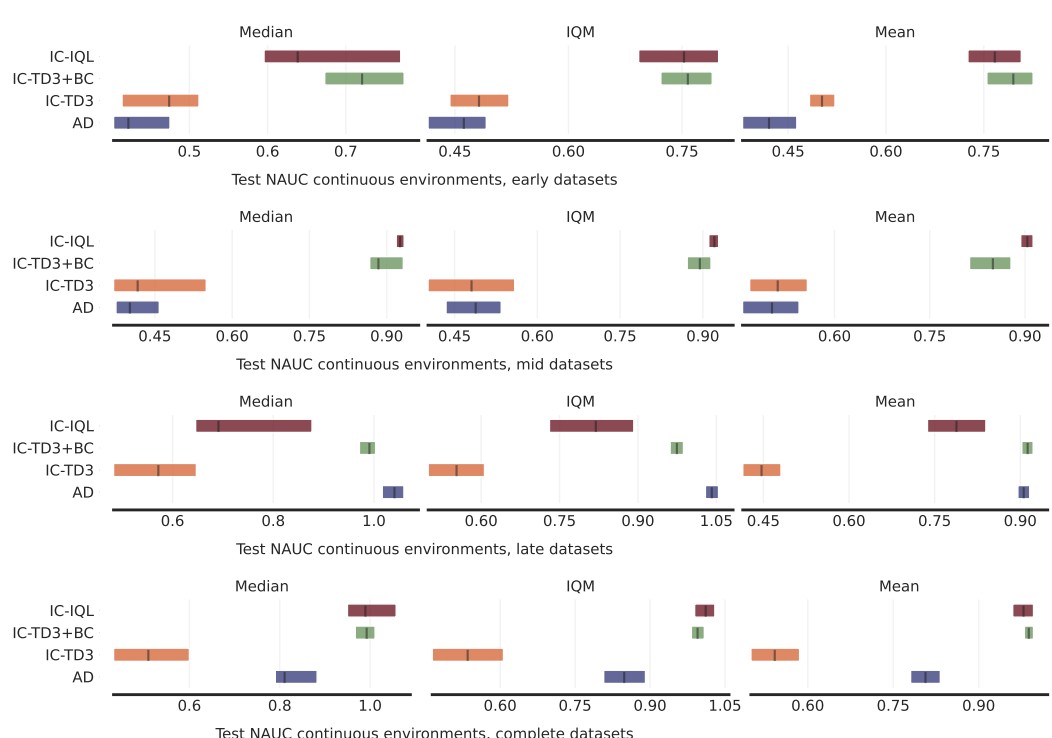

Figure 24: Median, IQM and mean of NAUC computed with rliable approach across continous datasets for test instances.

### I.3 No Learning Histories Structure

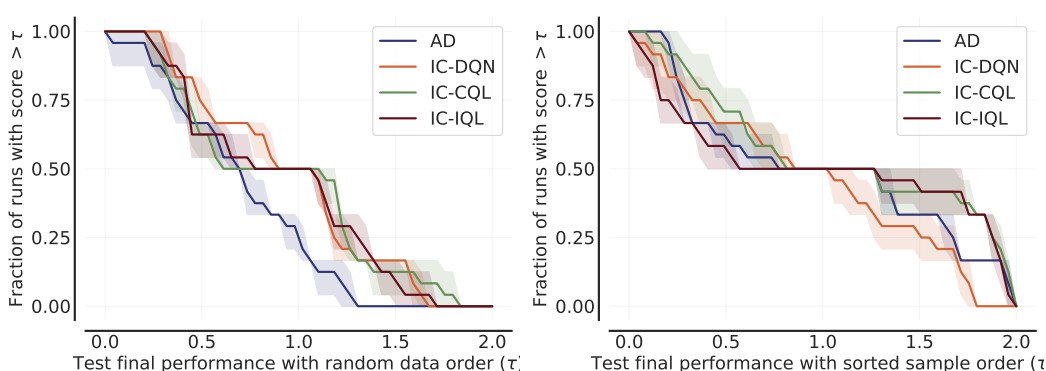

Figure 25: rliable performance profiles of the 100th episode scores on test targets across all DR19 and K2D13 complete datasets without learning histories. Left: random trajectories order. Right: trajectories sorted within random subset.

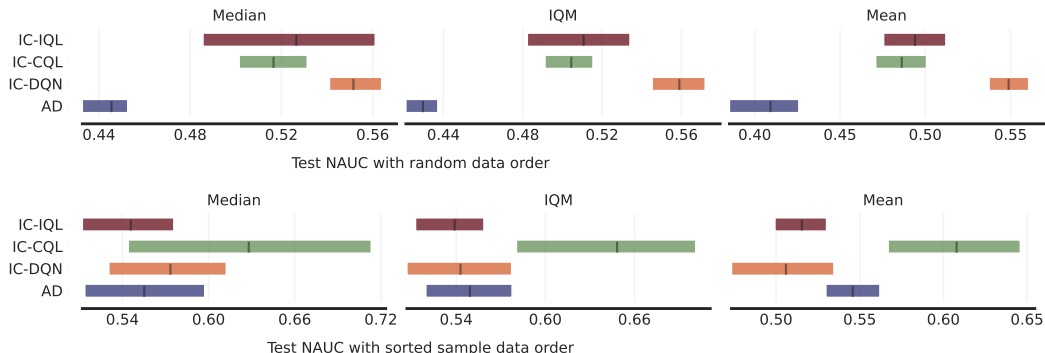

Figure 26: Median, IQM and mean of NAUC for test targets computed with rliable approach across all DR19 and K2D13 complete datasets without learning histories. Top: random trajectories order. Bottom: trajectories sorted within random subset.

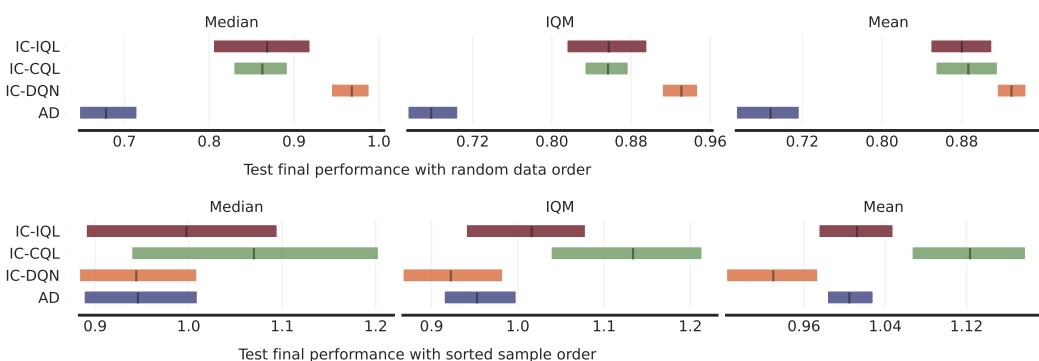

Figure 27: Median, IQM and mean of the 100th episode scores for test targets computed with rliable approach across all DR19 and K2D13 complete datasets without learning histories. Top: random trajectories order. Bottom: trajectories sorted within random subset.

## J  TABULAR RESULTS

In this section, we provide numerical values of NAUC and final performance (after 100 episodes) for all methods. Means and stds are computed over 4 random seeds.

### J.1  DISCRETE TRAIN NAUC

Table 21: Train targets performances measured with NAUC on `early` datasets.

| Dataset | AD | IC-DQN | IC-CQL | IC-IQL |
|---|---|---|---|---|
| DR9-20-1-early | $0.25 \pm 0.03$ | $0.26 \pm 0.04$ | $0.31 \pm 0.04$ | $0.33 \pm 0.02$ |
| DR9-20-5-early | $0.23 \pm 0.00$ | $0.64 \pm 0.08$ | $0.63 \pm 0.05$ | $0.55 \pm 0.01$ |
| DR9-40-1-early | $0.21 \pm 0.00$ | $0.42 \pm 0.09$ | $0.35 \pm 0.05$ | $0.37 \pm 0.04$ |
| DR9-40-5-early | $0.30 \pm 0.04$ | $0.72 \pm 0.10$ | $0.67 \pm 0.04$ | $0.75 \pm 0.01$ |
| DR9-70-1-early | $0.24 \pm 0.04$ | $0.56 \pm 0.14$ | $0.42 \pm 0.03$ | $0.34 \pm 0.16$ |
| DR9-70-5-early | $0.29 \pm 0.03$ | $0.81 \pm 0.07$ | $0.76 \pm 0.05$ | $0.64 \pm 0.04$ |
| DR19-75-1-early | $0.12 \pm 0.01$ | $0.25 \pm 0.07$ | $0.13 \pm 0.02$ | $0.18 \pm 0.06$ |
| DR19-75-5-early | $0.12 \pm 0.01$ | $0.60 \pm 0.05$ | $0.35 \pm 0.03$ | $0.32 \pm 0.08$ |
| DR19-150-1-early | $0.11 \pm 0.01$ | $0.30 \pm 0.06$ | $0.22 \pm 0.06$ | $0.16 \pm 0.06$ |
| DR19-150-5-early | $0.12 \pm 0.01$ | $0.65 \pm 0.04$ | $0.37 \pm 0.04$ | $0.29 \pm 0.18$ |
| DR19-300-1-early | $0.12 \pm 0.01$ | $0.31 \pm 0.08$ | $0.19 \pm 0.03$ | $0.03 \pm 0.04$ |
| DR19-300-5-early | $0.11 \pm 0.02$ | $0.68 \pm 0.23$ | $0.40 \pm 0.03$ | $0.33 \pm 0.19$ |
| K2D9-250-1-early | $0.13 \pm 0.01$ | $0.48 \pm 0.09$ | $0.14 \pm 0.01$ | $0.13 \pm 0.01$ |
| K2D9-250-5-early | $0.24 \pm 0.02$ | $0.77 \pm 0.04$ | $0.65 \pm 0.01$ | $0.61 \pm 0.02$ |
| K2D9-500-1-early | $0.16 \pm 0.01$ | $0.61 \pm 0.03$ | $0.41 \pm 0.04$ | $0.23 \pm 0.08$ |
| K2D9-500-5-early | $0.36 \pm 0.02$ | $0.84 \pm 0.08$ | $0.62 \pm 0.02$ | $0.57 \pm 0.04$ |
| K2D9-1000-1-early | $0.20 \pm 0.01$ | $0.71 \pm 0.09$ | $0.57 \pm 0.03$ | $0.53 \pm 0.05$ |
| K2D9-1000-5-early | $0.42 \pm 0.01$ | $0.87 \pm 0.04$ | $0.63 \pm 0.01$ | $0.55 \pm 0.02$ |
| K2D13-250-1-early | $0.12 \pm 0.00$ | $0.11 \pm 0.02$ | $0.12 \pm 0.00$ | $0.12 \pm 0.01$ |
| K2D13-250-5-early | $0.14 \pm 0.01$ | $0.68 \pm 0.03$ | $0.42 \pm 0.01$ | $0.40 \pm 0.03$ |
| K2D13-500-1-early | $0.12 \pm 0.00$ | $0.41 \pm 0.05$ | $0.12 \pm 0.00$ | $0.12 \pm 0.00$ |
| K2D13-500-5-early | $0.17 \pm 0.01$ | $0.68 \pm 0.04$ | $0.43 \pm 0.04$ | $0.38 \pm 0.01$ |
| K2D13-1000-1-early | $0.13 \pm 0.01$ | $0.58 \pm 0.05$ | $0.12 \pm 0.01$ | $0.12 \pm 0.00$ |
| K2D13-1000-5-early | $0.26 \pm 0.02$ | $0.56 \pm 0.31$ | $0.46 \pm 0.01$ | $0.42 \pm 0.03$ |
| Average | 0.19 | 0.56 | 0.40 | 0.35 |

Table 22: Train targets performances measured with NAUC on `mid` datasets.

| Dataset | AD | IC-DQN | IC-CQL | IC-IQL |
|---|---|---|---|---|
| DR9-20-1-mid | $0.55 \pm 0.07$ | $0.29 \pm 0.10$ | $0.42 \pm 0.17$ | $0.28 \pm 0.13$ |
| DR9-20-5-mid | $0.81 \pm 0.04$ | $0.67 \pm 0.20$ | $0.82 \pm 0.02$ | $0.77 \pm 0.04$ |
| DR9-40-1-mid | $0.51 \pm 0.01$ | $0.27 \pm 0.13$ | $0.62 \pm 0.06$ | $0.58 \pm 0.07$ |
| DR9-40-5-mid | $0.64 \pm 0.07$ | $0.50 \pm 0.10$ | $0.83 \pm 0.01$ | $0.80 \pm 0.02$ |
| DR9-70-1-mid | $0.57 \pm 0.07$ | $0.47 \pm 0.10$ | $0.71 \pm 0.07$ | $0.61 \pm 0.09$ |
| DR9-70-5-mid | $0.70 \pm 0.07$ | $0.77 \pm 0.14$ | $0.83 \pm 0.03$ | $0.86 \pm 0.03$ |
| DR19-75-1-mid | $0.25 \pm 0.04$ | $0.38 \pm 0.07$ | $0.28 \pm 0.04$ | $0.31 \pm 0.08$ |
| DR19-75-5-mid | $0.48 \pm 0.02$ | $0.73 \pm 0.05$ | $0.65 \pm 0.01$ | $0.70 \pm 0.13$ |
| DR19-150-1-mid | $0.26 \pm 0.10$ | $0.49 \pm 0.06$ | $0.38 \pm 0.08$ | $0.36 \pm 0.09$ |
| DR19-150-5-mid | $0.43 \pm 0.07$ | $0.81 \pm 0.01$ | $0.63 \pm 0.03$ | $0.78 \pm 0.03$ |
| DR19-300-1-mid | $0.28 \pm 0.06$ | $0.63 \pm 0.07$ | $0.48 \pm 0.04$ | $0.46 \pm 0.11$ |
| DR19-300-5-mid | $0.45 \pm 0.03$ | $0.85 \pm 0.04$ | $0.65 \pm 0.07$ | $0.73 \pm 0.09$ |
| K2D9-250-1-mid | $0.43 \pm 0.02$ | $0.89 \pm 0.03$ | $0.83 \pm 0.03$ | $0.85 \pm 0.03$ |
| K2D9-250-5-mid | $0.86 \pm 0.05$ | $0.94 \pm 0.03$ | $0.88 \pm 0.02$ | $0.91 \pm 0.01$ |
| K2D9-500-1-mid | $0.62 \pm 0.02$ | $0.93 \pm 0.02$ | $0.89 \pm 0.01$ | $0.88 \pm 0.01$ |
| K2D9-500-5-mid | $0.91 \pm 0.01$ | $0.95 \pm 0.01$ | $0.94 \pm 0.01$ | $0.94 \pm 0.02$ |
| K2D9-1000-1-mid | $0.71 \pm 0.03$ | $0.91 \pm 0.01$ | $0.88 \pm 0.02$ | $0.88 \pm 0.01$ |
| K2D9-1000-5-mid | $0.93 \pm 0.01$ | $0.96 \pm 0.02$ | $0.95 \pm 0.01$ | $0.96 \pm 0.01$ |
| K2D13-250-1-mid | $0.20 \pm 0.03$ | $0.81 \pm 0.04$ | $0.70 \pm 0.05$ | $0.63 \pm 0.09$ |
| K2D13-250-5-mid | $0.73 \pm 0.03$ | $0.85 \pm 0.03$ | $0.87 \pm 0.03$ | $0.85 \pm 0.03$ |
| K2D13-500-1-mid | $0.39 \pm 0.05$ | $0.81 \pm 0.08$ | $0.83 \pm 0.03$ | $0.80 \pm 0.02$ |
| K2D13-500-5-mid | $0.83 \pm 0.02$ | $0.91 \pm 0.03$ | $0.90 \pm 0.02$ | $0.89 \pm 0.03$ |
| K2D13-1000-1-mid | $0.64 \pm 0.01$ | $0.83 \pm 0.06$ | $0.87 \pm 0.02$ | $0.83 \pm 0.04$ |
| K2D13-1000-5-mid | $0.92 \pm 0.02$ | $0.92 \pm 0.02$ | $0.91 \pm 0.01$ | $0.95 \pm 0.01$ |
| Average | 0.59 | 0.73 | 0.74 | 0.73 |

Table 23: Train targets performances measured with NAUC on `late` datasets.

| Dataset | AD | IC-DQN | IC-CQL | IC-IQL |
|---|---|---|---|---|
| DR9-20-1-late | $0.20 \pm 0.16$ | $0.21 \pm 0.07$ | $0.15 \pm 0.06$ | $0.16 \pm 0.09$ |
| DR9-20-5-late | $0.62 \pm 0.10$ | $0.20 \pm 0.06$ | $0.25 \pm 0.07$ | $0.26 \pm 0.05$ |
| DR9-40-1-late | $0.39 \pm 0.12$ | $0.07 \pm 0.04$ | $0.21 \pm 0.02$ | $0.13 \pm 0.09$ |
| DR9-40-5-late | $0.74 \pm 0.01$ | $0.25 \pm 0.07$ | $0.49 \pm 0.14$ | $0.53 \pm 0.05$ |
| DR9-70-1-late | $0.35 \pm 0.14$ | $0.16 \pm 0.05$ | $0.05 \pm 0.07$ | $0.19 \pm 0.06$ |
| DR9-70-5-late | $0.69 \pm 0.05$ | $0.29 \pm 0.14$ | $0.62 \pm 0.06$ | $0.66 \pm 0.02$ |
| DR19-75-1-late | $0.37 \pm 0.07$ | $0.08 \pm 0.01$ | $0.20 \pm 0.05$ | $0.06 \pm 0.04$ |
| DR19-75-5-late | $0.30 \pm 0.05$ | $0.49 \pm 0.08$ | $0.45 \pm 0.04$ | $0.41 \pm 0.11$ |
| DR19-150-1-late | $0.34 \pm 0.06$ | $0.16 \pm 0.08$ | $0.31 \pm 0.08$ | $0.08 \pm 0.05$ |
| DR19-150-5-late | $0.39 \pm 0.11$ | $0.45 \pm 0.06$ | $0.51 \pm 0.04$ | $0.61 \pm 0.04$ |
| DR19-300-1-late | $0.24 \pm 0.03$ | $0.16 \pm 0.03$ | $0.28 \pm 0.09$ | $0.31 \pm 0.05$ |
| DR19-300-5-late | $0.48 \pm 0.08$ | $0.49 \pm 0.03$ | $0.44 \pm 0.04$ | $0.52 \pm 0.06$ |
| K2D9-250-1-late | $0.37 \pm 0.06$ | $0.16 \pm 0.02$ | $0.57 \pm 0.03$ | $0.58 \pm 0.03$ |
| K2D9-250-5-late | $0.74 \pm 0.03$ | $0.78 \pm 0.06$ | $0.86 \pm 0.04$ | $0.86 \pm 0.03$ |
| K2D9-500-1-late | $0.54 \pm 0.02$ | $0.58 \pm 0.02$ | $0.81 \pm 0.01$ | $0.84 \pm 0.02$ |
| K2D9-500-5-late | $0.81 \pm 0.02$ | $0.85 \pm 0.04$ | $0.84 \pm 0.03$ | $0.84 \pm 0.02$ |
| K2D9-1000-1-late | $0.60 \pm 0.02$ | $0.70 \pm 0.03$ | $0.82 \pm 0.02$ | $0.80 \pm 0.02$ |
| K2D9-1000-5-late | $0.83 \pm 0.02$ | $0.86 \pm 0.06$ | $0.87 \pm 0.02$ | $0.82 \pm 0.04$ |
| K2D13-250-1-late | $0.30 \pm 0.03$ | $0.07 \pm 0.00$ | $0.19 \pm 0.06$ | $0.27 \pm 0.07$ |
| K2D13-250-5-late | $0.69 \pm 0.05$ | $0.65 \pm 0.05$ | $0.77 \pm 0.03$ | $0.80 \pm 0.04$ |
| K2D13-500-1-late | $0.46 \pm 0.09$ | $0.40 \pm 0.03$ | $0.59 \pm 0.05$ | $0.58 \pm 0.08$ |
| K2D13-500-5-late | $0.69 \pm 0.10$ | $0.70 \pm 0.06$ | $0.83 \pm 0.02$ | $0.85 \pm 0.06$ |
| K2D13-1000-1-late | $0.55 \pm 0.07$ | $0.51 \pm 0.04$ | $0.84 \pm 0.02$ | $0.86 \pm 0.06$ |
| K2D13-1000-5-late | $0.82 \pm 0.04$ | $0.79 \pm 0.05$ | $0.88 \pm 0.03$ | $0.86 \pm 0.03$ |
| Average | 0.52 | 0.42 | 0.54 | 0.54 |

Table 24: Train targets performances measured with NAUC on complete datasets.

| Dataset | AD | IC-DQN | IC-CQL | IC-IQL |
|---|---|---|---|---|
| DR9-20-1 | $0.68 \pm 0.06$ | $0.23 \pm 0.08$ | $0.60 \pm 0.16$ | $0.60 \pm 0.08$ |
| DR9-20-5 | $0.77 \pm 0.03$ | $0.76 \pm 0.05$ | $0.87 \pm 0.04$ | $0.83 \pm 0.03$ |
| DR9-40-1 | $0.57 \pm 0.05$ | $0.47 \pm 0.07$ | $0.69 \pm 0.10$ | $0.67 \pm 0.08$ |
| DR9-40-5 | $0.66 \pm 0.20$ | $0.51 \pm 0.10$ | $0.84 \pm 0.07$ | $0.89 \pm 0.02$ |
| DR9-70-1 | $0.72 \pm 0.01$ | $0.46 \pm 0.10$ | $0.77 \pm 0.05$ | $0.68 \pm 0.05$ |
| DR9-70-5 | $0.70 \pm 0.07$ | $0.80 \pm 0.11$ | $0.89 \pm 0.05$ | $0.85 \pm 0.03$ |
| DR19-75-1 | $0.19 \pm 0.05$ | $0.35 \pm 0.14$ | $0.44 \pm 0.03$ | $0.32 \pm 0.08$ |
| DR19-75-5 | $0.48 \pm 0.25$ | $0.75 \pm 0.02$ | $0.71 \pm 0.07$ | $0.65 \pm 0.03$ |
| DR19-150-1 | $0.29 \pm 0.02$ | $0.58 \pm 0.08$ | $0.46 \pm 0.10$ | $0.44 \pm 0.20$ |
| DR19-150-5 | $0.62 \pm 0.04$ | $0.83 \pm 0.03$ | $0.72 \pm 0.03$ | $0.82 \pm 0.04$ |
| DR19-300-1 | $0.23 \pm 0.05$ | $0.59 \pm 0.07$ | $0.49 \pm 0.07$ | $0.55 \pm 0.05$ |
| DR19-300-5 | $0.70 \pm 0.03$ | $0.85 \pm 0.06$ | $0.75 \pm 0.04$ | $0.77 \pm 0.08$ |
| K2D9-250-1 | $0.42 \pm 0.03$ | $0.88 \pm 0.05$ | $0.78 \pm 0.06$ | $0.82 \pm 0.04$ |
| K2D9-250-5 | $0.85 \pm 0.03$ | $0.98 \pm 0.01$ | $0.95 \pm 0.01$ | $0.95 \pm 0.01$ |
| K2D9-500-1 | $0.60 \pm 0.05$ | $0.91 \pm 0.03$ | $0.92 \pm 0.01$ | $0.87 \pm 0.03$ |
| K2D9-500-5 | $0.89 \pm 0.02$ | $0.98 \pm 0.01$ | $0.96 \pm 0.00$ | $0.96 \pm 0.00$ |
| K2D9-1000-1 | $0.80 \pm 0.03$ | $0.95 \pm 0.02$ | $0.91 \pm 0.02$ | $0.94 \pm 0.03$ |
| K2D9-1000-5 | $0.92 \pm 0.02$ | $0.96 \pm 0.02$ | $0.96 \pm 0.00$ | $0.96 \pm 0.01$ |
| K2D13-250-1 | $0.22 \pm 0.05$ | $0.62 \pm 0.06$ | $0.57 \pm 0.09$ | $0.69 \pm 0.05$ |
| K2D13-250-5 | $0.81 \pm 0.01$ | $0.83 \pm 0.03$ | $0.87 \pm 0.03$ | $0.90 \pm 0.02$ |
| K2D13-500-1 | $0.39 \pm 0.03$ | $0.83 \pm 0.06$ | $0.87 \pm 0.03$ | $0.88 \pm 0.02$ |
| K2D13-500-5 | $0.89 \pm 0.01$ | $0.94 \pm 0.02$ | $0.94 \pm 0.01$ | $0.94 \pm 0.01$ |
| K2D13-1000-1 | $0.64 \pm 0.07$ | $0.88 \pm 0.03$ | $0.90 \pm 0.03$ | $0.92 \pm 0.01$ |
| K2D13-1000-5 | $0.91 \pm 0.02$ | $0.91 \pm 0.02$ | $0.92 \pm 0.02$ | $0.95 \pm 0.01$ |
| Average | 0.62 | 0.74 | 0.78 | 0.78 |

## J.2 DISCRETE TEST NAUC

Table 25: Test targets performances measured with NAUC on `early` datasets.

| Dataset | AD | IC-DQN | IC-CQL | IC-IQL |
|---|---|---|---|---|
| DR9-20-1-early | $0.18 \pm 0.00$ | $0.08 \pm 0.03$ | $0.16 \pm 0.01$ | $0.14 \pm 0.02$ |
| DR9-20-5-early | $0.17 \pm 0.02$ | $0.21 \pm 0.05$ | $0.23 \pm 0.03$ | $0.22 \pm 0.04$ |
| DR9-40-1-early | $0.18 \pm 0.01$ | $0.19 \pm 0.04$ | $0.20 \pm 0.01$ | $0.21 \pm 0.01$ |
| DR9-40-5-early | $0.21 \pm 0.03$ | $0.32 \pm 0.09$ | $0.40 \pm 0.02$ | $0.38 \pm 0.04$ |
| DR9-70-1-early | $0.19 \pm 0.01$ | $0.20 \pm 0.06$ | $0.24 \pm 0.03$ | $0.24 \pm 0.07$ |
| DR9-70-5-early | $0.22 \pm 0.03$ | $0.50 \pm 0.11$ | $0.50 \pm 0.07$ | $0.46 \pm 0.06$ |
| DR19-75-1-early | $0.12 \pm 0.00$ | $0.08 \pm 0.02$ | $0.10 \pm 0.01$ | $0.06 \pm 0.02$ |
| DR19-75-5-early | $0.12 \pm 0.01$ | $0.21 \pm 0.01$ | $0.17 \pm 0.02$ | $0.11 \pm 0.03$ |
| DR19-150-1-early | $0.12 \pm 0.00$ | $0.14 \pm 0.03$ | $0.14 \pm 0.02$ | $0.11 \pm 0.02$ |
| DR19-150-5-early | $0.12 \pm 0.00$ | $0.38 \pm 0.02$ | $0.23 \pm 0.02$ | $0.14 \pm 0.08$ |
| DR19-300-1-early | $0.12 \pm 0.02$ | $0.18 \pm 0.00$ | $0.15 \pm 0.02$ | $0.05 \pm 0.01$ |
| DR19-300-5-early | $0.12 \pm 0.01$ | $0.56 \pm 0.13$ | $0.34 \pm 0.02$ | $0.27 \pm 0.13$ |
| K2D9-250-1-early | $0.15 \pm 0.00$ | $0.38 \pm 0.08$ | $0.14 \pm 0.00$ | $0.14 \pm 0.00$ |
| K2D9-250-5-early | $0.23 \pm 0.01$ | $0.58 \pm 0.02$ | $0.58 \pm 0.01$ | $0.53 \pm 0.01$ |
| K2D9-500-1-early | $0.17 \pm 0.01$ | $0.61 \pm 0.03$ | $0.42 \pm 0.05$ | $0.24 \pm 0.09$ |
| K2D9-500-5-early | $0.37 \pm 0.03$ | $0.74 \pm 0.06$ | $0.58 \pm 0.01$ | $0.54 \pm 0.04$ |
| K2D9-1000-1-early | $0.22 \pm 0.00$ | $0.67 \pm 0.07$ | $0.63 \pm 0.04$ | $0.57 \pm 0.03$ |
| K2D9-1000-5-early | $0.45 \pm 0.02$ | $0.85 \pm 0.02$ | $0.67 \pm 0.03$ | $0.57 \pm 0.03$ |
| K2D13-250-1-early | $0.12 \pm 0.00$ | $0.08 \pm 0.01$ | $0.11 \pm 0.00$ | $0.11 \pm 0.00$ |
| K2D13-250-5-early | $0.13 \pm 0.00$ | $0.52 \pm 0.02$ | $0.38 \pm 0.02$ | $0.35 \pm 0.02$ |
| K2D13-500-1-early | $0.13 \pm 0.00$ | $0.37 \pm 0.05$ | $0.12 \pm 0.00$ | $0.12 \pm 0.00$ |
| K2D13-500-5-early | $0.16 \pm 0.01$ | $0.56 \pm 0.01$ | $0.42 \pm 0.02$ | $0.36 \pm 0.01$ |
| K2D13-1000-1-early | $0.12 \pm 0.00$ | $0.55 \pm 0.05$ | $0.12 \pm 0.01$ | $0.12 \pm 0.00$ |
| K2D13-1000-5-early | $0.26 \pm 0.02$ | $0.51 \pm 0.28$ | $0.44 \pm 0.01$ | $0.41 \pm 0.02$ |
| Average | 0.18 | 0.39 | 0.31 | 0.27 |

Table 26: Test targets performances measured with NAUC on `mid` datasets.

| Dataset | AD | IC-DQN | IC-CQL | IC-IQL |
|---|---|---|---|---|
| DR9-20-1-mid | $0.27 \pm 0.02$ | $0.07 \pm 0.02$ | $0.11 \pm 0.07$ | $0.09 \pm 0.04$ |
| DR9-20-5-mid | $0.34 \pm 0.02$ | $0.18 \pm 0.05$ | $0.33 \pm 0.00$ | $0.31 \pm 0.03$ |
| DR9-40-1-mid | $0.37 \pm 0.04$ | $0.12 \pm 0.04$ | $0.35 \pm 0.03$ | $0.34 \pm 0.02$ |
| DR9-40-5-mid | $0.40 \pm 0.03$ | $0.25 \pm 0.07$ | $0.49 \pm 0.02$ | $0.51 \pm 0.01$ |
| DR9-70-1-mid | $0.47 \pm 0.03$ | $0.32 \pm 0.08$ | $0.43 \pm 0.03$ | $0.42 \pm 0.02$ |
| DR9-70-5-mid | $0.46 \pm 0.08$ | $0.59 \pm 0.13$ | $0.77 \pm 0.04$ | $0.73 \pm 0.02$ |
| DR19-75-1-mid | $0.17 \pm 0.03$ | $0.14 \pm 0.03$ | $0.15 \pm 0.01$ | $0.09 \pm 0.02$ |
| DR19-75-5-mid | $0.22 \pm 0.02$ | $0.34 \pm 0.04$ | $0.30 \pm 0.01$ | $0.23 \pm 0.07$ |
| DR19-150-1-mid | $0.22 \pm 0.03$ | $0.27 \pm 0.03$ | $0.25 \pm 0.02$ | $0.15 \pm 0.04$ |
| DR19-150-5-mid | $0.28 \pm 0.01$ | $0.57 \pm 0.03$ | $0.45 \pm 0.05$ | $0.42 \pm 0.04$ |
| DR19-300-1-mid | $0.21 \pm 0.03$ | $0.43 \pm 0.09$ | $0.36 \pm 0.03$ | $0.27 \pm 0.07$ |
| DR19-300-5-mid | $0.36 \pm 0.02$ | $0.85 \pm 0.02$ | $0.61 \pm 0.08$ | $0.63 \pm 0.12$ |
| K2D9-250-1-mid | $0.35 \pm 0.03$ | $0.64 \pm 0.06$ | $0.73 \pm 0.04$ | $0.78 \pm 0.05$ |
| K2D9-250-5-mid | $0.68 \pm 0.02$ | $0.64 \pm 0.02$ | $0.80 \pm 0.04$ | $0.82 \pm 0.06$ |
| K2D9-500-1-mid | $0.56 \pm 0.01$ | $0.79 \pm 0.05$ | $0.86 \pm 0.03$ | $0.89 \pm 0.01$ |
| K2D9-500-5-mid | $0.86 \pm 0.01$ | $0.80 \pm 0.04$ | $0.91 \pm 0.02$ | $0.92 \pm 0.02$ |
| K2D9-1000-1-mid | $0.73 \pm 0.03$ | $0.83 \pm 0.02$ | $0.91 \pm 0.01$ | $0.93 \pm 0.01$ |
| K2D9-1000-5-mid | $0.93 \pm 0.00$ | $0.85 \pm 0.03$ | $0.94 \pm 0.01$ | $0.95 \pm 0.00$ |
| K2D13-250-1-mid | $0.18 \pm 0.01$ | $0.50 \pm 0.05$ | $0.56 \pm 0.04$ | $0.56 \pm 0.06$ |
| K2D13-250-5-mid | $0.53 \pm 0.03$ | $0.51 \pm 0.02$ | $0.75 \pm 0.02$ | $0.71 \pm 0.09$ |
| K2D13-500-5-mid | $0.71 \pm 0.03$ | $0.60 \pm 0.05$ | $0.82 \pm 0.02$ | $0.82 \pm 0.04$ |
| K2D13-500-1-mid | $0.35 \pm 0.04$ | $0.67 \pm 0.09$ | $0.78 \pm 0.02$ | $0.78 \pm 0.04$ |
| K2D13-1000-1-mid | $0.59 \pm 0.02$ | $0.80 \pm 0.04$ | $0.82 \pm 0.01$ | $0.80 \pm 0.02$ |
| K2D13-1000-5-mid | $0.87 \pm 0.01$ | $0.75 \pm 0.05$ | $0.91 \pm 0.01$ | $0.91 \pm 0.02$ |
| Average | 0.46 | 0.52 | 0.60 | 0.58 |

Table 27: Test targets performances measured with NAUC on `late` datasets.

| Dataset | AD | IC-DQN | IC-CQL | IC-IQL |
|---|---|---|---|---|
| DR9-20-1-late | $0.15 \pm 0.05$ | $0.12 \pm 0.04$ | $0.11 \pm 0.04$ | $0.09 \pm 0.03$ |
| DR9-20-5-late | $0.34 \pm 0.04$ | $0.10 \pm 0.02$ | $0.10 \pm 0.04$ | $0.10 \pm 0.05$ |
| DR9-40-1-late | $0.33 \pm 0.09$ | $0.12 \pm 0.03$ | $0.16 \pm 0.07$ | $0.14 \pm 0.04$ |
| DR9-40-5-late | $0.50 \pm 0.03$ | $0.22 \pm 0.03$ | $0.34 \pm 0.15$ | $0.31 \pm 0.05$ |
| DR9-70-1-late | $0.35 \pm 0.05$ | $0.21 \pm 0.12$ | $0.25 \pm 0.07$ | $0.21 \pm 0.04$ |
| DR9-70-5-late | $0.63 \pm 0.07$ | $0.21 \pm 0.14$ | $0.58 \pm 0.09$ | $0.49 \pm 0.06$ |
| DR19-75-1-late | $0.23 \pm 0.02$ | $0.06 \pm 0.01$ | $0.11 \pm 0.01$ | $0.04 \pm 0.02$ |
| DR19-75-5-late | $0.21 \pm 0.01$ | $0.19 \pm 0.02$ | $0.21 \pm 0.05$ | $0.19 \pm 0.06$ |
| DR19-150-1-late | $0.26 \pm 0.01$ | $0.13 \pm 0.04$ | $0.19 \pm 0.02$ | $0.07 \pm 0.03$ |
| DR19-150-5-late | $0.39 \pm 0.04$ | $0.30 \pm 0.05$ | $0.38 \pm 0.02$ | $0.40 \pm 0.03$ |
| DR19-300-1-late | $0.30 \pm 0.03$ | $0.19 \pm 0.02$ | $0.23 \pm 0.04$ | $0.20 \pm 0.08$ |
| DR19-300-5-late | $0.45 \pm 0.07$ | $0.40 \pm 0.03$ | $0.43 \pm 0.05$ | $0.42 \pm 0.05$ |
| K2D9-250-1-late | $0.26 \pm 0.02$ | $0.10 \pm 0.01$ | $0.38 \pm 0.01$ | $0.39 \pm 0.02$ |
| K2D9-250-5-late | $0.47 \pm 0.03$ | $0.60 \pm 0.03$ | $0.72 \pm 0.02$ | $0.74 \pm 0.03$ |
| K2D9-500-1-late | $0.39 \pm 0.01$ | $0.48 \pm 0.03$ | $0.72 \pm 0.03$ | $0.75 \pm 0.02$ |
| K2D9-500-5-late | $0.71 \pm 0.01$ | $0.71 \pm 0.02$ | $0.83 \pm 0.01$ | $0.83 \pm 0.01$ |
| K2D9-1000-1-late | $0.48 \pm 0.03$ | $0.55 \pm 0.02$ | $0.80 \pm 0.01$ | $0.80 \pm 0.01$ |
| K2D9-1000-5-late | $0.82 \pm 0.02$ | $0.71 \pm 0.03$ | $0.86 \pm 0.01$ | $0.84 \pm 0.02$ |
| K2D13-250-1-late | $0.20 \pm 0.01$ | $0.08 \pm 0.01$ | $0.22 \pm 0.02$ | $0.25 \pm 0.04$ |
| K2D13-250-5-late | $0.43 \pm 0.01$ | $0.51 \pm 0.02$ | $0.65 \pm 0.01$ | $0.68 \pm 0.02$ |
| K2D13-500-1-late | $0.29 \pm 0.01$ | $0.38 \pm 0.03$ | $0.57 \pm 0.01$ | $0.53 \pm 0.08$ |
| K2D13-500-5-late | $0.58 \pm 0.04$ | $0.62 \pm 0.05$ | $0.79 \pm 0.03$ | $0.81 \pm 0.05$ |
| K2D13-1000-1-late | $0.45 \pm 0.05$ | $0.51 \pm 0.03$ | $0.75 \pm 0.03$ | $0.78 \pm 0.05$ |
| K2D13-1000-5-late | $0.75 \pm 0.02$ | $0.71 \pm 0.03$ | $0.85 \pm 0.04$ | $0.86 \pm 0.03$ |
| Average | 0.42 | 0.34 | 0.47 | 0.45 |

Table 28: Test targets performances measured with NAUC on complete datasets.

| Dataset | AD | IC-DQN | IC-CQL | IC-IQL |
|---|---|---|---|---|
| DR9-20-1 | $0.32 \pm 0.02$ | $0.08 \pm 0.02$ | $0.23 \pm 0.05$ | $0.20 \pm 0.04$ |
| DR9-20-5 | $0.35 \pm 0.02$ | $0.22 \pm 0.06$ | $0.33 \pm 0.03$ | $0.34 \pm 0.03$ |
| DR9-40-1 | $0.43 \pm 0.03$ | $0.20 \pm 0.05$ | $0.36 \pm 0.09$ | $0.35 \pm 0.03$ |
| DR9-40-5 | $0.42 \pm 0.07$ | $0.27 \pm 0.06$ | $0.54 \pm 0.03$ | $0.53 \pm 0.03$ |
| DR9-70-1 | $0.54 \pm 0.07$ | $0.33 \pm 0.09$ | $0.62 \pm 0.09$ | $0.56 \pm 0.13$ |
| DR9-70-5 | $0.68 \pm 0.02$ | $0.72 \pm 0.11$ | $0.78 \pm 0.04$ | $0.79 \pm 0.07$ |
| DR19-75-1 | $0.16 \pm 0.01$ | $0.13 \pm 0.05$ | $0.18 \pm 0.00$ | $0.10 \pm 0.02$ |
| DR19-75-5 | $0.22 \pm 0.11$ | $0.39 \pm 0.03$ | $0.31 \pm 0.00$ | $0.23 \pm 0.06$ |
| DR19-150-1 | $0.23 \pm 0.02$ | $0.31 \pm 0.05$ | $0.26 \pm 0.04$ | $0.22 \pm 0.11$ |
| DR19-150-5 | $0.40 \pm 0.01$ | $0.64 \pm 0.03$ | $0.49 \pm 0.04$ | $0.50 \pm 0.07$ |
| DR19-300-1 | $0.26 \pm 0.01$ | $0.47 \pm 0.03$ | $0.39 \pm 0.04$ | $0.36 \pm 0.06$ |
| DR19-300-5 | $0.63 \pm 0.04$ | $0.82 \pm 0.04$ | $0.66 \pm 0.02$ | $0.73 \pm 0.06$ |
| K2D9-250-1 | $0.33 \pm 0.02$ | $0.65 \pm 0.02$ | $0.65 \pm 0.04$ | $0.67 \pm 0.05$ |
| K2D9-250-5 | $0.68 \pm 0.02$ | $0.69 \pm 0.03$ | $0.83 \pm 0.01$ | $0.86 \pm 0.05$ |
| K2D9-500-1 | $0.50 \pm 0.01$ | $0.85 \pm 0.05$ | $0.87 \pm 0.02$ | $0.81 \pm 0.05$ |
| K2D9-500-5 | $0.87 \pm 0.01$ | $0.85 \pm 0.03$ | $0.93 \pm 0.01$ | $0.94 \pm 0.01$ |
| K2D9-1000-1 | $0.77 \pm 0.02$ | $0.89 \pm 0.03$ | $0.92 \pm 0.01$ | $0.94 \pm 0.01$ |
| K2D9-1000-5 | $0.93 \pm 0.01$ | $0.92 \pm 0.01$ | $0.95 \pm 0.01$ | $0.96 \pm 0.01$ |
| K2D13-250-1 | $0.17 \pm 0.02$ | $0.50 \pm 0.05$ | $0.45 \pm 0.09$ | $0.55 \pm 0.02$ |
| K2D13-250-5 | $0.52 \pm 0.03$ | $0.59 \pm 0.02$ | $0.83 \pm 0.03$ | $0.85 \pm 0.01$ |
| K2D13-500-1 | $0.35 \pm 0.02$ | $0.75 \pm 0.03$ | $0.83 \pm 0.04$ | $0.83 \pm 0.02$ |
| K2D13-500-5 | $0.78 \pm 0.03$ | $0.76 \pm 0.04$ | $0.89 \pm 0.01$ | $0.92 \pm 0.00$ |
| K2D13-1000-1 | $0.60 \pm 0.02$ | $0.86 \pm 0.05$ | $0.86 \pm 0.03$ | $0.87 \pm 0.02$ |
| K2D13-1000-5 | $0.89 \pm 0.01$ | $0.85 \pm 0.01$ | $0.92 \pm 0.01$ | $0.93 \pm 0.00$ |
| Average | 0.50 | 0.57 | 0.63 | 0.63 |

## J.3 DISCRETE TRAIN FINAL SCORES

Table 29: Train targets performances measured with 100th episode score on `early` datasets.

| Dataset | AD | IC-DQN | IC-CQL | IC-IQL |
|---|---|---|---|---|
| DR9-20-1-early | $0.33 \pm 0.08$ | $0.25 \pm 0.04$ | $0.37 \pm 0.11$ | $0.36 \pm 0.04$ |
| DR9-20-5-early | $0.24 \pm 0.05$ | $0.73 \pm 0.11$ | $0.70 \pm 0.00$ | $0.62 \pm 0.06$ |
| DR9-40-1-early | $0.23 \pm 0.06$ | $0.45 \pm 0.11$ | $0.43 \pm 0.03$ | $0.44 \pm 0.09$ |
| DR9-40-5-early | $0.31 \pm 0.02$ | $0.73 \pm 0.15$ | $0.74 \pm 0.05$ | $0.83 \pm 0.04$ |
| DR9-70-1-early | $0.23 \pm 0.06$ | $0.65 \pm 0.19$ | $0.48 \pm 0.05$ | $0.35 \pm 0.16$ |
| DR9-70-5-early | $0.36 \pm 0.08$ | $0.89 \pm 0.08$ | $0.88 \pm 0.07$ | $0.76 \pm 0.03$ |
| DR19-75-1-early | $0.11 \pm 0.02$ | $0.31 \pm 0.10$ | $0.17 \pm 0.05$ | $0.18 \pm 0.07$ |
| DR19-75-5-early | $0.10 \pm 0.03$ | $0.65 \pm 0.05$ | $0.43 \pm 0.03$ | $0.38 \pm 0.10$ |
| DR19-150-1-early | $0.07 \pm 0.05$ | $0.32 \pm 0.06$ | $0.23 \pm 0.06$ | $0.23 \pm 0.09$ |
| DR19-150-5-early | $0.14 \pm 0.05$ | $0.81 \pm 0.03$ | $0.46 \pm 0.04$ | $0.36 \pm 0.22$ |
| DR19-300-1-early | $0.10 \pm 0.03$ | $0.32 \pm 0.10$ | $0.18 \pm 0.04$ | $0.01 \pm 0.02$ |
| DR19-300-5-early | $0.12 \pm 0.05$ | $0.79 \pm 0.26$ | $0.49 \pm 0.10$ | $0.39 \pm 0.23$ |
| K2D9-250-1-early | $0.26 \pm 0.05$ | $0.99 \pm 0.11$ | $0.25 \pm 0.09$ | $0.20 \pm 0.10$ |
| K2D9-250-5-early | $0.42 \pm 0.13$ | $1.57 \pm 0.09$ | $1.37 \pm 0.04$ | $1.17 \pm 0.10$ |
| K2D9-500-1-early | $0.33 \pm 0.12$ | $1.31 \pm 0.08$ | $0.90 \pm 0.13$ | $0.50 \pm 0.13$ |
| K2D9-500-5-early | $0.73 \pm 0.15$ | $1.80 \pm 0.14$ | $1.29 \pm 0.07$ | $1.12 \pm 0.19$ |
| K2D9-1000-1-early | $0.42 \pm 0.13$ | $1.45 \pm 0.22$ | $1.23 \pm 0.14$ | $1.20 \pm 0.04$ |
| K2D9-1000-5-early | $0.96 \pm 0.11$ | $1.83 \pm 0.15$ | $1.30 \pm 0.11$ | $1.10 \pm 0.09$ |
| K2D13-250-1-early | $0.29 \pm 0.09$ | $0.21 \pm 0.09$ | $0.23 \pm 0.02$ | $0.21 \pm 0.09$ |
| K2D13-250-5-early | $0.30 \pm 0.04$ | $1.42 \pm 0.06$ | $0.83 \pm 0.10$ | $0.92 \pm 0.07$ |
| K2D13-500-1-early | $0.21 \pm 0.10$ | $0.81 \pm 0.13$ | $0.20 \pm 0.16$ | $0.18 \pm 0.07$ |
| K2D13-500-5-early | $0.37 \pm 0.07$ | $1.42 \pm 0.04$ | $0.90 \pm 0.11$ | $0.83 \pm 0.08$ |
| K2D13-1000-1-early | $0.18 \pm 0.04$ | $1.20 \pm 0.14$ | $0.26 \pm 0.08$ | $0.25 \pm 0.10$ |
| K2D13-1000-5-early | $0.56 \pm 0.11$ | $1.18 \pm 0.69$ | $0.89 \pm 0.16$ | $0.76 \pm 0.09$ |
| Average | 0.31 | 0.92 | 0.63 | 0.56 |

Table 30: Train targets performances measured with 100th episode score on mid datasets.

| Dataset | AD | IC-DQN | IC-CQL | IC-IQL |
|---|---|---|---|---|
| DR9-20-1-mid | $0.67 \pm 0.08$ | $0.29 \pm 0.10$ | $0.42 \pm 0.20$ | $0.27 \pm 0.13$ |
| DR9-20-5-mid | $0.94 \pm 0.04$ | $0.69 \pm 0.19$ | $0.95 \pm 0.04$ | $0.89 \pm 0.04$ |
| DR9-40-1-mid | $0.61 \pm 0.13$ | $0.27 \pm 0.14$ | $0.79 \pm 0.11$ | $0.68 \pm 0.11$ |
| DR9-40-5-mid | $0.85 \pm 0.04$ | $0.52 \pm 0.09$ | $1.00 \pm 0.00$ | $0.99 \pm 0.02$ |
| DR9-70-1-mid | $0.71 \pm 0.11$ | $0.46 \pm 0.15$ | $0.82 \pm 0.06$ | $0.85 \pm 0.07$ |
| DR9-70-5-mid | $0.89 \pm 0.04$ | $0.81 \pm 0.16$ | $0.95 \pm 0.03$ | $0.98 \pm 0.02$ |
| DR19-75-1-mid | $0.26 \pm 0.04$ | $0.42 \pm 0.11$ | $0.35 \pm 0.06$ | $0.36 \pm 0.10$ |
| DR19-75-5-mid | $0.63 \pm 0.08$ | $0.83 \pm 0.08$ | $0.70 \pm 0.02$ | $0.71 \pm 0.14$ |
| DR19-150-1-mid | $0.29 \pm 0.14$ | $0.52 \pm 0.10$ | $0.42 \pm 0.11$ | $0.39 \pm 0.12$ |
| DR19-150-5-mid | $0.54 \pm 0.06$ | $0.89 \pm 0.02$ | $0.71 \pm 0.03$ | $0.93 \pm 0.02$ |
| DR19-300-1-mid | $0.35 \pm 0.10$ | $0.71 \pm 0.09$ | $0.64 \pm 0.02$ | $0.51 \pm 0.14$ |
| DR19-300-5-mid | $0.62 \pm 0.08$ | $0.98 \pm 0.02$ | $0.73 \pm 0.08$ | $0.81 \pm 0.09$ |
| K2D9-250-1-mid | $0.85 \pm 0.09$ | $1.87 \pm 0.07$ | $1.70 \pm 0.13$ | $1.76 \pm 0.07$ |
| K2D9-250-5-mid | $1.79 \pm 0.09$ | $1.96 \pm 0.04$ | $1.89 \pm 0.07$ | $1.88 \pm 0.04$ |
| K2D9-500-1-mid | $1.27 \pm 0.05$ | $1.96 \pm 0.04$ | $1.88 \pm 0.05$ | $1.77 \pm 0.05$ |
| K2D9-500-5-mid | $1.93 \pm 0.04$ | $2.00 \pm 0.00$ | $1.95 \pm 0.03$ | $1.95 \pm 0.06$ |
| K2D9-1000-1-mid | $1.39 \pm 0.16$ | $1.90 \pm 0.03$ | $1.88 \pm 0.05$ | $1.86 \pm 0.03$ |
| K2D9-1000-5-mid | $1.99 \pm 0.02$ | $1.96 \pm 0.04$ | $2.00 \pm 0.00$ | $1.99 \pm 0.02$ |
| K2D13-250-1-mid | $0.39 \pm 0.15$ | $1.80 \pm 0.04$ | $1.55 \pm 0.11$ | $1.45 \pm 0.23$ |
| K2D13-250-5-mid | $1.45 \pm 0.07$ | $1.85 \pm 0.06$ | $1.94 \pm 0.05$ | $1.90 \pm 0.09$ |
| K2D13-500-5-mid | $1.76 \pm 0.11$ | $1.96 \pm 0.04$ | $1.93 \pm 0.04$ | $1.94 \pm 0.06$ |
| K2D13-500-1-mid | $0.89 \pm 0.17$ | $1.85 \pm 0.13$ | $1.79 \pm 0.07$ | $1.76 \pm 0.17$ |
| K2D13-1000-1-mid | $1.32 \pm 0.06$ | $1.93 \pm 0.08$ | $1.79 \pm 0.12$ | $1.82 \pm 0.05$ |
| K2D13-1000-5-mid | $1.96 \pm 0.04$ | $1.96 \pm 0.02$ | $1.99 \pm 0.02$ | $2.00 \pm 0.00$ |
| Average | 1.01 | 1.27 | 1.28 | 1.27 |

Table 31: Train targets performances measured with 100th episode score on late datasets.

| Dataset | AD | IC-DQN | IC-CQL | IC-IQL |
|---|---|---|---|---|
| DR9-20-1-late | $0.20 \pm 0.15$ | $0.21 \pm 0.08$ | $0.15 \pm 0.06$ | $0.16 \pm 0.09$ |
| DR9-20-5-late | $0.69 \pm 0.14$ | $0.19 \pm 0.05$ | $0.26 \pm 0.06$ | $0.26 \pm 0.05$ |
| DR9-40-1-late | $0.42 \pm 0.15$ | $0.07 \pm 0.04$ | $0.21 \pm 0.02$ | $0.13 \pm 0.09$ |
| DR9-40-5-late | $0.88 \pm 0.02$ | $0.30 \pm 0.14$ | $0.57 \pm 0.20$ | $0.63 \pm 0.05$ |
| DR9-70-1-late | $0.36 \pm 0.17$ | $0.18 \pm 0.05$ | $0.05 \pm 0.06$ | $0.20 \pm 0.07$ |
| DR9-70-5-late | $0.85 \pm 0.12$ | $0.27 \pm 0.13$ | $0.85 \pm 0.09$ | $0.81 \pm 0.03$ |
| DR19-75-1-late | $0.48 \pm 0.10$ | $0.11 \pm 0.02$ | $0.26 \pm 0.10$ | $0.07 \pm 0.05$ |
| DR19-75-5-late | $0.42 \pm 0.13$ | $0.54 \pm 0.09$ | $0.49 \pm 0.07$ | $0.44 \pm 0.12$ |
| DR19-150-1-late | $0.42 \pm 0.12$ | $0.18 \pm 0.09$ | $0.39 \pm 0.08$ | $0.08 \pm 0.05$ |
| DR19-150-5-late | $0.51 \pm 0.15$ | $0.50 \pm 0.04$ | $0.60 \pm 0.04$ | $0.68 \pm 0.04$ |
| DR19-300-1-late | $0.30 \pm 0.06$ | $0.14 \pm 0.03$ | $0.37 \pm 0.15$ | $0.30 \pm 0.06$ |
| DR19-300-5-late | $0.60 \pm 0.13$ | $0.49 \pm 0.05$ | $0.60 \pm 0.07$ | $0.63 \pm 0.13$ |
| K2D9-250-1-late | $0.82 \pm 0.26$ | $0.29 \pm 0.07$ | $1.32 \pm 0.16$ | $1.35 \pm 0.10$ |
| K2D9-250-5-late | $1.65 \pm 0.06$ | $1.77 \pm 0.06$ | $1.86 \pm 0.06$ | $1.85 \pm 0.05$ |
| K2D9-500-1-late | $1.21 \pm 0.11$ | $1.26 \pm 0.05$ | $1.71 \pm 0.06$ | $1.79 \pm 0.07$ |
| K2D9-500-5-late | $1.74 \pm 0.10$ | $1.86 \pm 0.07$ | $1.80 \pm 0.09$ | $1.81 \pm 0.03$ |
| K2D9-1000-1-late | $1.30 \pm 0.11$ | $1.54 \pm 0.05$ | $1.81 \pm 0.03$ | $1.74 \pm 0.07$ |
| K2D9-1000-5-late | $1.80 \pm 0.07$ | $1.87 \pm 0.06$ | $1.89 \pm 0.05$ | $1.80 \pm 0.07$ |
| K2D13-250-1-late | $0.67 \pm 0.17$ | $0.14 \pm 0.05$ | $0.45 \pm 0.22$ | $0.58 \pm 0.23$ |
| K2D13-250-5-late | $1.64 \pm 0.09$ | $1.54 \pm 0.21$ | $1.71 \pm 0.09$ | $1.79 \pm 0.04$ |
| K2D13-500-1-late | $0.95 \pm 0.30$ | $1.06 \pm 0.07$ | $1.40 \pm 0.09$ | $1.39 \pm 0.32$ |
| K2D13-500-5-late | $1.68 \pm 0.14$ | $1.63 \pm 0.10$ | $1.87 \pm 0.06$ | $1.94 \pm 0.05$ |
| K2D13-1000-1-late | $1.33 \pm 0.22$ | $1.25 \pm 0.14$ | $1.90 \pm 0.06$ | $1.88 \pm 0.16$ |
| K2D13-1000-5-late | $1.88 \pm 0.13$ | $1.77 \pm 0.04$ | $1.94 \pm 0.02$ | $1.92 \pm 0.07$ |
| Average | 0.95 | 0.80 | 1.02 | 1.01 |

Table 32: Train targets performances measured with 100th episode score on complete datasets.

| Dataset | AD | IC-DQN | IC-CQL | IC-IQL |
|---|---|---|---|---|
| DR9-20-1 | $0.80 \pm 0.08$ | $0.23 \pm 0.07$ | $0.74 \pm 0.20$ | $0.66 \pm 0.08$ |
| DR9-20-5 | $0.93 \pm 0.06$ | $0.79 \pm 0.05$ | $0.99 \pm 0.02$ | $0.98 \pm 0.03$ |
| DR9-40-1 | $0.74 \pm 0.05$ | $0.51 \pm 0.09$ | $0.82 \pm 0.16$ | $0.83 \pm 0.05$ |
| DR9-40-5 | $0.79 \pm 0.27$ | $0.56 \pm 0.12$ | $0.94 \pm 0.05$ | $1.00 \pm 0.00$ |
| DR9-70-1 | $0.89 \pm 0.07$ | $0.51 \pm 0.15$ | $0.93 \pm 0.05$ | $0.77 \pm 0.04$ |
| DR9-70-5 | $0.86 \pm 0.10$ | $0.88 \pm 0.12$ | $0.99 \pm 0.02$ | $0.99 \pm 0.02$ |
| DR19-75-1 | $0.18 \pm 0.11$ | $0.36 \pm 0.14$ | $0.51 \pm 0.13$ | $0.38 \pm 0.07$ |
| DR19-75-5 | $0.58 \pm 0.32$ | $0.82 \pm 0.04$ | $0.80 \pm 0.11$ | $0.70 \pm 0.08$ |
| DR19-150-1 | $0.37 \pm 0.04$ | $0.71 \pm 0.16$ | $0.62 \pm 0.08$ | $0.50 \pm 0.24$ |
| DR19-150-5 | $0.82 \pm 0.04$ | $0.96 \pm 0.04$ | $0.83 \pm 0.07$ | $0.95 \pm 0.06$ |
| DR19-300-1 | $0.32 \pm 0.11$ | $0.75 \pm 0.06$ | $0.60 \pm 0.07$ | $0.68 \pm 0.08$ |
| DR19-300-5 | $0.88 \pm 0.07$ | $0.95 \pm 0.05$ | $0.94 \pm 0.05$ | $0.92 \pm 0.05$ |
| K2D9-250-1 | $0.85 \pm 0.19$ | $1.82 \pm 0.13$ | $1.63 \pm 0.16$ | $1.75 \pm 0.12$ |
| K2D9-250-5 | $1.86 \pm 0.07$ | $2.00 \pm 0.00$ | $1.98 \pm 0.02$ | $1.98 \pm 0.02$ |
| K2D9-500-1 | $1.39 \pm 0.12$ | $1.95 \pm 0.03$ | $1.93 \pm 0.02$ | $1.88 \pm 0.10$ |
| K2D9-500-5 | $1.94 \pm 0.02$ | $2.00 \pm 0.00$ | $1.99 \pm 0.02$ | $2.00 \pm 0.00$ |
| K2D9-1000-1 | $1.65 \pm 0.10$ | $2.00 \pm 0.00$ | $1.95 \pm 0.06$ | $1.94 \pm 0.08$ |
| K2D9-1000-5 | $1.96 \pm 0.04$ | $1.99 \pm 0.02$ | $2.00 \pm 0.00$ | $2.00 \pm 0.00$ |
| K2D13-250-1 | $0.40 \pm 0.11$ | $1.45 \pm 0.15$ | $1.24 \pm 0.28$ | $1.48 \pm 0.18$ |
| K2D13-250-5 | $1.71 \pm 0.10$ | $1.87 \pm 0.07$ | $1.98 \pm 0.02$ | $1.98 \pm 0.02$ |
| K2D13-500-1 | $0.93 \pm 0.10$ | $1.85 \pm 0.09$ | $1.87 \pm 0.06$ | $1.86 \pm 0.08$ |
| K2D13-500-5 | $1.92 \pm 0.07$ | $1.98 \pm 0.02$ | $2.00 \pm 0.00$ | $2.00 \pm 0.00$ |
| K2D13-1000-1 | $1.32 \pm 0.21$ | $1.93 \pm 0.05$ | $1.89 \pm 0.09$ | $1.94 \pm 0.04$ |
| K2D13-1000-5 | $1.99 \pm 0.02$ | $1.99 \pm 0.02$ | $2.00 \pm 0.00$ | $2.00 \pm 0.00$ |
| Average | 1.09 | 1.29 | 1.34 | 1.34 |

## J.4 DISCRETE TEST FINAL SCORES

Table 33: Test targets performances measured with 100th episode score on `early` datasets.

| Dataset | AD | IC-DQN | IC-CQL | IC-IQL |
|---|---|---|---|---|
| DR9-20-1-early | $0.16 \pm 0.06$ | $0.07 \pm 0.02$ | $0.15 \pm 0.03$ | $0.14 \pm 0.02$ |
| DR9-20-5-early | $0.18 \pm 0.04$ | $0.22 \pm 0.07$ | $0.28 \pm 0.03$ | $0.24 \pm 0.05$ |
| DR9-40-1-early | $0.16 \pm 0.06$ | $0.20 \pm 0.05$ | $0.19 \pm 0.05$ | $0.20 \pm 0.07$ |
| DR9-40-5-early | $0.20 \pm 0.03$ | $0.37 \pm 0.15$ | $0.45 \pm 0.04$ | $0.41 \pm 0.03$ |
| DR9-70-1-early | $0.23 \pm 0.05$ | $0.20 \pm 0.08$ | $0.30 \pm 0.04$ | $0.34 \pm 0.10$ |
| DR9-70-5-early | $0.16 \pm 0.08$ | $0.70 \pm 0.13$ | $0.55 \pm 0.06$ | $0.64 \pm 0.11$ |
| DR19-75-1-early | $0.12 \pm 0.02$ | $0.09 \pm 0.04$ | $0.09 \pm 0.01$ | $0.05 \pm 0.02$ |
| DR19-75-5-early | $0.11 \pm 0.03$ | $0.24 \pm 0.02$ | $0.19 \pm 0.01$ | $0.13 \pm 0.02$ |
| DR19-150-1-early | $0.11 \pm 0.03$ | $0.17 \pm 0.03$ | $0.14 \pm 0.03$ | $0.14 \pm 0.02$ |
| DR19-150-5-early | $0.12 \pm 0.04$ | $0.47 \pm 0.04$ | $0.26 \pm 0.02$ | $0.17 \pm 0.10$ |
| DR19-300-1-early | $0.11 \pm 0.02$ | $0.19 \pm 0.01$ | $0.13 \pm 0.04$ | $0.06 \pm 0.01$ |
| DR19-300-5-early | $0.14 \pm 0.02$ | $0.72 \pm 0.17$ | $0.35 \pm 0.01$ | $0.32 \pm 0.16$ |
| K2D9-250-1-early | $0.30 \pm 0.03$ | $0.77 \pm 0.15$ | $0.30 \pm 0.05$ | $0.24 \pm 0.04$ |
| K2D9-250-5-early | $0.46 \pm 0.07$ | $1.20 \pm 0.05$ | $1.20 \pm 0.01$ | $1.07 \pm 0.03$ |
| K2D9-500-1-early | $0.36 \pm 0.06$ | $1.26 \pm 0.07$ | $0.80 \pm 0.13$ | $0.47 \pm 0.15$ |
| K2D9-500-5-early | $0.78 \pm 0.03$ | $1.56 \pm 0.14$ | $1.16 \pm 0.05$ | $1.12 \pm 0.07$ |
| K2D9-1000-1-early | $0.47 \pm 0.02$ | $1.42 \pm 0.15$ | $1.35 \pm 0.07$ | $1.19 \pm 0.02$ |
| K2D9-1000-5-early | $0.91 \pm 0.11$ | $1.81 \pm 0.04$ | $1.39 \pm 0.08$ | $1.19 \pm 0.06$ |
| K2D13-250-1-early | $0.20 \pm 0.02$ | $0.17 \pm 0.02$ | $0.26 \pm 0.04$ | $0.23 \pm 0.02$ |
| K2D13-250-5-early | $0.26 \pm 0.02$ | $1.06 \pm 0.02$ | $0.81 \pm 0.10$ | $0.71 \pm 0.07$ |
| K2D13-500-1-early | $0.20 \pm 0.03$ | $0.73 \pm 0.07$ | $0.25 \pm 0.03$ | $0.20 \pm 0.02$ |
| K2D13-500-5-early | $0.31 \pm 0.08$ | $1.17 \pm 0.04$ | $0.81 \pm 0.06$ | $0.77 \pm 0.04$ |
| K2D13-1000-1-early | $0.29 \pm 0.06$ | $1.16 \pm 0.12$ | $0.24 \pm 0.04$ | $0.23 \pm 0.05$ |
| K2D13-1000-5-early | $0.53 \pm 0.08$ | $1.09 \pm 0.60$ | $0.86 \pm 0.06$ | $0.82 \pm 0.01$ |
| Average | 0.29 | 0.71 | 0.52 | 0.46 |

Table 34: Test targets performances measured with 100th episode score on `mid` datasets.

| Dataset | AD | IC-DQN | IC-CQL | IC-IQL |
|---|---|---|---|---|
| DR9-20-1-mid | $0.29 \pm 0.06$ | $0.07 \pm 0.02$ | $0.11 \pm 0.07$ | $0.10 \pm 0.04$ |
| DR9-20-5-mid | $0.39 \pm 0.02$ | $0.18 \pm 0.06$ | $0.34 \pm 0.05$ | $0.33 \pm 0.01$ |
| DR9-40-1-mid | $0.43 \pm 0.03$ | $0.14 \pm 0.06$ | $0.43 \pm 0.06$ | $0.38 \pm 0.04$ |
| DR9-40-5-mid | $0.43 \pm 0.03$ | $0.24 \pm 0.07$ | $0.62 \pm 0.03$ | $0.62 \pm 0.04$ |
| DR9-70-1-mid | $0.59 \pm 0.19$ | $0.30 \pm 0.08$ | $0.41 \pm 0.05$ | $0.59 \pm 0.05$ |
| DR9-70-5-mid | $0.52 \pm 0.13$ | $0.70 \pm 0.16$ | $0.91 \pm 0.00$ | $0.89 \pm 0.08$ |
| DR19-75-1-mid | $0.19 \pm 0.05$ | $0.16 \pm 0.03$ | $0.15 \pm 0.01$ | $0.09 \pm 0.03$ |
| DR19-75-5-mid | $0.27 \pm 0.03$ | $0.39 \pm 0.06$ | $0.32 \pm 0.01$ | $0.26 \pm 0.08$ |
| DR19-150-1-mid | $0.24 \pm 0.07$ | $0.29 \pm 0.03$ | $0.28 \pm 0.03$ | $0.17 \pm 0.05$ |
| DR19-150-5-mid | $0.34 \pm 0.02$ | $0.67 \pm 0.01$ | $0.52 \pm 0.06$ | $0.51 \pm 0.04$ |
| DR19-300-1-mid | $0.24 \pm 0.04$ | $0.48 \pm 0.11$ | $0.43 \pm 0.03$ | $0.32 \pm 0.09$ |
| DR19-300-5-mid | $0.46 \pm 0.04$ | $0.95 \pm 0.01$ | $0.70 \pm 0.09$ | $0.75 \pm 0.14$ |
| K2D9-250-1-mid | $0.74 \pm 0.11$ | $1.40 \pm 0.12$ | $1.55 \pm 0.10$ | $1.63 \pm 0.13$ |
| K2D9-250-5-mid | $1.41 \pm 0.07$ | $1.40 \pm 0.03$ | $1.77 \pm 0.07$ | $1.77 \pm 0.13$ |
| K2D9-500-1-mid | $1.11 \pm 0.05$ | $1.78 \pm 0.10$ | $1.84 \pm 0.06$ | $1.88 \pm 0.02$ |
| K2D9-500-5-mid | $1.86 \pm 0.03$ | $1.79 \pm 0.08$ | $1.93 \pm 0.04$ | $1.96 \pm 0.04$ |
| K2D9-1000-1-mid | $1.53 \pm 0.11$ | $1.89 \pm 0.05$ | $1.92 \pm 0.04$ | $1.95 \pm 0.01$ |
| K2D9-1000-5-mid | $1.97 \pm 0.02$ | $1.90 \pm 0.04$ | $2.00 \pm 0.01$ | $2.00 \pm 0.00$ |
| K2D13-250-1-mid | $0.34 \pm 0.07$ | $1.11 \pm 0.10$ | $1.18 \pm 0.14$ | $1.29 \pm 0.15$ |
| K2D13-250-5-mid | $1.12 \pm 0.06$ | $1.11 \pm 0.03$ | $1.75 \pm 0.11$ | $1.65 \pm 0.24$ |
| K2D13-500-5-mid | $1.52 \pm 0.07$ | $1.39 \pm 0.09$ | $1.80 \pm 0.02$ | $1.90 \pm 0.04$ |
| K2D13-500-1-mid | $0.72 \pm 0.10$ | $1.56 \pm 0.20$ | $1.73 \pm 0.03$ | $1.72 \pm 0.09$ |
| K2D13-1000-1-mid | $1.20 \pm 0.06$ | $1.83 \pm 0.07$ | $1.78 \pm 0.04$ | $1.77 \pm 0.03$ |
| K2D13-1000-5-mid | $1.85 \pm 0.01$ | $1.65 \pm 0.10$ | $1.96 \pm 0.02$ | $1.94 \pm 0.03$ |
| Average | 0.82 | 0.97 | 1.10 | 1.10 |

Table 35: Test targets performances measured with 100th episode score on `late` datasets.

| Dataset | AD | IC-DQN | IC-CQL | IC-IQL |
|---|---|---|---|---|
| DR9-20-1-late | $0.15 \pm 0.05$ | $0.13 \pm 0.04$ | $0.11 \pm 0.04$ | $0.09 \pm 0.04$ |
| DR9-20-5-late | $0.40 \pm 0.03$ | $0.09 \pm 0.01$ | $0.08 \pm 0.03$ | $0.10 \pm 0.05$ |
| DR9-40-1-late | $0.35 \pm 0.11$ | $0.12 \pm 0.04$ | $0.16 \pm 0.07$ | $0.14 \pm 0.04$ |
| DR9-40-5-late | $0.61 \pm 0.12$ | $0.24 \pm 0.08$ | $0.41 \pm 0.20$ | $0.37 \pm 0.07$ |
| DR9-70-1-late | $0.36 \pm 0.06$ | $0.20 \pm 0.12$ | $0.25 \pm 0.08$ | $0.20 \pm 0.04$ |
| DR9-70-5-late | $0.75 \pm 0.08$ | $0.27 \pm 0.17$ | $0.73 \pm 0.06$ | $0.64 \pm 0.11$ |
| DR19-75-1-late | $0.27 \pm 0.04$ | $0.06 \pm 0.02$ | $0.13 \pm 0.02$ | $0.04 \pm 0.02$ |
| DR19-75-5-late | $0.26 \pm 0.02$ | $0.20 \pm 0.02$ | $0.23 \pm 0.07$ | $0.20 \pm 0.07$ |
| DR19-150-1-late | $0.33 \pm 0.02$ | $0.13 \pm 0.04$ | $0.21 \pm 0.03$ | $0.06 \pm 0.04$ |
| DR19-150-5-late | $0.47 \pm 0.05$ | $0.32 \pm 0.07$ | $0.45 \pm 0.04$ | $0.44 \pm 0.04$ |
| DR19-300-1-late | $0.36 \pm 0.06$ | $0.18 \pm 0.01$ | $0.30 \pm 0.05$ | $0.22 \pm 0.11$ |
| DR19-300-5-late | $0.57 \pm 0.10$ | $0.43 \pm 0.04$ | $0.55 \pm 0.07$ | $0.52 \pm 0.07$ |
| K2D9-250-1-late | $0.53 \pm 0.10$ | $0.18 \pm 0.01$ | $0.89 \pm 0.02$ | $0.83 \pm 0.04$ |
| K2D9-250-5-late | $0.98 \pm 0.04$ | $1.35 \pm 0.11$ | $1.53 \pm 0.05$ | $1.63 \pm 0.07$ |
| K2D9-500-1-late | $0.85 \pm 0.05$ | $1.18 \pm 0.08$ | $1.59 \pm 0.06$ | $1.64 \pm 0.05$ |
| K2D9-500-5-late | $1.62 \pm 0.02$ | $1.64 \pm 0.05$ | $1.80 \pm 0.07$ | $1.82 \pm 0.03$ |
| K2D9-1000-1-late | $1.05 \pm 0.11$ | $1.35 \pm 0.12$ | $1.77 \pm 0.04$ | $1.73 \pm 0.03$ |
| K2D9-1000-5-late | $1.87 \pm 0.02$ | $1.67 \pm 0.04$ | $1.90 \pm 0.04$ | $1.85 \pm 0.03$ |
| K2D13-250-1-late | $0.39 \pm 0.08$ | $0.18 \pm 0.03$ | $0.44 \pm 0.08$ | $0.51 \pm 0.09$ |
| K2D13-250-5-late | $0.93 \pm 0.05$ | $1.13 \pm 0.06$ | $1.39 \pm 0.03$ | $1.48 \pm 0.08$ |
| K2D13-500-1-late | $0.62 \pm 0.05$ | $0.95 \pm 0.09$ | $1.27 \pm 0.07$ | $1.24 \pm 0.20$ |
| K2D13-500-5-late | $1.33 \pm 0.10$ | $1.42 \pm 0.09$ | $1.80 \pm 0.05$ | $1.82 \pm 0.11$ |
| K2D13-1000-1-late | $0.98 \pm 0.09$ | $1.22 \pm 0.05$ | $1.68 \pm 0.08$ | $1.73 \pm 0.08$ |
| K2D13-1000-5-late | $1.73 \pm 0.04$ | $1.64 \pm 0.04$ | $1.91 \pm 0.06$ | $1.91 \pm 0.06$ |
| Average | 0.74 | 0.68 | 0.90 | 0.88 |

Table 36: Test targets performances measured with 100th episode score on complete datasets.

| Dataset | AD | IC-DQN | IC-CQL | IC-IQL |
|---|---|---|---|---|
| DR9-20-1 | $0.39 \pm 0.04$ | $0.08 \pm 0.02$ | $0.28 \pm 0.09$ | $0.22 \pm 0.06$ |
| DR9-20-5 | $0.41 \pm 0.06$ | $0.24 \pm 0.07$ | $0.40 \pm 0.04$ | $0.38 \pm 0.03$ |
| DR9-40-1 | $0.57 \pm 0.03$ | $0.23 \pm 0.05$ | $0.46 \pm 0.14$ | $0.40 \pm 0.02$ |
| DR9-40-5 | $0.57 \pm 0.13$ | $0.27 \pm 0.09$ | $0.66 \pm 0.05$ | $0.60 \pm 0.04$ |
| DR9-70-1 | $0.68 \pm 0.10$ | $0.34 \pm 0.16$ | $0.75 \pm 0.12$ | $0.61 \pm 0.20$ |
| DR9-70-5 | $0.86 \pm 0.05$ | $0.80 \pm 0.16$ | $0.95 \pm 0.05$ | $0.93 \pm 0.08$ |
| DR19-75-1 | $0.18 \pm 0.02$ | $0.14 \pm 0.06$ | $0.22 \pm 0.01$ | $0.10 \pm 0.01$ |
| DR19-75-5 | $0.26 \pm 0.14$ | $0.44 \pm 0.04$ | $0.36 \pm 0.01$ | $0.27 \pm 0.07$ |
| DR19-150-1 | $0.25 \pm 0.03$ | $0.35 \pm 0.07$ | $0.31 \pm 0.07$ | $0.25 \pm 0.13$ |
| DR19-150-5 | $0.52 \pm 0.04$ | $0.75 \pm 0.03$ | $0.61 \pm 0.07$ | $0.58 \pm 0.07$ |
| DR19-300-1 | $0.30 \pm 0.04$ | $0.55 \pm 0.07$ | $0.43 \pm 0.06$ | $0.45 \pm 0.10$ |
| DR19-300-5 | $0.82 \pm 0.06$ | $0.96 \pm 0.04$ | $0.82 \pm 0.01$ | $0.88 \pm 0.04$ |
| K2D9-250-1 | $0.67 \pm 0.05$ | $1.44 \pm 0.04$ | $1.33 \pm 0.08$ | $1.39 \pm 0.12$ |
| K2D9-250-5 | $1.50 \pm 0.07$ | $1.52 \pm 0.04$ | $1.78 \pm 0.04$ | $1.86 \pm 0.11$ |
| K2D9-500-1 | $1.07 \pm 0.07$ | $1.87 \pm 0.08$ | $1.87 \pm 0.05$ | $1.76 \pm 0.11$ |
| K2D9-500-5 | $1.89 \pm 0.01$ | $1.85 \pm 0.05$ | $1.96 \pm 0.02$ | $1.99 \pm 0.01$ |
| K2D9-1000-1 | $1.64 \pm 0.09$ | $1.97 \pm 0.02$ | $1.97 \pm 0.02$ | $1.98 \pm 0.01$ |
| K2D9-1000-5 | $1.98 \pm 0.01$ | $1.98 \pm 0.00$ | $2.00 \pm 0.01$ | $2.00 \pm 0.00$ |
| K2D13-250-1 | $0.36 \pm 0.02$ | $1.20 \pm 0.18$ | $0.97 \pm 0.23$ | $1.17 \pm 0.08$ |
| K2D13-250-5 | $1.17 \pm 0.09$ | $1.35 \pm 0.09$ | $1.86 \pm 0.07$ | $1.89 \pm 0.01$ |
| K2D13-500-1 | $0.79 \pm 0.08$ | $1.76 \pm 0.01$ | $1.77 \pm 0.08$ | $1.82 \pm 0.05$ |
| K2D13-500-5 | $1.71 \pm 0.06$ | $1.71 \pm 0.07$ | $1.94 \pm 0.02$ | $1.97 \pm 0.02$ |
| K2D13-1000-1 | $1.27 \pm 0.09$ | $1.90 \pm 0.11$ | $1.84 \pm 0.06$ | $1.90 \pm 0.03$ |
| K2D13-1000-5 | $1.92 \pm 0.03$ | $1.88 \pm 0.03$ | $1.99 \pm 0.01$ | $1.99 \pm 0.01$ |
| Average | 0.91 | 1.07 | 1.15 | 1.14 |

## J.5 CONTINUOUS TEST NAUC

Table 37: Test instances performances in continuous environments measured with NAUC on `early` datasets.

| Dataset | AD | IC-TD3 | IC-TD3+BC | IC-IQL |
|---|---|---|---|---|
| HCV-25-1-early | $0.57 \pm 0.03$ | $0.59 \pm 0.03$ | $0.62 \pm 0.03$ | $0.61 \pm 0.17$ |
| HCV-50-1-early | $0.55 \pm 0.01$ | $0.53 \pm 0.10$ | $0.79 \pm 0.02$ | $0.60 \pm 0.25$ |
| HCV-100-1-early | $0.48 \pm 0.05$ | $0.52 \pm 0.21$ | $0.71 \pm 0.06$ | $0.91 \pm 0.02$ |
| ANT-25-1-early | $0.44 \pm 0.02$ | $0.41 \pm 0.01$ | $0.55 \pm 0.01$ | $0.46 \pm 0.09$ |
| ANT-50-1-early | $0.46 \pm 0.01$ | $0.42 \pm 0.02$ | $0.60 \pm 0.02$ | $0.51 \pm 0.12$ |
| ANT-100-1-early | $0.48 \pm 0.02$ | $0.43 \pm 0.05$ | $0.67 \pm 0.02$ | $0.58 \pm 0.14$ |
| HPP-25-1-early | $0.26 \pm 0.23$ | $0.21 \pm 0.00$ | $0.55 \pm 0.09$ | $0.67 \pm 0.17$ |
| HPP-50-1-early | $0.37 \pm 0.27$ | $0.21 \pm 0.01$ | $0.84 \pm 0.11$ | $0.60 \pm 0.26$ |
| HPP-100-1-early | $0.34 \pm 0.18$ | $0.21 \pm 0.00$ | $0.73 \pm 0.42$ | $0.74 \pm 0.13$ |
| WLP-25-1-early | $0.40 \pm 0.15$ | $0.83 \pm 0.00$ | $1.14 \pm 0.03$ | $1.15 \pm 0.03$ |
| WLP-50-1-early | $0.30 \pm 0.17$ | $0.83 \pm 0.01$ | $1.16 \pm 0.01$ | $1.19 \pm 0.01$ |
| WLP-100-1-early | $0.40 \pm 0.21$ | $0.83 \pm 0.00$ | $1.17 \pm 0.02$ | $1.20 \pm 0.01$ |
| Average | 0.42 | 0.50 | 0.80 | 0.77 |

Table 38: Test instances performances in continuous environments measured with NAUC on `mid` datasets.

| Dataset | AD | IC-TD3 | IC-TD3+BC | IC-IQL |
|---|---|---|---|---|
| HCV-25-1-mid | $0.81 \pm 0.01$ | $0.38 \pm 0.11$ | $0.93 \pm 0.03$ | $0.82 \pm 0.04$ |
| HCV-50-1-mid | $0.80 \pm 0.02$ | $0.52 \pm 0.09$ | $0.98 \pm 0.02$ | $0.91 \pm 0.01$ |
| HCV-100-1-mid | $0.86 \pm 0.02$ | $0.45 \pm 0.15$ | $1.02 \pm 0.00$ | $0.97 \pm 0.01$ |
| ANT-25-1-mid | $0.35 \pm 0.01$ | $0.26 \pm 0.01$ | $0.52 \pm 0.04$ | $0.53 \pm 0.02$ |
| ANT-50-1-mid | $0.41 \pm 0.03$ | $0.27 \pm 0.04$ | $0.66 \pm 0.04$ | $0.68 \pm 0.03$ |
| ANT-100-1-mid | $0.47 \pm 0.02$ | $0.24 \pm 0.04$ | $0.84 \pm 0.01$ | $0.83 \pm 0.03$ |
| HPP-25-1-mid | $0.33 \pm 0.08$ | $0.38 \pm 0.30$ | $0.56 \pm 0.12$ | $0.82 \pm 0.09$ |
| HPP-50-1-mid | $0.35 \pm 0.06$ | $0.38 \pm 0.30$ | $0.76 \pm 0.07$ | $0.99 \pm 0.03$ |
| HPP-100-1-mid | $0.37 \pm 0.05$ | $0.71 \pm 0.29$ | $0.65 \pm 0.38$ | $0.94 \pm 0.01$ |
| WLP-25-1-mid | $0.60 \pm 0.35$ | $0.83 \pm 0.00$ | $1.10 \pm 0.01$ | $1.12 \pm 0.01$ |
| WLP-50-1-mid | $0.28 \pm 0.20$ | $0.87 \pm 0.04$ | $1.10 \pm 0.01$ | $1.15 \pm 0.02$ |
| WLP-100-1-mid | $0.39 \pm 0.34$ | $0.84 \pm 0.02$ | $1.07 \pm 0.02$ | $1.09 \pm 0.02$ |
| Average | 0.50 | 0.51 | 0.85 | 0.90 |

Table 39: Test instances performances in continuous environments measured with NAUC on `late` datasets.

| Dataset | AD | IC-TD3 | IC-TD3+BC | IC-IQL |
|---|---|---|---|---|
| HCV-25-1-late | $0.96 \pm 0.02$ | $0.30 \pm 0.01$ | $0.82 \pm 0.08$ | $0.67 \pm 0.23$ |
| HCV-50-1-late | $1.00 \pm 0.02$ | $0.50 \pm 0.15$ | $0.95 \pm 0.02$ | $0.46 \pm 0.17$ |
| HCV-100-1-late | $1.02 \pm 0.00$ | $0.65 \pm 0.08$ | $1.01 \pm 0.02$ | $0.86 \pm 0.12$ |
| ANT-25-1-late | $0.25 \pm 0.03$ | $-0.29 \pm 0.10$ | $0.45 \pm 0.04$ | $0.41 \pm 0.02$ |
| ANT-50-1-late | $0.36 \pm 0.02$ | $-0.20 \pm 0.12$ | $0.56 \pm 0.03$ | $0.58 \pm 0.02$ |
| ANT-100-1-late | $0.48 \pm 0.05$ | $-0.05 \pm 0.11$ | $0.76 \pm 0.03$ | $0.68 \pm 0.02$ |
| HPP-25-1-late | $1.17 \pm 0.09$ | $0.21 \pm 0.30$ | $1.02 \pm 0.04$ | $0.69 \pm 0.34$ |
| HPP-50-1-late | $1.06 \pm 0.05$ | $0.79 \pm 0.05$ | $1.06 \pm 0.04$ | $1.03 \pm 0.22$ |
| HPP-100-1-late | $1.09 \pm 0.04$ | $0.82 \pm 0.02$ | $0.97 \pm 0.03$ | $0.69 \pm 0.38$ |
| WLP-25-1-late | $1.15 \pm 0.01$ | $0.91 \pm 0.05$ | $1.12 \pm 0.01$ | $1.13 \pm 0.02$ |
| WLP-50-1-late | $1.16 \pm 0.01$ | $0.87 \pm 0.07$ | $1.11 \pm 0.02$ | $1.13 \pm 0.01$ |
| WLP-100-1-late | $1.17 \pm 0.01$ | $0.84 \pm 0.02$ | $1.11 \pm 0.01$ | $1.14 \pm 0.01$ |
| Average | 0.91 | 0.45 | 0.91 | 0.79 |

Table 40: Test instances performances in continuous environments measured with NAUC on complete datasets.

| Dataset | AD | IC-TD3 | IC-TD3+BC | IC-IQL |
|---|---|---|---|---|
| HCV-25-1 | $0.76 \pm 0.03$ | $0.50 \pm 0.03$ | $0.90 \pm 0.02$ | $0.73 \pm 0.06$ |
| HCV-50-1 | $0.79 \pm 0.04$ | $0.61 \pm 0.05$ | $0.98 \pm 0.01$ | $0.90 \pm 0.04$ |
| HCV-100-1 | $0.84 \pm 0.03$ | $0.57 \pm 0.04$ | $0.95 \pm 0.03$ | $0.88 \pm 0.10$ |
| ANT-25-1 | $0.40 \pm 0.02$ | $0.28 \pm 0.02$ | $0.77 \pm 0.03$ | $0.76 \pm 0.03$ |
| ANT-50-1 | $0.41 \pm 0.02$ | $0.31 \pm 0.02$ | $0.89 \pm 0.02$ | $0.91 \pm 0.02$ |
| ANT-100-1 | $0.43 \pm 0.02$ | $0.22 \pm 0.09$ | $1.03 \pm 0.03$ | $1.12 \pm 0.02$ |
| HPP-25-1 | $1.14 \pm 0.04$ | $0.52 \pm 0.31$ | $0.96 \pm 0.03$ | $1.03 \pm 0.07$ |
| HPP-50-1 | $0.97 \pm 0.03$ | $0.37 \pm 0.29$ | $1.01 \pm 0.07$ | $0.95 \pm 0.17$ |
| HPP-100-1 | $1.06 \pm 0.05$ | $0.49 \pm 0.29$ | $1.06 \pm 0.03$ | $1.07 \pm 0.04$ |
| WLP-25-1 | $0.62 \pm 0.30$ | $0.92 \pm 0.05$ | $1.09 \pm 0.01$ | $1.13 \pm 0.02$ |
| WLP-50-1 | $1.15 \pm 0.01$ | $0.86 \pm 0.04$ | $1.11 \pm 0.00$ | $1.14 \pm 0.02$ |
| WLP-100-1 | $1.11 \pm 0.02$ | $0.84 \pm 0.03$ | $1.12 \pm 0.01$ | $1.12 \pm 0.01$ |
| Average | 0.81 | 0.54 | 0.99 | 0.98 |

## J.6 CONTINUOUS TEST 0-SHOT

Table 41: Test instances performances of the first rollout episode in continuous environments measured with return on `early` datasets.

| Dataset | AD | IC-TD3 | IC-TD3+BC | IC-IQL |
|---|---|---|---|---|
| HCV-25-1-early | $-220.88 \pm 12.71$ | $-216.98 \pm 11.24$ | $-195.75 \pm 12.36$ | $-203.54 \pm 61.11$ |
| HCV-50-1-early | $-226.05 \pm 2.65$ | $-234.09 \pm 34.27$ | $-145.02 \pm 9.76$ | $-208.96 \pm 87.45$ |
| HCV-100-1-early | $-252.61 \pm 15.14$ | $-236.58 \pm 71.26$ | $-167.89 \pm 16.19$ | $-100.60 \pm 5.49$ |
| ANT-25-1-early | $296.45 \pm 15.06$ | $276.53 \pm 10.25$ | $396.82 \pm 10.75$ | $315.65 \pm 73.67$ |
| ANT-50-1-early | $312.87 \pm 10.10$ | $286.16 \pm 21.89$ | $443.73 \pm 20.54$ | $364.51 \pm 102.41$ |
| ANT-100-1-early | $335.23 \pm 19.18$ | $292.10 \pm 40.28$ | $507.59 \pm 15.86$ | $414.96 \pm 118.52$ |
| HPP-25-1-early | $70.16 \pm 45.37$ | $59.92 \pm 0.92$ | $128.53 \pm 17.04$ | $153.59 \pm 34.73$ |
| HPP-50-1-early | $92.26 \pm 54.04$ | $60.23 \pm 1.39$ | $186.88 \pm 22.43$ | $138.56 \pm 53.00$ |
| HPP-100-1-early | $81.68 \pm 38.08$ | $59.98 \pm 1.00$ | $160.95 \pm 93.44$ | $162.59 \pm 28.93$ |
| WLP-25-1-early | $90.60 \pm 34.63$ | $183.04 \pm 0.29$ | $250.06 \pm 6.48$ | $250.81 \pm 8.46$ |
| WLP-50-1-early | $68.30 \pm 36.22$ | $182.75 \pm 1.06$ | $254.51 \pm 3.10$ | $260.00 \pm 3.45$ |
| WLP-100-1-early | $90.52 \pm 44.10$ | $183.19 \pm 0.21$ | $257.57 \pm 4.77$ | $263.04 \pm 0.63$ |
| Average | 61.54 | 74.69 | 173.16 | 150.88 |

Table 42: Test instances performances of the first rollout episode in continuous environments measured with return on `mid` datasets.

| Dataset | AD | IC-TD3 | IC-TD3+BC | IC-IQL |
|---|---|---|---|---|
| HCV-25-1-mid | $-144.43 \pm 4.02$ | $-281.79 \pm 35.57$ | $-99.10 \pm 9.57$ | $-128.08 \pm 7.88$ |
| HCV-50-1-mid | $-145.12 \pm 6.35$ | $-239.95 \pm 26.18$ | $-82.35 \pm 5.22$ | $-99.88 \pm 4.52$ |
| HCV-100-1-mid | $-127.20 \pm 6.50$ | $-255.47 \pm 53.90$ | $-68.67 \pm 2.03$ | $-80.13 \pm 4.35$ |
| ANT-25-1-mid | $221.32 \pm 17.53$ | $151.01 \pm 9.08$ | $368.69 \pm 47.00$ | $379.55 \pm 22.53$ |
| ANT-50-1-mid | $264.25 \pm 24.07$ | $159.72 \pm 29.75$ | $498.22 \pm 39.80$ | $523.85 \pm 35.23$ |
| ANT-100-1-mid | $327.30 \pm 21.66$ | $129.54 \pm 28.38$ | $641.88 \pm 6.17$ | $631.84 \pm 25.30$ |
| HPP-25-1-mid | $83.21 \pm 16.85$ | $94.06 \pm 60.11$ | $130.49 \pm 24.03$ | $181.91 \pm 19.12$ |
| HPP-50-1-mid | $87.11 \pm 8.47$ | $94.00 \pm 60.09$ | $169.52 \pm 12.86$ | $218.14 \pm 6.05$ |
| HPP-100-1-mid | $89.27 \pm 9.61$ | $160.51 \pm 58.46$ | $142.99 \pm 83.97$ | $207.09 \pm 2.06$ |
| WLP-25-1-mid | $134.39 \pm 77.04$ | $182.78 \pm 0.80$ | $241.65 \pm 2.48$ | $244.75 \pm 3.17$ |
| WLP-50-1-mid | $61.60 \pm 37.31$ | $191.04 \pm 8.61$ | $240.21 \pm 2.30$ | $251.76 \pm 5.87$ |
| WLP-100-1-mid | $88.13 \pm 73.03$ | $185.02 \pm 4.51$ | $237.71 \pm 7.18$ | $240.59 \pm 5.59$ |
| Average | 78.32 | 47.54 | 201.77 | 214.28 |

Table 43: Test instances performances of the first rollout episode in continuous environments measured with return on `late` datasets.

| Dataset | AD | IC-TD3 | IC-TD3+BC | IC-IQL |
|---|---|---|---|---|
| HCV-25-1-late | -90.65 ± 5.90 | -310.54 ± 4.94 | -138.33 ± 21.60 | -187.80 ± 75.81 |
| HCV-50-1-late | -80.11 ± 6.86 | -239.39 ± 51.84 | -91.72 ± 8.79 | -255.96 ± 58.87 |
| HCV-100-1-late | -70.57 ± 1.70 | -189.29 ± 32.35 | -72.90 ± 6.92 | -122.16 ± 41.06 |
| ANT-25-1-late | 134.41 ± 27.34 | -314.67 ± 79.10 | 322.36 ± 40.44 | 276.73 ± 22.98 |
| ANT-50-1-late | 226.56 ± 9.01 | -239.67 ± 103.66 | 417.01 ± 29.61 | 422.26 ± 14.96 |
| ANT-100-1-late | 333.57 ± 44.83 | -110.66 ± 90.69 | 595.82 ± 27.73 | 514.41 ± 20.74 |
| HPP-25-1-late | 257.15 ± 19.85 | 61.20 ± 60.66 | 223.65 ± 6.98 | 160.56 ± 65.08 |
| HPP-50-1-late | 232.62 ± 10.70 | 177.48 ± 10.66 | 231.47 ± 9.88 | 228.31 ± 43.02 |
| HPP-100-1-late | 240.23 ± 7.26 | 183.94 ± 4.57 | 214.53 ± 5.90 | 157.99 ± 77.44 |
| WLP-25-1-late | 252.27 ± 1.76 | 200.43 ± 10.59 | 246.99 ± 4.64 | 247.37 ± 4.98 |
| WLP-50-1-late | 253.73 ± 1.86 | 190.96 ± 14.73 | 245.21 ± 3.92 | 245.41 ± 2.79 |
| WLP-100-1-late | 257.50 ± 2.49 | 185.19 ± 3.90 | 243.79 ± 2.43 | 249.52 ± 2.51 |
| Average | 162.23 | -33.75 | 203.16 | 161.39 |

Table 44: Test instances performances of the first rollout episode in continuous environments measured with return on complete datasets.

| Dataset | AD | IC-TD3 | IC-TD3+BC | IC-IQL |
|---|---|---|---|---|
| HCV-25-1 | -159.11 ± 9.96 | -243.09 ± 8.07 | -100.69 ± 9.79 | -153.71 ± 22.02 |
| HCV-50-1 | -155.25 ± 14.95 | -202.80 ± 17.80 | -78.41 ± 5.83 | -96.91 ± 14.95 |
| HCV-100-1 | -132.12 ± 11.34 | -223.83 ± 10.49 | -90.97 ± 7.47 | -102.68 ± 35.25 |
| ANT-25-1 | 260.28 ± 16.57 | 169.30 ± 16.66 | 580.73 ± 25.74 | 583.55 ± 35.53 |
| ANT-50-1 | 273.01 ± 18.56 | 193.74 ± 16.41 | 686.60 ± 14.75 | 705.95 ± 13.92 |
| ANT-100-1 | 282.22 ± 17.18 | 124.20 ± 68.25 | 804.79 ± 24.51 | 903.03 ± 19.30 |
| HPP-25-1 | 255.59 ± 11.33 | 121.74 ± 62.44 | 211.72 ± 6.57 | 226.34 ± 13.86 |
| HPP-50-1 | 215.10 ± 6.01 | 92.90 ± 58.08 | 223.82 ± 10.52 | 207.17 ± 39.44 |
| HPP-100-1 | 232.62 ± 9.27 | 117.72 ± 59.35 | 231.18 ± 7.09 | 233.35 ± 7.11 |
| WLP-25-1 | 136.92 ± 66.61 | 201.06 ± 10.72 | 236.39 ± 2.67 | 247.13 ± 6.48 |
| WLP-50-1 | 250.49 ± 1.95 | 190.35 ± 8.43 | 241.80 ± 1.75 | 248.35 ± 3.43 |
| WLP-100-1 | 243.97 ± 4.13 | 186.02 ± 5.81 | 244.99 ± 4.45 | 244.25 ± 4.11 |
| Average | 141.98 | 60.61 | 266.00 | 270.49 |

## J.7 CONTINUOUS TEST FINAL SCORES

Table 45: Test instances performances of the 4th rollout episode in continuous environments measured with return on `early` datasets.

| Dataset | AD | IC-TD3 | IC-TD3+BC | IC-IQL |
|---|---|---|---|---|
| HCV-25-1-early | -220.27 ± 10.64 | -216.15 ± 7.26 | -187.76 ± 7.24 | -199.02 ± 62.56 |
| HCV-50-1-early | -222.63 ± 4.79 | -232.30 ± 37.14 | -144.25 ± 6.33 | -208.40 ± 87.96 |
| HCV-100-1-early | -238.62 ± 13.55 | -235.61 ± 72.85 | -169.86 ± 20.42 | -101.70 ± 7.59 |
| ANT-25-1-early | 326.96 ± 21.36 | 276.64 ± 12.45 | 394.83 ± 9.52 | 326.52 ± 77.87 |
| ANT-50-1-early | 330.73 ± 13.11 | 280.53 ± 19.65 | 437.84 ± 21.84 | 357.35 ± 97.74 |
| ANT-100-1-early | 348.13 ± 13.21 | 287.63 ± 44.14 | 497.54 ± 27.44 | 420.80 ± 119.04 |
| HPP-25-1-early | 69.71 ± 44.99 | 59.92 ± 0.91 | 133.83 ± 18.61 | 153.69 ± 34.75 |
| HPP-50-1-early | 91.93 ± 54.10 | 60.21 ± 1.37 | 183.49 ± 25.17 | 138.11 ± 51.90 |
| HPP-100-1-early | 83.96 ± 35.38 | 59.98 ± 0.98 | 158.39 ± 91.98 | 165.28 ± 28.19 |
| WLP-25-1-early | 91.89 ± 35.17 | 183.40 ± 0.57 | 249.73 ± 6.05 | 251.75 ± 6.51 |
| WLP-50-1-early | 69.34 ± 37.06 | 182.85 ± 0.80 | 254.60 ± 2.52 | 259.35 ± 4.71 |
| WLP-100-1-early | 90.62 ± 44.10 | 183.22 ± 0.46 | 256.69 ± 4.30 | 263.17 ± 1.24 |
| Average | 68.48 | 74.19 | 172.09 | 152.24 |

Table 46: Test instances performances of the 4th rollout episode in continuous environments measured with return on `mid` datasets.

| Dataset | AD | IC-TD3 | IC-TD3+BC | IC-IQL |
|---|---|---|---|---|
| HCV-25-1-mid | -134.48 ± 3.56 | -284.53 ± 35.44 | -98.42 ± 10.31 | -126.23 ± 7.78 |
| HCV-50-1-mid | -137.81 ± 6.95 | -234.04 ± 30.58 | -81.65 ± 5.05 | -98.54 ± 4.35 |
| HCV-100-1-mid | -121.71 ± 5.17 | -264.95 ± 49.55 | -68.81 ± 1.03 | -81.75 ± 6.35 |
| ANT-25-1-mid | 253.87 ± 18.76 | 153.26 ± 4.51 | 394.57 ± 26.40 | 403.92 ± 20.90 |
| ANT-50-1-mid | 314.18 ± 31.27 | 156.40 ± 37.07 | 504.20 ± 33.82 | 516.62 ± 17.09 |
| ANT-100-1-mid | 351.32 ± 17.21 | 136.17 ± 33.15 | 662.59 ± 8.88 | 667.48 ± 35.57 |
| HPP-25-1-mid | 76.54 ± 17.71 | 93.75 ± 59.55 | 132.09 ± 24.44 | 181.37 ± 19.54 |
| HPP-50-1-mid | 82.01 ± 8.86 | 94.03 ± 60.12 | 167.93 ± 13.21 | 217.87 ± 5.81 |
| HPP-100-1-mid | 84.13 ± 7.77 | 160.74 ± 58.60 | 142.65 ± 83.98 | 207.48 ± 2.01 |
| WLP-25-1-mid | 134.25 ± 77.26 | 182.51 ± 0.92 | 244.71 ± 2.75 | 244.95 ± 2.84 |
| WLP-50-1-mid | 68.14 ± 52.88 | 190.68 ± 8.14 | 243.05 ± 5.79 | 253.67 ± 4.80 |
| WLP-100-1-mid | 91.81 ± 83.73 | 184.87 ± 4.30 | 235.25 ± 4.51 | 237.75 ± 4.37 |
| Average | 88.52 | 47.41 | 206.51 | 218.72 |

Table 47: Test instances performances of the 4th rollout episode in continuous environments measured with return on `late` datasets.

| Dataset | AD | IC-TD3 | IC-TD3+BC | IC-IQL |
|---|---|---|---|---|
| HCV-25-1-late | -90.90 ± 5.69 | -310.27 ± 5.34 | -133.34 ± 27.29 | -183.96 ± 80.78 |
| HCV-50-1-late | -80.48 ± 7.41 | -238.50 ± 54.31 | -94.12 ± 8.01 | -260.40 ± 55.57 |
| HCV-100-1-late | -72.25 ± 2.57 | -197.66 ± 27.44 | -70.92 ± 6.14 | -118.99 ± 35.02 |
| ANT-25-1-late | 157.69 ± 35.91 | -312.21 ± 82.17 | 314.88 ± 22.20 | 279.86 ± 14.61 |
| ANT-50-1-late | 250.33 ± 30.26 | -242.14 ± 105.57 | 424.98 ± 19.28 | 439.85 ± 25.70 |
| ANT-100-1-late | 386.29 ± 34.52 | -100.55 ± 99.34 | 605.69 ± 13.62 | 508.49 ± 16.07 |
| HPP-25-1-late | 254.75 ± 17.57 | 61.32 ± 61.46 | 218.08 ± 6.55 | 155.96 ± 70.33 |
| HPP-50-1-late | 224.88 ± 8.33 | 177.31 ± 10.46 | 231.93 ± 10.07 | 225.00 ± 47.23 |
| HPP-100-1-late | 226.75 ± 12.30 | 183.90 ± 4.57 | 212.45 ± 3.33 | 158.24 ± 77.56 |
| WLP-25-1-late | 251.62 ± 1.87 | 200.73 ± 10.87 | 249.07 ± 3.11 | 246.40 ± 4.21 |
| WLP-50-1-late | 254.88 ± 2.10 | 189.56 ± 12.18 | 247.18 ± 3.80 | 246.61 ± 2.76 |
| WLP-100-1-late | 258.04 ± 2.87 | 185.39 ± 3.96 | 244.10 ± 2.65 | 249.25 ± 2.77 |
| Average | 168.47 | -33.59 | 204.16 | 162.19 |

Table 48: Test instances performances of the 4th rollout episode in continuous environments measured with return on complete datasets.

| Dataset | AD | IC-TD3 | IC-TD3+BC | IC-IQL |
|---|---|---|---|---|
| HCV-25-1 | -137.99 ± 6.51 | -246.07 ± 10.92 | -97.39 ± 7.06 | -152.26 ± 21.52 |
| HCV-50-1 | -126.71 ± 12.30 | -208.23 ± 11.86 | -80.84 ± 3.98 | -102.25 ± 19.10 |
| HCV-100-1 | -121.41 ± 11.54 | -221.09 ± 16.37 | -90.01 ± 8.74 | -112.84 ± 38.29 |
| ANT-25-1 | 299.17 ± 25.54 | 169.21 ± 18.56 | 604.39 ± 46.33 | 582.13 ± 17.10 |
| ANT-50-1 | 306.79 ± 22.22 | 190.44 ± 13.15 | 695.14 ± 15.47 | 704.26 ± 19.52 |
| ANT-100-1 | 320.41 ± 26.25 | 121.99 ± 76.33 | 812.39 ± 44.33 | 893.94 ± 5.97 |
| HPP-25-1 | 233.29 ± 8.55 | 122.08 ± 62.79 | 210.01 ± 6.47 | 225.81 ± 13.75 |
| HPP-50-1 | 211.35 ± 2.16 | 92.84 ± 57.92 | 222.60 ± 13.03 | 210.21 ± 35.40 |
| HPP-100-1 | 221.10 ± 10.98 | 116.96 ± 58.84 | 229.78 ± 6.97 | 234.06 ± 8.13 |
| WLP-25-1 | 141.21 ± 64.92 | 201.35 ± 10.81 | 235.46 ± 4.94 | 246.65 ± 6.30 |
| WLP-50-1 | 251.47 ± 1.27 | 190.00 ± 8.00 | 242.39 ± 0.43 | 248.17 ± 3.68 |
| WLP-100-1 | 248.44 ± 2.69 | 185.97 ± 5.84 | 242.39 ± 3.76 | 244.71 ± 3.65 |
| Average | 153.93 | 59.62 | 268.86 | 268.55 |

## J.8 JANUS TEST NAUC TABLES

Table 49: Janus NAUC scores for Dynamic 1 test targets when deployed in DR19.

| Dataset | AD | IC-DQN | IC-CQL | IC-IQL |
|---|---|---|---|---|
| Janus19-75-1 | $0.21 \pm 0.01$ | $0.14 \pm 0.03$ | $0.10 \pm 0.06$ | $0.08 \pm 0.04$ |
| Janus19-75-5 | $0.20 \pm 0.02$ | $0.32 \pm 0.07$ | $0.34 \pm 0.03$ | $0.24 \pm 0.03$ |
| Janus19-150-1 | $0.15 \pm 0.01$ | $0.22 \pm 0.04$ | $0.28 \pm 0.04$ | $0.15 \pm 0.06$ |
| Janus19-150-5 | $0.16 \pm 0.08$ | $0.59 \pm 0.01$ | $0.58 \pm 0.04$ | $0.41 \pm 0.17$ |
| Janus19-300-1 | $0.17 \pm 0.02$ | $0.32 \pm 0.03$ | $0.34 \pm 0.11$ | $0.23 \pm 0.03$ |
| Janus19-300-5 | $0.31 \pm 0.15$ | $0.86 \pm 0.06$ | $0.78 \pm 0.05$ | $0.74 \pm 0.07$ |
| Average | 0.20 | 0.41 | 0.40 | 0.31 |

Table 50: Janus NAUC scores for Dynamic 2 test targets when deployed in DR19.

| Dataset | AD | IC-DQN | IC-CQL | IC-IQL |
|---|---|---|---|---|
| Janus19-75-1 | $0.22 \pm 0.01$ | $0.10 \pm 0.02$ | $0.14 \pm 0.03$ | $0.09 \pm 0.03$ |
| Janus19-75-5 | $0.20 \pm 0.02$ | $0.33 \pm 0.05$ | $0.31 \pm 0.02$ | $0.20 \pm 0.04$ |
| Janus19-150-1 | $0.16 \pm 0.01$ | $0.20 \pm 0.06$ | $0.24 \pm 0.05$ | $0.17 \pm 0.02$ |
| Janus19-150-5 | $0.20 \pm 0.01$ | $0.63 \pm 0.02$ | $0.64 \pm 0.03$ | $0.47 \pm 0.03$ |
| Janus19-300-1 | $0.15 \pm 0.01$ | $0.27 \pm 0.01$ | $0.32 \pm 0.05$ | $0.21 \pm 0.04$ |
| Janus19-300-5 | $0.21 \pm 0.20$ | $0.86 \pm 0.05$ | $0.82 \pm 0.02$ | $0.73 \pm 0.05$ |
| Average | 0.19 | 0.40 | 0.41 | 0.31 |

Table 51: Janus NAUC scores for Dynamic 1 test targets when deployed in Janus grid.

| Dataset | AD | IC-DQN | IC-CQL | IC-IQL |
|---|---|---|---|---|
| Janus19-75-1 | $0.09 \pm 0.10$ | $0.09 \pm 0.04$ | $0.11 \pm 0.05$ | $0.12 \pm 0.04$ |
| Janus19-75-5 | $0.08 \pm 0.01$ | $0.30 \pm 0.06$ | $0.27 \pm 0.03$ | $0.17 \pm 0.06$ |
| Janus19-150-1 | $0.08 \pm 0.07$ | $0.23 \pm 0.02$ | $0.24 \pm 0.04$ | $0.17 \pm 0.05$ |
| Janus19-150-5 | $0.08 \pm 0.06$ | $0.42 \pm 0.04$ | $0.45 \pm 0.05$ | $0.44 \pm 0.03$ |
| Janus19-300-1 | $0.13 \pm 0.03$ | $0.25 \pm 0.05$ | $0.28 \pm 0.11$ | $0.20 \pm 0.11$ |
| Janus19-300-5 | $0.18 \pm 0.06$ | $0.77 \pm 0.10$ | $0.63 \pm 0.16$ | $0.63 \pm 0.09$ |
| Average | 0.11 | 0.34 | 0.33 | 0.29 |

Table 52: Janus NAUC scores for Dynamic 2 test targets when deployed in Janus grid.

| Dataset | AD | IC-DQN | IC-CQL | IC-IQL |
|---|---|---|---|---|
| Janus19-75-1 | $0.01 \pm 0.01$ | $0.04 \pm 0.02$ | $0.02 \pm 0.01$ | $0.04 \pm 0.00$ |
| Janus19-75-5 | $0.01 \pm 0.00$ | $0.07 \pm 0.01$ | $0.08 \pm 0.02$ | $0.06 \pm 0.01$ |
| Janus19-150-1 | $0.05 \pm 0.05$ | $0.11 \pm 0.02$ | $0.09 \pm 0.04$ | $0.10 \pm 0.02$ |
| Janus19-150-5 | $0.04 \pm 0.04$ | $0.06 \pm 0.01$ | $0.07 \pm 0.01$ | $0.05 \pm 0.01$ |
| Janus19-300-1 | $0.05 \pm 0.03$ | $0.09 \pm 0.03$ | $0.06 \pm 0.02$ | $0.04 \pm 0.02$ |
| Janus19-300-5 | $0.07 \pm 0.04$ | $0.09 \pm 0.02$ | $0.09 \pm 0.02$ | $0.06 \pm 0.03$ |
| Average | 0.04 | 0.08 | 0.07 | 0.06 |

Table 53: Janus NAUC scores for all targets when deployed in Janus grid.

| Dataset | AD | IC-DQN | IC-CQL | IC-IQL |
|---|---|---|---|---|
| Janus19-75-1 | $0.05 \pm 0.06$ | $0.07 \pm 0.03$ | $0.06 \pm 0.03$ | $0.08 \pm 0.02$ |
| Janus19-75-5 | $0.04 \pm 0.01$ | $0.18 \pm 0.03$ | $0.18 \pm 0.02$ | $0.12 \pm 0.03$ |
| Janus19-150-1 | $0.06 \pm 0.06$ | $0.17 \pm 0.01$ | $0.17 \pm 0.04$ | $0.13 \pm 0.03$ |
| Janus19-150-5 | $0.06 \pm 0.05$ | $0.24 \pm 0.03$ | $0.26 \pm 0.02$ | $0.24 \pm 0.02$ |
| Janus19-300-1 | $0.09 \pm 0.03$ | $0.17 \pm 0.04$ | $0.16 \pm 0.05$ | $0.11 \pm 0.06$ |
| Janus19-300-5 | $0.12 \pm 0.05$ | $0.40 \pm 0.05$ | $0.33 \pm 0.08$ | $0.32 \pm 0.06$ |
| Average | 0.07 | 0.20 | 0.19 | 0.17 |

## K  ACTOR GRADIENT ABLATION

In this section, we ablate our design choice of detaching gradients from the actor networks, following Grigsby et al. (2023). We compare variants with and without backpropagation from the policy head on a subset of continuous-control datasets. Results are reported in Table 54.

Overall, we do not observe any consistent or substantial difference between the two variants. For IC-TD3, the average performance is effectively identical. For IC-TD3+BC, the variant without actor gradients performs slightly better on average, whereas for IC-IQL, the variant with actor gradients has a marginal advantage. Taken together, these results suggest that this design choice is algorithm-dependent and that its impact on performance is small in our setup. We note, however, that more complex continuous tasks might reveal larger differences, which we leave for future work.

Table 54: Ablation of passing actor gradients to the backbone (marked with $\nabla\pi$) on a subset of continuous-control datasets. NAUC results are averaged over 4 random seeds. We highlight higher average performance between two variants for each task with **bold** text.

| Dataset | IC-TD3 | IC-TD3, $\nabla\pi$ | IC-TD3+BC | IC-TD3+BC, $\nabla\pi$ | IC-IQL | IC-IQL, $\nabla\pi$ |
|---|---|---|---|---|---|---|
| HCV-100-1 | $\textbf{0.57} \pm 0.04$ | $0.50 \pm 0.05$ | $0.95 \pm 0.03$ | $\textbf{0.98} \pm 0.02$ | $0.88 \pm 0.10$ | $\textbf{0.91} \pm 0.05$ |
| ANT-100-1 | $0.22 \pm 0.09$ | $\textbf{0.28} \pm 0.05$ | $\textbf{1.03} \pm 0.03$ | $1.02 \pm 0.02$ | $1.12 \pm 0.02$ | $\textbf{1.14} \pm 0.03$ |
| HPP-100-1 | $0.49 \pm 0.29$ | $\textbf{0.50} \pm 0.29$ | $\textbf{1.06} \pm 0.03$ | $0.98 \pm 0.06$ | $1.07 \pm 0.04$ | $\textbf{1.09} \pm 0.09$ |
| WLP-100-1 | $0.84 \pm 0.03$ | $0.84 \pm 0.03$ | $\textbf{1.12} \pm 0.01$ | $1.11 \pm 0.02$ | $1.12 \pm 0.01$ | $1.12 \pm 0.02$ |
| Average | 0.53 | 0.53 | 1.04 | 1.02 | 1.05 | 1.06 |

## L  ALGORITHM DISTILLATION AUGMENTED WITH RETURN-TO-GO

Inspired by the Return-To-Go (RTG) conditioning used in Decision Transformer (Chen et al., 2021) and by the fact that AD can already achieve some performance when trained on near-expert data, we augment our AD implementation with per-episode RTG. During training, we append the RTG to each timestep in the trajectory, and during inference we set the target RTG to the maximal possible value for the corresponding environment. Results on a subset of datasets are reported in Table 55.

For discrete tasks, adding RTG slightly degrades performance. One possible explanation is that DR and K2D require substantial exploration in the initial episodes, while conditioning AD on a non-zero

target return (implicitly associated with successful task completion) may bias the policy away from exploratory behavior and thus hinder learning.

In continuous-control tasks, this augmentation has essentially no effect: the numbers are nearly identical for AD and AD+RTG. A likely reason is that, in these environments, sufficient exploration can already be achieved within a single episode, mitigating the exploration issue that may arise in the discrete case. At the same time, RTG is not particularly informative across tasks with different reward scales, and the model appears to effectively ignore this additional input.

Table 55: Effect of Return-To-Go (RTG) conditioning on Algorithm Distillation (AD). NAUC results are averaged over 4 random seeds.

| Dataset | AD | AD+RTG |
|---|---|---|
| DR9-20-1 | $0.32 \pm 0.02$ | $0.29 \pm 0.01$ |
| DR9-40-1 | $0.43 \pm 0.03$ | $0.41 \pm 0.05$ |
| DR9-70-1 | $0.54 \pm 0.07$ | $0.55 \pm 0.08$ |
| K2D9-1000-1 | $0.77 \pm 0.02$ | $0.75 \pm 0.01$ |
| HCV-100-1 | $0.84 \pm 0.03$ | $0.84 \pm 0.03$ |
| ANT-100-1 | $0.43 \pm 0.02$ | $0.43 \pm 0.02$ |
| HPP-100-1 | $1.06 \pm 0.05$ | $1.06 \pm 0.05$ |
| WLP-100-1 | $1.11 \pm 0.02$ | $1.11 \pm 0.02$ |
| Average | 0.69 | 0.68 |

## M  THE EFFECT OF MODEL SIZE

We also study how the scale of the Transformer backbone affects performance. In all main experiments we use a default configuration with width 512 (hidden dimension) and depth 4 (number of Transformer layers). Here, we fix one of these dimensions to its default value and vary the other:

- width $\in \{64, 128, 256, 512, 1024\}$ with depth fixed to 4,
- depth $\in \{1, 2, 4, 6, 8\}$ with width fixed to 512.

We run these ablations for all our approaches on a subset of datasets. Results are reported Tables 56-67.

For width, we observe the expected pattern that very small models underperform, medium-sized models work best, and very large models start to degrade. For example, the average NAUC of IC-CQL increases from 0.31 at width 64 to 0.39 at width 128 and 0.51 at width 256, and then only slightly improves to 0.52 at width 512 before dropping to 0.43 at width 1024. IC-DQN behaves similarly ($0.27 \rightarrow 0.35 \rightarrow 0.37 \rightarrow 0.40 \rightarrow 0.34$ from width 64 to 1024). For IC-TD3+BC on HCV-100-1, the best performance is achieved at width 256 (average 1.00) compared to 0.77, 0.92, 0.95, and 0.30 for widths 64, 128, 512, and 1024, respectively. Thus, moderate widths (256–512) provide a good trade-off between capacity and stability, while both very small and very large models are suboptimal.

For depth, we find a consistent "sweet spot" around 2–4 layers across methods. Extremely shallow models (depth 1) and very deep models (depth 8) both hurt performance. For instance, the average NAUC of IC-IQL is 0.46 at depth 1, 0.57 at depths 2 and 4, and then decreases to 0.49 and 0.33 at depths 6 and 8. IC-CQL shows the same trend, improving from 0.28 (depth 1) to 0.42 (depth 2) and 0.52–0.54 (depths 4–6), before dropping to 0.45 at depth 8. AD also benefits from moderate depth: its average score grows from 0.39 at depth 1 to 0.51 at depth 2 and 0.57 at depth 4, then slightly decreases to 0.55 and 0.49 at depths 6 and 8. IC-DQN follows a similar pattern, with a peak around depths 2–4 and a drop at depth 8.

One motivation for these experiments was the hypothesis that, on the very small DR9-20-1 and DR9-40-1 datasets, our default backbone might be overparameterized, potentially explaining why AD can outperform RL-based methods there (see Figure 4). The depth ablation for AD is consistent with this hypothesis: on DR9-20-1 and DR9-40-1, reducing depth from 4 to 2 slightly improves performance (from 0.30 to 0.31 and from 0.41 to 0.43, respectively), while making the model either too shallow (depth 1) or deeper (depth 6 or 8) is worse overall. In contrast, IC-DQN does not exhibit

a similar benefit from reducing capacity: its performance on DR9 remains low for all depths, and in the width ablation it actually tends to improve with larger widths (e.g., DR9-40-1 peaks at width 512). Thus, our capacity–data mismatch hypothesis seems to explain AD's behavior on these tiny datasets, but not the weakness of RL-based methods. A plausible explanation is that, in this extreme low-data regime, learning accurate value functions is substantially harder than mimicking actions with a supervised objective, so AD can still extract useful behavior while RL objectives struggle to obtain reliable value estimates regardless of moderate changes in model size.

Overall, these experiments indicate that moderate model scales (width 256–512, depth 2–4) provide a good compromise between performance and compute, and our default configuration (width 512, depth 4) lies in this stable regime. Importantly, the qualitative conclusions of the paper–in particular, the advantages of RL-based ICRL over AD on most datasets and the benefits of offline RL regularization– remain stable across tested widths and depths.

We note that, unlike in large language models, clear "scaling laws" for ICRL have yet to be identified (Kaplan et al., 2020). Our results suggest that, at least in our offline ICRL setting, simply increasing model size does not automatically translate into better performance, and that matching capacity to data and objective remains non-trivial. Recent work has begun to explore scaling behavior in RL more systematically (Fu et al., 2025), but understanding scaling laws for ICRL, remains an open and important direction for future research.

Table 56: Width ablation for AD. Test NAUC as a function of Transformer hidden size. We report mean over 4 random seeds. The depth is fixed to 4 layers.

| Dataset | AD 64 | AD 128 | AD 256 | AD 512 | AD 1024 |
|---|---|---|---|---|---|
| DR9-20-1 | $0.36 \pm 0.02$ | $0.33 \pm 0.02$ | $0.32 \pm 0.02$ | $0.29 \pm 0.01$ | $0.27 \pm 0.01$ |
| DR9-40-1 | $0.46 \pm 0.07$ | $0.43 \pm 0.03$ | $0.42 \pm 0.04$ | $0.41 \pm 0.05$ | $0.35 \pm 0.02$ |
| DR9-70-1 | $0.64 \pm 0.10$ | $0.61 \pm 0.15$ | $0.60 \pm 0.11$ | $0.55 \pm 0.08$ | $0.46 \pm 0.18$ |
| K2D9-1000-1 | $0.68 \pm 0.02$ | $0.81 \pm 0.03$ | $0.82 \pm 0.03$ | $0.75 \pm 0.01$ | $0.68 \pm 0.03$ |
| HCV-100-1 | $0.34 \pm 0.01$ | $0.53 \pm 0.12$ | $0.73 \pm 0.18$ | $0.84 \pm 0.03$ | $0.66 \pm 0.03$ |
| Average | 0.50 | 0.54 | 0.58 | 0.57 | 0.48 |

Table 57: Depth ablation for AD. Test NAUC as a function of the number of Transformer layers. We report mean over 4 random seeds. The hidden size is fixed to 512.

| Dataset | AD 1 | AD 2 | AD 4 | AD 6 | AD 8 |
|---|---|---|---|---|---|
| DR9-20-1 | $0.22 \pm 0.13$ | $0.31 \pm 0.04$ | $0.30 \pm 0.01$ | $0.09 \pm 0.16$ | $0.21 \pm 0.12$ |
| DR9-40-1 | $0.39 \pm 0.02$ | $0.43 \pm 0.03$ | $0.41 \pm 0.05$ | $0.42 \pm 0.05$ | $0.29 \pm 0.17$ |
| DR9-70-1 | $0.52 \pm 0.08$ | $0.62 \pm 0.08$ | $0.55 \pm 0.08$ | $0.56 \pm 0.06$ | $0.50 \pm 0.13$ |
| K2D9-1000-1 | $0.48 \pm 0.01$ | $0.65 \pm 0.04$ | $0.75 \pm 0.01$ | $0.81 \pm 0.02$ | $0.81 \pm 0.01$ |
| HCV-100-1 | $0.35 \pm 0.03$ | $0.54 \pm 0.02$ | $0.84 \pm 0.03$ | $0.88 \pm 0.03$ | $0.67 \pm 0.39$ |
| Average | 0.39 | 0.51 | 0.57 | 0.55 | 0.49 |

Table 58: Width ablation for IC-DQN. Test NAUC as a function of Transformer hidden size. We report mean over 4 random seeds. The depth is fixed to 4 layers.

| Dataset | IC-DQN 64 | IC-DQN 128 | IC-DQN 256 | IC-DQN 512 | IC-DQN 1024 |
|---|---|---|---|---|---|
| DR9-20-1 | $0.08 \pm 0.02$ | $0.10 \pm 0.03$ | $0.09 \pm 0.02$ | $0.09 \pm 0.05$ | $0.10 \pm 0.03$ |
| DR9-40-1 | $0.10 \pm 0.03$ | $0.14 \pm 0.04$ | $0.14 \pm 0.02$ | $0.21 \pm 0.07$ | $0.18 \pm 0.03$ |
| DR9-70-1 | $0.37 \pm 0.04$ | $0.33 \pm 0.04$ | $0.36 \pm 0.05$ | $0.40 \pm 0.03$ | $0.32 \pm 0.09$ |
| K2D9-1000-1 | $0.54 \pm 0.05$ | $0.84 \pm 0.04$ | $0.89 \pm 0.01$ | $0.89 \pm 0.03$ | $0.78 \pm 0.13$ |
| Average | 0.27 | 0.35 | 0.37 | 0.40 | 0.34 |

Table 59: Depth ablation for IC-DQN. Test NAUC as a function of the number of Transformer layers. We report mean over 4 random seeds. The hidden size is fixed to 512.

| Dataset | IC-DQN 1 | IC-DQN 2 | IC-DQN 4 | IC-DQN 6 | IC-DQN 8 |
|---|---|---|---|---|---|
| DR9-20-1 | $0.10 \pm 0.02$ | $0.07 \pm 0.02$ | $0.07 \pm 0.06$ | $0.09 \pm 0.03$ | $0.09 \pm 0.03$ |
| DR9-40-1 | $0.10 \pm 0.02$ | $0.18 \pm 0.06$ | $0.15 \pm 0.10$ | $0.19 \pm 0.02$ | $0.19 \pm 0.09$ |
| DR9-70-1 | $0.12 \pm 0.12$ | $0.18 \pm 0.11$ | $0.12 \pm 0.21$ | $0.00 \pm 0.00$ | $0.31 \pm 0.08$ |
| K2D9-1000-1 | $0.59 \pm 0.02$ | $0.87 \pm 0.02$ | $0.89 \pm 0.03$ | $0.91 \pm 0.02$ | $0.48 \pm 0.45$ |
| Average | 0.23 | 0.33 | 0.31 | 0.30 | 0.27 |

Table 60: Width ablation for IC-CQL. Test NAUC as a function of Transformer hidden size. We report mean over 4 random seeds. The depth is fixed to 4 layers.

| Dataset | IC-CQL 64 | IC-CQL 128 | IC-CQL 256 | IC-CQL 512 | IC-CQL 1024 |
|---|---|---|---|---|---|
| DR9-20-1 | $0.09 \pm 0.05$ | $0.14 \pm 0.03$ | $0.21 \pm 0.04$ | $0.25 \pm 0.03$ | $0.24 \pm 0.04$ |
| DR9-40-1 | $0.09 \pm 0.07$ | $0.23 \pm 0.03$ | $0.34 \pm 0.02$ | $0.33 \pm 0.05$ | $0.32 \pm 0.01$ |
| DR9-70-1 | $0.38 \pm 0.08$ | $0.42 \pm 0.16$ | $0.56 \pm 0.04$ | $0.59 \pm 0.03$ | $0.47 \pm 0.23$ |
| K2D9-1000-1 | $0.68 \pm 0.03$ | $0.79 \pm 0.01$ | $0.92 \pm 0.01$ | $0.92 \pm 0.02$ | $0.70 \pm 0.39$ |
| Average | 0.31 | 0.39 | 0.51 | 0.52 | 0.43 |

Table 61: Depth ablation for IC-CQL. Test NAUC as a function of the number of Transformer layers. We report mean over 4 random seeds. The hidden size is fixed to 512.

| Dataset | IC-CQL 1 | IC-CQL 2 | IC-CQL 4 | IC-CQL 6 | IC-CQL 8 |
|---|---|---|---|---|---|
| DR9-20-1 | $0.15 \pm 0.03$ | $0.24 \pm 0.02$ | $0.25 \pm 0.03$ | $0.27 \pm 0.02$ | $0.21 \pm 0.09$ |
| DR9-40-1 | $0.20 \pm 0.12$ | $0.30 \pm 0.04$ | $0.33 \pm 0.05$ | $0.34 \pm 0.03$ | $0.32 \pm 0.03$ |
| DR9-70-1 | $0.31 \pm 0.21$ | $0.29 \pm 0.29$ | $0.59 \pm 0.03$ | $0.60 \pm 0.06$ | $0.55 \pm 0.11$ |
| K2D9-1000-1 | $0.47 \pm 0.28$ | $0.83 \pm 0.07$ | $0.92 \pm 0.02$ | $0.94 \pm 0.00$ | $0.72 \pm 0.40$ |
| Average | 0.28 | 0.42 | 0.52 | 0.54 | 0.45 |

Table 62: Width ablation for IC-IQL. Test NAUC as a function of Transformer hidden size. We report mean over 4 random seeds. The depth is fixed to 4 layers.

| Dataset | IC-IQL 64 | IC-IQL 128 | IC-IQL 256 | IC-IQL 512 | IC-IQL 1024 |
|---|---|---|---|---|---|
| DR9-20-1 | $0.08 \pm 0.02$ | $0.13 \pm 0.02$ | $0.16 \pm 0.04$ | $0.21 \pm 0.04$ | $0.07 \pm 0.00$ |
| DR9-40-1 | $0.13 \pm 0.04$ | $0.22 \pm 0.05$ | $0.32 \pm 0.04$ | $0.31 \pm 0.04$ | $0.11 \pm 0.04$ |
| DR9-70-1 | $0.46 \pm 0.08$ | $0.43 \pm 0.17$ | $0.63 \pm 0.06$ | $0.53 \pm 0.06$ | $0.37 \pm 0.17$ |
| K2D9-1000-1 | $0.66 \pm 0.04$ | $0.81 \pm 0.03$ | $0.92 \pm 0.01$ | $0.93 \pm 0.01$ | $0.70 \pm 0.39$ |
| HCV-100-1 | $0.71 \pm 0.03$ | $0.83 \pm 0.01$ | $0.93 \pm 0.04$ | $0.88 \pm 0.10$ | $0.31 \pm 0.00$ |
| Average | 0.41 | 0.49 | 0.59 | 0.57 | 0.31 |

Table 63: Depth ablation for IC-IQL. Test NAUC as a function of the number of Transformer layers. We report mean over 4 random seeds. The hidden size is fixed to 512.

| Dataset | IC-IQL 1 | IC-IQL 2 | IC-IQL 4 | IC-IQL 6 | IC-IQL 8 |
|---|---|---|---|---|---|
| DR9-20-1 | $0.13 \pm 0.06$ | $0.19 \pm 0.04$ | $0.21 \pm 0.04$ | $0.20 \pm 0.08$ | $0.07 \pm 0.04$ |
| DR9-40-1 | $0.23 \pm 0.05$ | $0.34 \pm 0.07$ | $0.31 \pm 0.04$ | $0.36 \pm 0.03$ | $0.23 \pm 0.10$ |
| DR9-70-1 | $0.40 \pm 0.05$ | $0.55 \pm 0.11$ | $0.53 \pm 0.06$ | $0.59 \pm 0.10$ | $0.35 \pm 0.26$ |
| K2D9-1000-1 | $0.61 \pm 0.04$ | $0.89 \pm 0.02$ | $0.93 \pm 0.01$ | $0.71 \pm 0.41$ | $0.72 \pm 0.40$ |
| HCV-100-1 | $0.92 \pm 0.02$ | $0.89 \pm 0.02$ | $0.88 \pm 0.10$ | $0.62 \pm 0.31$ | $0.31 \pm 0.00$ |
| Average | 0.46 | 0.57 | 0.57 | 0.49 | 0.33 |

Table 64: Width ablation for IC-TD3. Test NAUC as a function of Transformer hidden size. We report mean over 4 random seeds. The depth is fixed to 4 layers.

| Dataset | IC-TD3 64 | IC-TD3 128 | IC-TD3 256 | IC-TD3 512 | IC-TD3 1024 |
|---|---|---|---|---|---|
| HCV-100-1 | $0.31 \pm 0.17$ | $0.75 \pm 0.15$ | $0.42 \pm 0.22$ | $0.57 \pm 0.04$ | $0.20 \pm 0.06$ |

Table 65: Depth ablation for IC-TD3. Test NAUC as a function of the number of Transformer layers. We report mean over 4 random seeds. The hidden size is fixed to 512.

| Dataset | IC-TD3 1 | IC-TD3 2 | IC-TD3 4 | IC-TD3 6 | IC-TD3 8 |
|---|---|---|---|---|---|
| HCV-100-1 | $0.57 \pm 0.06$ | $0.52 \pm 0.16$ | $0.57 \pm 0.04$ | $0.44 \pm 0.16$ | $0.31 \pm 0.01$ |

Table 66: Width ablation for IC-TD3+BC. Test NAUC as a function of Transformer hidden size. We report mean over 4 random seeds. The depth is fixed to 4 layers.

| Dataset | IC-TD3+BC 64 | IC-TD3+BC 128 | IC-TD3+BC 256 | IC-TD3+BC 512 | IC-TD3+BC 1024 |
|---|---|---|---|---|---|
| HCV-100-1 | $0.77 \pm 0.11$ | $0.92 \pm 0.05$ | $1.00 \pm 0.02$ | $0.95 \pm 0.03$ | $0.30 \pm 0.00$ |

Table 67: Depth ablation for IC-TD3+BC. Test NAUC as a function of the number of Transformer layers. We report mean over 4 random seeds. The hidden size is fixed to 512.

| Dataset | IC-TD3+BC 1 | IC-TD3+BC 2 | IC-TD3+BC 4 | IC-TD3+BC 6 | IC-TD3+BC 8 |
|---|---|---|---|---|---|
| HCV-100-1 | $0.95 \pm 0.04$ | $0.96 \pm 0.03$ | $0.95 \pm 0.03$ | $0.80 \pm 0.24$ | $0.31 \pm 0.00$ |

## N  Environment with Visual States

In this section, we test whether our findings transfer to environments with visual observations. To this end, we consider a visual variant of the Dark Room 9×9 environment based on MiniWorld (Chevalier-Boisvert et al., 2023) from Zisman et al. (2024), with RGB observations of size $64 \times 64 \times 3$. The environment represents a grid embedded in a 3D room where the agent can rotate $90°$ left or right or move one step forward. The objective is to locate an unknown target grid cell. We refer to this environment as MW-DR9.

We use PPO (Schulman et al., 2017) from Stable Baselines (Raffin et al., 2021), augmented with a convolutional encoder (the same encoder is used for all ICRL baselines; see Table 69 for the architecture), to collect training data. To allow PPO to learn in a reasonable amount of time, we set the episode length to 50. Due to computational constraints, we construct datasets with 20, 40, and 70 training targets, each with a single learning history per target. PPO hyperparameters that differ from the Stable Baselines defaults are listed in Table 68. For the ICRL methods, we reuse the same hyperparameter search procedure as for the grid-based DR9 task.

Table 68: PPO hyperparameters for MW-DR-9 data collection.

| Hyperparameter | Value |
|---|---|
| Learning rate | 1e-3 |
| Discount factor ($\gamma$) | 0.9 |
| Number of timesteps | 4000 |
| Number of parallel environments | 1 |

Table 69: Convolutional neural network architecture used to encode visual observations in MW-DR9.

| Layer | Type | Out channels | Kernel size | Stride | Padding |
|---|---|---|---|---|---|
| 1 | Conv2d | 32 | $8 \times 8$ | 4 | 0 |
|   | ReLU | – | – | – | – |
| 2 | Conv2d | 64 | $4 \times 4$ | 2 | 0 |
|   | ReLU | – | – | – | – |
| 3 | Conv2d | 64 | $3 \times 3$ | 1 | 0 |
|   | ReLU | – | – | – | – |

The results are shown in Table 70. We observe trends similar to those in the grid-based DR9 environment. First, IC-CQL and IC-IQL substantially outperform IC-DQN across all three datasets. Second, under extremely limited data coverage (MW-DR9-20-1), AD is competitive and slightly better than the RL-based methods, but as the number of targets increases (MW-DR9-40-1 and MW-DR9-70-1), IC-CQL and IC-IQL clearly dominate AD. These findings support that our main conclusions about the advantages of RL-based ICRL over supervised AD also extend to visual-based environments.

Table 70: Test targets performances measured with NAUC on visual-based datasets.

| Dataset | AD | IC-DQN | IC-CQL | IC-IQL |
|---|---|---|---|---|
| MW-DR9-20-1 | $0.32 \pm 0.08$ | $0.16 \pm 0.08$ | $0.31 \pm 0.01$ | $0.28 \pm 0.02$ |
| MW-DR9-40-1 | $0.29 \pm 0.07$ | $0.25 \pm 0.05$ | $0.35 \pm 0.02$ | $0.32 \pm 0.02$ |
| MW-DR9-70-1 | $0.29 \pm 0.03$ | $0.21 \pm 0.05$ | $0.31 \pm 0.04$ | $0.33 \pm 0.07$ |
| Average | 0.30 | 0.20 | 0.32 | 0.31 |

## O  The Use of Large Language Models (LLMs)

LLMs were used only to improve the writing in final script of our work, i.e. improving flow and fixing grammar of the original text.

