# OpenReview forum: "Yes, Q-learning Helps Offline In-Context RL"
_ICLR.cc/2026/Conference — Submitted to ICLR 2026_

### Official Review · Reviewer_6tnv · 2025-10-22

**Soundness:** 3
**Presentation:** 3
**Contribution:** 2
**Rating:** 4
**Confidence:** 3

**Summary:**

This paper demonstrates that incorporating a reinforcement learning (RL) objective into offline in-context reinforcement learning (ICRL) leads to significant performance improvements. Furthermore, the authors show through extensive experiments that their approach consistently outperforms previous ICRL studies across diverse environments and dataset configurations.

**Strengths:**

* The paper is clearly written and easy to follow.
* It presents experimental results on a wide range of environments, including both discrete and continuous control tasks.
* The proposed method shows robustness across various dataset variations, suggesting strong applicability in practical settings.

**Weaknesses:**

* The paper’s main contribution lies in showing that existing RL algorithms can be effectively adapted within the ICRL framework. While the approach achieves strong empirical results, it mainly builds upon established RL algorithms rather than proposing a new method specifically tailored for ICRL.
* Although the paper evaluates multiple RL algorithms and concludes that RL-based objectives outperform prior approaches, it provides little analysis on why certain algorithms perform better or worse under specific conditions. A discussion on the characteristics and trade-offs of each algorithm would enhance the paper's insight.
* Figure 2 is difficult to interpret. According to the diagram, $a_t$ appears to be derived directly from $c_t$. Still, as I understand, in the discrete case, the action is selected via an argmax over the value function, whereas in the continuous case, it is produced by a policy head. The current figure seems to mix both formulations ambiguously.
* Figure 3 occupies a large space but conveys an unclear message. The upper part of the figure, in particular, shows nearly identical trends across all episodes, which makes it difficult to discern the specific insight or conclusion the figure intends to highlight.
* The sentence at line 165 is duplicated.

**Questions:**

* Is there a specific reason for using the NAUC metric? The reported results seem consistent with other metrics, so presenting multiple metrics in a table might be more informative.
* Why does CQL achieve the best test performance in Figure 3? Since this work focuses on ICRL, it would be helpful to explain why models perform similarly during training but diverge at test time.
* Similarly, can the authors provide an explanation for the following patterns: why AD performs best at DR9 with 20 and 40 in Figure 4, why IC-DQN performs uniquely well at DR19, and why AD outperforms IC-DQN at K2D13 in Figure 6? Clarifying these results would strengthen the empirical analysis.

---

> ### Author Response · Authors · 2025-11-18
> **Addressing weaknesses**
>
> We thank the reviewer for their time and constructive feedback.
>
> ---
>
> ### W1. Limited contribution / mainly adapting existing RL algorithms
>
> Our primary goal is to establish a solid empirical foundation for offline ICRL with explicit RL objectives, rather than to immediately propose a heavily engineered new algorithm. We see this as an important prerequisite: if a new, more complex ICRL method underperforms, it is otherwise unclear whether the issue lies in the method itself or in the suitability of RL objectives in this setting.
>
> Concretely, our contribution is threefold:
>
> * We adapt standard offline RL algorithms to the offline ICRL setting using a unified Transformer-based backbone.
>
> * We systematically evaluate these methods on more than 150 datasets across multiple axes: coverage, expertise, learning-history structure, discrete vs. continuous control, and mixture-of-dynamics / OOD dynamics.
>
> * We show that explicit RL optimization is consistently beneficial for offline ICRL and that conservative / regularized offline RL variants are often crucial.
>
> To our knowledge, this is the first work to explicitly optimize RL objectives in an offline ICRL setting with a scalable Transformer architecture, and to do so across such a broad range of environments and dataset regimes.
>
> ---
>
> ### W2. Limited analysis of why specific RL algorithms perform differently
>
> We agree that a deeper qualitative analysis of algorithm behavior would further strengthen the paper. Our current analysis already distinguishes several regimes:
>
> * In overall performance (Figure 3, lines 260-278), we observe that RL methods are similar on train targets but diverge on test targets, with CQL performing best, followed by IQL, and DQN weakest on average.
>
> * In expertise experiments (Figure 5, lines 324-361), DQN tends to do best on low-quality (“early”) datasets, while CQL and IQL dominate on mid/late/complete datasets. This aligns with the intuition that strong pessimistic regularization can hurt when data is very suboptimal but is beneficial when high-quality trajectories are present.
>
> Our interpretation is:
>
> * Offline-regularized methods handle OOD state-action pairs better and thus generalize better to unseen tasks in test-time ICRL, especially in complex or continuous settings.
>
> * Less-regularized methods (DQN, TD3) can perform well when datasets are small/simple or very suboptimal (where strong pessimism over-constrains learning).
>
> ---
>
> ### W3. Figure 2 is ambiguous about how actions are produced
>
> Thank you for pointing this out. The intention of Figure 2 is to provide a unified schematic of the architecture, not to claim that all variants share the same action-selection mechanism.
>
> More precisely (as already stated in Section 3.1):
>
> * For discrete environments, the backbone outputs Q-values, and actions are chosen via argmax over Q during inference.
>
> * For continuous environments, we additionally attach a policy head, and actions are produced by this policy head.
>
> The orange action is not derivation of action but concatenation of $c_t$ and $a_t$ as it usually done for Q function input if needed (in continuous case). We will update figure caption with this remark.
>
> We hope this resolves the ambiguity.
>
> ---
>
> ### W4. Figure 3 is large but its message is unclear
>
> The intention of Figure 3 is twofold:
>
> Top row (Episode returns vs. NAUC):
>
> * Show that our proposed NAUC metric tracks episode-wise performance closely, so that the ranking of algorithms is stable across metrics.
>
> * Contrast train vs. test performance: on train targets, RL-based methods look similar, while on test targets their differences become more pronounced.
>
> Bottom row (rliable performance profiles of NAUC):
>
> Provide a statistically robust comparison using rliable metrics (https://arxiv.org/abs/2108.13264) across all datasets, confirming that CQL > IQL > DQN > AD on average, and that the RL methods significantly dominate AD.
>
> ---
>
> ### W5. Duplicated sentence at line 165
>
> Thank you for noticing, we will remove the duplicate.

---

> ### Author Response · Authors · 2025-11-18
> **Addressing questions**
>
> ### Q1. NAUC metric
>
> We chose NAUC as our main scalar metric because it:
>
> * Aggregates performance across episodes into a single number, capturing both adaptation speed and final performance.
>
> * Is more robust to noise than using a single fixed episode, which can be misleading if a policy is unstable or if rare failures occur late in the rollout.
>
> * Converts standard “learning curves” into a reusable numeric metric, making it easier to compare methods across many datasets and to reuse our results in future studies.
>
> As suggested, we do not rely solely on NAUC: Appendix I and Appendix J provide final return tables and rliable statistics (median, IQM, mean) for both NAUC and last-episode scores across all environments. We also provide 0-shot performance (only the first episode performance) where suitable.
>
> Figure 3 explicitly shows that NAUC and fixed-episode returns yield consistent rankings.
>
> ---
>
> ### Q2. Why does CQL achieve the best test performance in Figure 3?
>
> As we mentioned before, our goal was not to find the best approach for offline ICRL but to demonstrate the usefulness of RL and offline RL components in this setup. Because of that, we did not focus on providing a detailed analysis of why one algorithm performs better than another. In practice, it is often difficult to give a precise explanation for why a particular RL algorithm outperforms others, and we typically rely on empirical evidence. Our best guess here is the following: CQL performs better than DQN because offline regularization is needed to handle OOD state–action pairs, and its slightly better performance compared to IQL is consistent with observations in the offline RL literature (https://arxiv.org/abs/2210.07105, https://arxiv.org/abs/2504.11453), where methods based on simple BC-style regularization such as ReBRAC (https://arxiv.org/abs/2305.09836) are among the strongest baselines. In our case, IC-CQL can effectively be seen as an IC-DQN+BC variant, which may explain its advantage.
>
> ---
>
> ### Q3. Explaining specific patterns in Figures 4 and 6
>
> We appreciate the reviewer’s request for more detailed interpretations. While providing definitive causal explanations is difficult (as is often the case in RL), we can offer plausible interpretations grounded in our dataset statistics and experiments:
>
> **Why does AD perform best on DR9 with 20 and 40 targets (Figure 4)?**
>
> In these settings, datasets are very small and the model capacity is fixed (the same Transformer size across all tasks; see Appendix H, Table 16). Our hypothesis is that the model is overparameterized relative to the data, which can hurt value-based RL training more than supervised next-action prediction. Poorly estimated value functions may lead to unstable TD updates and degrade performance. AD, which uses a purely supervised objective, can behave more like a strong behavior cloning / imitation learner in this extreme low-data regime, explaining its slight advantage.
>
> To validate this, we are running additional experiments with smaller backbones, and will report results as soon as possible and will let you know about it.
>
> **Why does IC-DQN perform uniquely well on DR19 but AD outperforms IC-DQN on K2D13 (Figure 6)?**
>
> DR19 is larger than DR9 but still a relatively simple discrete environment with good target coverage (21%, 42% and 83% of possible targets compared to 3.5% for the largest dataset of K2D13). In this setting, the combination of high coverage and simple dynamics reduces the need for strong pessimism. A non-regularized method like DQN can fully exploit the data without being overly conservative.
>
> K2D13 is substantially more complex and less covered than DR19. In this harder POMDP-like environment, TD learning may suffer more from partial observability and value overestimation, making plain DQN brittle. Small differences in value estimates can have large downstream effects on policy quality. AD’s supervised objective might be more stable here.
>
> There remain interesting edge cases (very small datasets or very simple environments) where supervised AD or unregularized DQN can temporarily close the gap or even slightly outperform others.

---

> ### Author Response · Authors · 2025-11-22
> **Scale experiments**
>
> Dear reviewer,
>
> Based on our hypothesis on the model overparametrization, we provide the ablation on the model scale in Appendix M of the updated PDF. Other novel experiments might be also be interesting for you, please have a look.

---

> > ### Comment · Reviewer_6tnv · 2025-11-28
> >
> > I appreciate your detailed response.
> > However, even after reviewing your feedback, I still maintain my concerns regarding the contribution of the paper.
> >
> > While I understand the contribution mentioned in W1, I feel that the current presentation focuses somewhat heavily on listing performance results across various datasets.
> > In my view, for a paper with this structure to be accepted, the experimental results must be consistent and accompanied by sufficiently thorough explanations.
> > I believe it would be beneficial for the authors to further substantiate their claims by grounding them in the specific characteristics of both offline RL and ICRL.

---

> > > ### Author Response · Authors · 2025-11-28
> > >
> > > We thank reviewer for the discussion. We would like to emphasize how our paper goes beyond merely applying existing RL algorithms by conducting a large-scale empirical study that yields new insights and understanding in the offline ICRL setting. We summarize our main contributions and findings below:
> > >
> > > * **First large-scale study with offline RL objectives in ICRL:** To our knowledge, ours is the first paper to systematically integrate explicit offline RL objectives into the ICRL framework using a Transformer backbone. We evaluate across over 150 tasks and datasets, covering a broad spectrum of environments and data regimes. This scale of study goes far beyond prior work and allows us to draw robust conclusions about ICRL behavior.
> > >
> > > * **New empirical phenomena and observations:** Our work uncovers a range of novel phenomena that have not been reported before. For example, we find systematic effects related to the coverage of the offline data (i.e., how much of the state-action space is represented), the expertise of trajectories (degree of optimality), the structure of the learning history provided in context, and out-of-distribution dynamics (shifts between train and test). These observations, collected across diverse datasets, reveal when and how ICRL succeeds or fails under different conditions. By cataloging these effects, we provide the community with a clearer picture of the factors that influence ICRL performance.
> > >
> > > * **Actionable insights on RL objectives:** Importantly, our experiments do more than report numbers: they yield concrete guidance. We identify specific regimes where incorporating offline RL objectives is most beneficial. In particular, we show that conservative or regularized RL variants (e.g., TD3+BC, IQL) become critical when the offline data has limited coverage or high variability. In contrast, when data is dense and high-quality, simpler methods like AD might be enough. These insights help explain why certain algorithms work better in different settings and can directly inform practitioners which methods to use under which data conditions.
> > >
> > > * **Understanding realistic constraints:** Our study explicitly examines ICRL under realistic, constrained settings that reflect practical scenarios. For example, we simulate low-coverage and unstructured datasets that one might encounter in real applications. We show how ICRL performance degrades under these constraints and how choosing the right objective can mitigate failures. By highlighting these challenges, our work improves the understanding of ICRL’s limitations and robustness in practice.
> > >
> > > * **Finding on regularization in continuous control:** One of our key discoveries is that regularization is crucial in continuous-control tasks. We observe that a vanilla online RL algorithm (TD3) performs worse than even our AD baseline when used in the in-context setting. However, when we apply offline-style regularization (e.g. TD3+BC, IQL), performance dramatically improves, significantly outperforming both pure online TD3 and AD. This novel result highlights that offline-inspired regularization is not just helpful but essential in continuous-control ICRL scenarios.
> > >
> > > * **Guiding future research:** While our methods are adapted from existing RL algorithms, the empirical findings themselves are novel and deeply informative. We believe these findings form a foundation for future work. By systematically mapping out where different algorithms succeed or fail in ICRL, we provide a roadmap that researchers can use when developing new methods. In other words, the value of our paper lies not in inventing a new algorithm, but in generating and interpreting extensive empirical evidence. This evidence is essential for understanding in-context RL and cannot be gleaned from theoretical arguments alone.
> > >
> > > We hope these points clarify the unique contributions of our work. We emphasize that our paper is much more than a benchmark: it is a comprehensive study that discovers new behavior of ICRL systems and distills lessons for the field.

---

### Official Review · Reviewer_jKPw · 2025-10-25

**Soundness:** 4
**Presentation:** 3
**Contribution:** 3
**Rating:** 8
**Confidence:** 4

**Summary:**

This paper explores **integrating RL objectives to offline ICRL** framework, namely Algorithm Distillation (AD) [1], and investigate whether it can achieve significantly better results in offline ICRL in various axes (coverage, expertise, learning history, continuous state/action).
Experimental results show that additionally optimizing RL objectives improves performance compared to standard AD.
Specifically, combining AD with RL objectives enhances data efficiency in both discrete and continuous domains, and robustness to various axes (e.g. coverage and expertise of the dataset, number of target repetition)

[1] Michael Laskin et al., “In-context Reinforcement Learning with Algorithm Distillation”, ICLR 2023

**Strengths:**

**S1. Clear organization and presentation**
* The paper is well structured, allowing the reader to easily follow the main argument and its supporting evidence.
* Experimental results are presented clearly, and the setup is well designed to validate the central claim that “RL objectives improve AD.”

**S2. Useful insights in offline ICRL**
* Through its experimental findings, the paper provides valuable insights into the role of offline RL objectives in ICRL.
* For example, it highlights the benefits of using offline rather than online RL objectives, and demonstrates improved data efficiency.

**Weaknesses:**

**W1. (Slightly) limited novelty**
* The methodological novelty of the work is somewhat limited, as prior studies have explored similarly applying RL objectives in online ICRL settings, namely AMAGO-2 [2], ReLIC [3] (note that AMAGO-2 also employs Transformers in a similar context).
* That said, I view this as a moderate rather than critical weakness; offline RL setting poses unique challenges, the paper’s analytical perspective and insights remain valuable even if the methodological novelty is limited.

**W2. Scalability claims**
* While the authors claims “this is the first work to apply RL objectives in an offline ICRL setting with a **scalable Transformer architecture**” for the use of a Transformer architecture, the experiments are limited to small-scale datasets.
* Including experiments on larger-scale datasets (e.g., the full xland-minigrid dataset) or with larger Transformer models would provide stronger evidence for the claimed scalability.

[2] Jake Grigsby et al., “AMAGO-2: Breaking the Multi-Task Barrier in Meta-Reinforcement Learning with Transformers”, NeurIPS 2024
[3] Ahmad Elawady et al., "ReLIC: A Recipe for 64k Steps of In-Context Reinforcement Learning for Embodied AI", 2025.

**Questions:**

**Q1. Large-scale experiments.**
* Have the observed improvements been evaluated on large-scale datasets (e.g., the full xland-minigrid dataset) or with higher-capacity Transformer architectures (i.e., models with more parameters)?
* Such experiments could provide deeper insights into the role and importance of the Transformer architecture.

**Q2. Pixel-based environments**
* Do the proposed methods and conclusions generalize to pixel-based environments (e.g., DMLab Watermaze from the AD paper)?
* While not strictly necessary, demonstrating this would substantially strengthen the generality and robustness of the claims.

---

> ### Author Response · Authors · 2025-11-18
>
> We thank the reviewer for their positive and detailed feedback.
>
> # Addressing weaknesses
>
> ### W1. (Slightly) limited novelty
>
> We agree that prior work has explored applying RL objectives in (online) ICRL settings, e.g., AMAGO-2 and ReLIC, which both use Transformers and large-scale online RL training for in-context adaptation.  Our claim is not to be the first to combine RL and ICRL in general, but rather to be, to the best of our knowledge, the first to systematically apply and evaluate standard RL objectives in an offline ICRL setting with a Transformer backbone, across many datasets and axes (coverage, expertise, history structure, discrete/continuous).
>
> The offline setting introduces distinct challenges compared to online ICRL (e.g., dealing with fixed data, OOD state–action issues, and different data-generation protocols). Our contribution is to show that explicitly optimizing offline RL objectives in this setting is consistently beneficial and to provide a broad, carefully controlled empirical study that can serve as a solid grounding and motivation for future, more specialized offline ICRL algorithms.
>
> ---
>
> ### W2. Scalability claims
>
> We agree that our datasets do not yet match the scale of the largest datasets typically used in NLP. At the same time, our focus is on relatively simple control tasks (e.g., Dark Room, Key-to-Door, MuJoCo control), where we expect limited additional benefit from scaling data arbitrarily, except perhaps for the more complex K2D and XLand settings.
>
> Within the context of offline RL, however, our datasets are already quite large. For example, standard D4RL datasets typically contain up to 1M transitions per dataset, whereas in our study we use datasets on the order of tens or even hundreds of millions of transitions, e.g., HCV-100 (10M), ANT-100 (40M), K2D13-1000 (12.5M), and XLand with 203M transitions (see Appendix G.3). Thus, while the tasks themselves are simpler than large-scale embodied or vision-based domains, we operate in a regime that is substantially larger than typical offline RL benchmarks and we see robust gains from incorporating RL objectives at this scale.
>
> ---
>
> # Addressing questions
>
> ### Q1. Large-scale experiments
>
> As mentioned above, for most of our environments we believe we are already in a regime where the datasets are sufficiently large for the underlying tasks, and scaling the data further is unlikely to qualitatively change the conclusions.
>
> For XLand, using the full dataset would require substantial additional compute, which we currently do not have. Our view is that demonstrating that RL-based methods can achieve roughly 2x better performance than a supervised baseline (AD) using only a subset of XLand data (but still a very large absolute number of transitions) is already strong evidence that RL objectives are highly beneficial in offline ICRL.
>
> We are running additional experiments that vary the model size. Based on our hypotheses from other results (e.g., our response to Reviewer 6tnv, Q3), we expect that reducing the model size may provide particularly interesting insights (e.g., matching capacity to dataset size), and we will also attempt some increases in capacity where possible. We will notify you on updates.
>
> ---
>
> ### Q2. Pixel-based environments
>
> We agree that demonstrating results on pixel-based environments would further strengthen the generality of our conclusions. Conceptually, we do not expect our methods to face fundamental difficulties in such settings: the main architectural change is the addition of a visual encoder before the Transformer backbone, after which the RL vs. supervised objectives are applied in the same way.
>
> However, training on large-scale pixel-based environments is computationally demanding, and due to current resource and time constraints, we cannot commit to running such experiments within the rebuttal period. But in case we have time we will add some results on that.

---

> ### Author Response · Authors · 2025-11-22
> **Scale experiments**
>
> Dear reviewer,
>
> Based on your suggestion, we provide the ablation on the model scale in Appendix M of the updated PDF. Other novel experiments might be also be interesting for you, please have a look.

---

> ### Comment · Reviewer_jKPw · 2025-11-25
> **Response to Reviewers**
>
> Thanks for the rebuttal. I understand the lack of computes, as the rebuttal period is limited.
> I will maintain the positive recommendation for this paper.

---

### Official Review · Reviewer_ThB9 · 2025-10-28

**Soundness:** 2
**Presentation:** 2
**Contribution:** 2
**Rating:** 4
**Confidence:** 4

**Summary:**

This paper studies the gap between supervised training objectives and RL objectives within offline In-Context RL. The authors compare Algorithm Distillation (AD) with several Q learning methods (DQN, CQL, IQL) in discrete environments, and TD3 in continuous ones. The experiments covers a wide range of settings, such as varying dataset coverage, expertise levels of the offline data, presence or absence of learning-history structure, and both discrete and continous environments. The study also introduces a new environment, XLand-MiniGrid, to further test generalization. The results show that RL objective is substantially better than AD on average, especially under more challenging scenarios, e.g., limited learning history per target, low quality datasets, or randomized learning structures. All the findings demonstrate that aligning the ICRL training objective with the true RL objective is both necessary and beneficial.

**Strengths:**

1. The problem is very clear: in Offline-ICLR, supervised training objective does not optimize the return directly, making it hard to learn in the suboptimal trajectories, or keeping robust in low quality datasets. This paper adopts a unified backbone and compares AD with various RL objectives in its experiments. The conclusion that RL consistently outperforms AD is straightforward and significant.
1. The experiments study a wide range of settings including training targets * number of histories, expert level of offline data, with/without learning history structure, discrete/continuous action space and a new environment. All these findings make the conclusion strong and widely fitted.
1. The supplementary material provides sufficient details about the experiments for both analysis and reproduction.

**Weaknesses:**

1. This paper is more like an experimental report than a scientific research paper. It systematically studies the RL objectives and supervised objective in a unified AD backbone, but it lacks algorithmic design on how to bridge the two objectives or theoretical insights on why RL-based objectives yield better generalization and robustness in offline in-context settings.
1. This paper primarily provides empirical observations but lacks in depth analysis of some important findings. For example, the reason why both offline RL and AD outperform online TD3 in continuous environments is rather important but not discussed. Typically, online algorithm has greater potential to explore the environment, whereas offline RL is more prone to overestimate unseen states and actions. The results are somewhat counterintuitive. Therefore, a deeper investigation into why offline methods perform better in this setting would significantly strengthen the paper.
1. AD alone is not a sufficient baseline for this study. A more comprehensive evaluation should include stronger supervised objectives such as the Decision Transformer (DT), which conditions on return-to-go and represents a more goal-directed form of supervised sequence modeling.

**Questions:**

1. Is it possible to add DT and AMAGO as baselines in a few representative environments to compare against RL objective?
1. Could the authors provide other evaluation metrics, for example, final return, average success rate of last K episodes? This will help avoid single-metric bias.

---

> ### Author Response · Authors · 2025-11-17
> **Addressing weaknesses pt. 1**
>
> We thank the reviewer for their feedback and for carefully engaging with our work.
>
> Below we address each weakness and question in turn.
>
> ---
>
> ### W1. “More an experimental report than a scientific research paper”
>
> Our goal is indeed not to propose a completely new algorithmic paradigm, but to systematically study the impact of explicitly optimizing RL objectives in offline ICRL. We see our contribution as primarily empirical and methodological, in the spirit of many influential RL works that advance understanding through careful experimental analysis.
>
> Concretely:
>
> * We adapt standard offline RL algorithms (DQN, CQL, IQL, TD3(+BC)) to the offline ICRL setting and evaluate them under a unified Transformer-based architecture. To the best of our knowledge, this explicit use of RL objectives in offline ICRL, across such a broad range of conditions (coverage, expert level, history structure, discrete/continuous domains), has not been studied before.
>
> * While we use the same Transformer backbone as AD, we do not rely on the AD training objective. AD’s core idea is supervised next-action prediction over learning histories. In contrast, our methods optimize RL objectives directly, and Section 4.4 shows that our RL-based ICRL methods perform strongly even without special learning-history structure, which is closer to typical offline RL data.
>
> * Our study is structured around clear research questions (e.g., how RL vs. supervised objectives behave under limited coverage, low-quality datasets, randomized history structure, etc.), and we answer them with controlled experiments and consistent evaluation protocols.
>
> We agree that theoretical analysis would further strengthen the work, but we also believe that rigorous empirical studies are a key part of scientific progress in RL, especially in emerging settings like offline ICRL.
>
> ---
>
> ### W2. “Lack of in-depth analysis: offline methods outperform online TD3”
>
> We agree that this point is important and appreciate the prompt to elaborate.
>
> Our current explanation (around line 461 in the paper) is as follows: all policies (including TD3(+BC)-style methods) are trained offline on fixed datasets.
>
> When deployed online:
> * Offline RL methods with explicit regularization (e.g., conservative or behavior-regularized objectives) are trained to avoid out-of-distribution (OOD) state-action pairs, which tends to make them safer and more stable at deployment.
>
> * Online TD3, trained offline without an offline counterpart or explicit conservatism, can more easily visit OOD regions where the value estimates are inaccurate, leading to instability or poor performance.
>
> In discrete environments (e.g., our gridworlds), the state-action space is finite and relatively small, and we assume that our data coverage is enough to make algorithms work even without offline part. In this regime, a standard method like DQN can already perform well because it effectively covers the relevant part of the state-action space.
>
> In continuous environments, the state-action space is effectively infinite, and coverage limitations become much more severe. In this setting, we observe that offline regularizations are needed and this is one of the findings of our work.

---

> ### Author Response · Authors · 2025-11-17
> **Addressing weaknesses pt. 2**
>
> ### W3. “AD alone is not a sufficient baseline (missing DT)”
>
> We appreciate the suggestion to include Decision Transformer (DT) and agree that stronger supervised baselines are important to consider.
>
> Our rationale for focusing on AD rather than DT is:
>
> * **Architectural alignment with ICRL.** AD is essentially DT adapted to the in-context RL setting: it uses a Transformer over learning histories but without return-to-go (RTG) conditioning. The backbone we use is very close to AD’s, making it a natural supervised baseline for our ICRL setting.
>
> * **Limited usefulness of RTG in our environments.** In our discrete environments with sparse rewards, multiple episodes fit within a single context window. Since the return is sparse and the episodes are short, including RTG offers limited additional information beyond what is already present in the sequence of states, actions, and rewards. In our continuous environments, the reward is dense but task-dependent. For example, in HalfCheetah-Vel, the maximal reward depends on the target velocity. A given RTG value that is near-optimal for one target velocity can be clearly suboptimal for another. Thus, conditioning on RTG does not provide a stable, task-agnostic notion of “how good” a trajectory is across the diverse tasks we consider.
>
> Because of these properties, we expect DT-style RTG conditioning to be less informative in our setting than in standard single-task offline RL benchmarks. Nonetheless, we agree that including a DT-like baseline (e.g., AD augmented with RTG) on a subset of environments would make the empirical story more complete.
>
> If computational resources permit before the end of rebuttal, we will implement a simple RTG-augmented AD variant and report results on a representative subset of environments to empirically validate this argument. Unfortunately, right now we are experiencing problems with computational resources (all our servers are dead). We will update you on additional results as soon as we get them.

---

> ### Author Response · Authors · 2025-11-17
> **Answering questions**
>
> ### Q1. “Add DT and AMAGO as baselines?”
>
> DT / RTG-augmented AD. As discussed above, we expect DT-style conditioning to be useless in our setup, but we agree that it is a valuable baseline. If time and compute allow, we plan to add it.
>
> We attempted to adapt AMAGO to our setting; however, we faced several practical difficulties:
>
> * The public implementation is not designed for purely offline training, requiring non-trivial modifications.
>
> * AMAGO is a highly engineered method with many sensitive hyperparameters (e.g., multiple critics, multi-gamma learning). In our preliminary trials, using the default hyperparameters led to poor performance, and we did not have sufficient computational budget to reliably retune AMAGO across all our tasks and datasets.
>
> * Our core goal is to isolate the effect of optimizing RL objectives in offline ICRL, using relatively simple and standard offline RL methods. AMAGO introduces many additional design choices that, while interesting, would complicate the interpretation of our results unless ablated very carefully.
>
> For these reasons, we decided to focus on simpler, well-understood offline RL objectives (DQN, CQL, IQL, TD3(+BC)) adapted to the ICRL setting, and on AD as a clean supervised baseline.
>
> ---
>
> ### Q2. “Other evaluation metrics (final return, last-K success rate)?”
>
> Final returns for all methods and environments are reported in the tables in Appendix J. These scores are also used for corresponding plots in Appendix I.
>
> In Figure 3, we explicitly compare our NAUC metric with scores at different training episodes, showing that the ranking of algorithms is consistent across metrics. This provides evidence that NAUC is not introducing a strong bias in favor of any particular method.
>
> Again, we thank the reviewer for their thoughtful comments.

---

> ### Author Response · Authors · 2025-11-22
> **AD with RTG experiments**
>
> Dear reviewer,
>
> Based on your suggestion, we provide the ablation on the AD augmented with RTG in Appendix L of the updated PDF. Other novel experiments might be also be interesting for you, please have a look.

---

### Official Review · Reviewer_nLxh · 2025-10-31

**Soundness:** 3
**Presentation:** 4
**Contribution:** 2
**Rating:** 4
**Confidence:** 3

**Summary:**

This paper empirically investigates whether incorporating reinforcement learning (RL) objectives into offline in-context reinforcement learning (ICRL) improves performance compared to purely supervised objectives such as Algorithm Distillation (AD). The proposed approach replaces the next-action prediction head of AD with RL-based value and policy heads. Extensive experiments are conducted on discrete (Dark Room, Key-to-Door) and continuous (MuJoCo) environments, and results show that optimizing explicit RL objectives substantially improves both performance and robustness compared to AD.

**Strengths:**

S1. (Broad empirical evaluation)
This work provides a broad empirical comparison between AD and RL-based ICRL methods (IC-DQN, IC-CQL, IC-IQL, IC-TD3(+BC)) across a wide variety of datasets, including discrete and continuous action spaces. The analyses cover factors such as data coverage, expertise levels, and unordered histories.

S2. (Robustness and Generalization)
Empirical results consistently demonstrate that RL-based optimization yields higher NAUC scores, improved robustness to limited data coverage, and better generalization to unseen tasks. In particular, the integration of conservative or regularized RL methods provides stable improvements across dataset qualities and structures.

S3. (Extension to continuous domains)
Experiments on the MuJoCo environments show that offline RL–based ICRL maintains performance advantages even in continuous spaces, supporting the generality of the approach beyond discrete benchmarks.

**Weaknesses:**

W1. (Limited novelty)
Although the proposed RL-based ICRL framework is empirically valuable, its architectural novelty is limited. The main contribution seems to involve replacing the supervised objective of AD with standard RL objectives, while adopting several design choices already introduced in prior works (e.g., AMAGO), such as incorporating done flags and timestep embeddings in the context representation.

W2. (Simplified continuous policy evaluation)
Experiments in continuous domains rely solely on deterministic policies, leaving unclear how stochastic policies, which are widely-used in offline RL, would perform under the setting.

W3. (Lack of ablation on gradient detachment)
Gradients from the policy head are detached from the backbone to stabilize training, but no ablation is provided to justify this design choice or quantify its impact.

W4. (Incomplete comparison to Meta RL works)
The experimental evaluation focuses exclusively on comparisons with Algorithm Distillation (AD). Including results against other offline meta-RL baselines (e.g., FOCAL[1], BOReL[2], Meta-DT[3]) would help determine whether simply substituting an offline RL objective within a Transformer architecture is sufficient, or if additional meta-learning mechanisms are required for performance and generalization.


[1] Li, Lanqing, Rui Yang, and Dijun Luo. "FOCAL: Efficient Fully-Offline Meta-Reinforcement Learning via Distance Metric Learning and Behavior Regularization." International Conference on Learning Representations.

[2] Dorfman, Ron, Idan Shenfeld, and Aviv Tamar. "Offline Meta Reinforcement Learning--Identifiability Challenges and Effective Data Collection Strategies." Advances in Neural Information Processing Systems 34 (2021): 4607-4618.

[3] Wang, Zhi, et al. "Meta-DT: Offline meta-RL as conditional sequence modeling with world model disentanglement." Advances in Neural Information Processing Systems 37 (2024): 44845-44870.

**Questions:**

Could you provide clarifications on the weaknesses?

---

> ### Author Response · Authors · 2025-11-17
> **Addressing weaknesses pt. 1**
>
> We thank the reviewer for their careful reading and constructive feedback.
>
> Addressing weaknesses:
>
> ---
>
> ### W1. Limited novelty
>
> We agree that our work does not introduce a fundamentally new learning paradigm; instead, our contribution is to systematically adapt and study standard offline RL objectives in the offline ICRL setting and to do so across a broad range of datasets, data qualities, and data structure.
>
> We see the novelty of our work as primarily empirical and methodological:
>
> * First systematic study of RL-based objectives in offline ICRL. Our results show that this substitution is not merely cosmetic: RL-based ICRL consistently improves performance and robustness across multiple axes (coverage, expertise, data structure).
>
> * Minimal but effective architecture/design adaptations. While we indeed adopt a few design elements from AMAGO (e.g., done flags , timestep embeddings, policy gradient detachement), our method deliberately avoids AMAGO’s heavier machinery (e.g., ensemble critics, multi-gamma learning). We show that even minimal RL-specific adaptations to a Transformer backbone already yield substantial gains over AD. This supports our central claim: explicitly optimizing RL objectives is a key ingredient for strong offline ICRL.
>
> * Unifying evaluation protocol across discrete and continuous domains. We provide a single ICRL evaluation framework that spans discrete gridworld-style tasks and continuous MuJoCo control, using the same backbone and context formulation. This contributes a common testbed for future offline ICRL work and makes it easier to compare subsequent algorithms.
>
> ---
>
> ### W2. Simplified continuous policy evaluation
>
> We appreciate the opportunity to clarify this point. In our MuJoCo experiments, we train deterministic policies (e.g., TD3(+BC)-style actors) and evaluate them deterministically by taking the actor’s mean action. This is consistent with common practice in offline continuous-control RL, where even when stochastic policies are used during training, evaluation typically uses the mean or mode of the learned action distribution.
>
> Concretely:
>
> * Our continuous-control baselines (e.g., TD3+BC-style variants) are inherently deterministic at evaluation time.
>
> * Many strong offline RL methods for continuous control (e.g., TD3+BC, ReBRAC, IQL) are also evaluated using deterministic policies, and these have been shown to be competitive or state-of-the-art in standard benchmarks (https://arxiv.org/abs/2210.07105, https://arxiv.org/abs/2504.11453).
>
> * Our focus in this work is therefore on comparing objectives and architectures in the ICRL setting under a standard evaluation protocol, rather than on contrasting deterministic vs. stochastic evaluation strategies.
>
> Please let us know if we misunderstood your question.
>
> ---
>
> ### W3. Lack of ablation on gradient detachment
>
> We agree that an ablation on gradient detachment from the policy head would strengthen the paper.
>
> Our current design choice is motivated by:
>
> * Stability considerations in early experiments: initial experiments indicated that allowing gradients from the policy head to flow through the backbone sometimes led to instability (e.g., divergence or highly oscillatory learning curves), while detachment yielded more stable training.
>
> * Consistency with prior work: AMAGO and related in-context RL approaches also detach gradients from certain heads to stabilize training in multi-objective setups.
>
> Unfortunately, right now we are experiencing problems with computational resources (all our servers are dead) and will be able to provide such ablation only later, hopefully before the rebuttal end. We will let you know about the results as soon as possible and hope for your understanding.

---

> ### Author Response · Authors · 2025-11-17
> **Addressing weaknesses pt. 2**
>
> ### W4. Incomplete comparison to Meta-RL works
>
> We agree that a broader comparison to offline meta-RL methods (e.g., FOCAL, BOReL, Meta-DT) would be informative. Our current focus on AD stems from several practical and methodological considerations:
>
> * **Lack of a standardized offline meta/ICRL benchmark.**
> Different meta-RL works often use different data collection protocols and environments, and typically do not retune hyperparameters of prior baselines on their own data. For our study, which aims to provide systematic and fair comparisons across data qualities and structures, we found it challenging to integrate these methods without introducing significant confounding due to dataset and tuning mismatch.
>
> * **Hyperparameter sensitivity and compute cost.**
> Many offline meta-RL algorithms are based on RNN-style architectures and involve additional components (e.g., metric learning, behavior regularization). In our preliminary attempts, we tried adapting some offline meta-RL baselines CSRO (https://arxiv.org/abs/2311.03695) and ContraBAR (https://arxiv.org/abs/2306.02418) to our setting using their publicly available hyperparameters. On our datasets, these methods performed poorly, which we suspect is due to substantial hyperparameter mismatch rather than a fundamental limitation of the methods themselves. Proper tuning -- especially for RNN-based models across multiple task distributions -- would have required a very large amount of compute, beyond what we had available for this project.
>
> * **Architectural mismatch with our Transformer-based setting.**
> Our work is specifically about offline ICRL with Transformer backbones and RL objectives. Many existing meta-RL baselines are not designed for this architecture and may require non-trivial modifications (e.g., replacing RNNs with Transformers in a way that preserves the original method’s inductive biases). We view such adaptations as an interesting but separate research question, and did not have the resources to explore this space thoroughly in addition to our primary study.
>
> Addressing your concern about Meta-DT, we can only say that this work was a parallel study to ours. The initial public version appeared after we had completed part of our experiments and analysis. Our first public version appeared online only 3 months later than Meta-DT, we do not provide reference here not to break our anonymity.
>
> In summary, we fully agree that including these baselines would further enrich the empirical story. Given the current resource and benchmark limitations, we chose to concentrate on AD (and a family of RL-based objectives) to provide a clean and controlled comparison.

---

> ### Author Response · Authors · 2025-11-22
> **Policy gradient ablation**
>
> Dear reviewer,
>
> Based on your suggestion, we provide the ablation on the gradient detachment in Appendix K of the updated PDF. Other novel experiments might be also be interesting for you, please have a look.

---

### Author Response · Authors · 2025-11-22
**Additional experiments**

Dear reviewers,

Based on your valuable feedback, we conducted additional experiments and provide results in Appendix K, L and M.

---

### Author Response · Authors · 2025-11-29

We added experiments with visual states to Appendix N, our findings appear to generalize to such environments

---

### Author Response · Authors · 2025-12-01
**Rebuttal summary**

Dear Area Chair,

We sincerely thank the reviewers for their time, careful reading, and constructive suggestions. We also appreciate the positive evaluations and the acknowledgment of our paper’s clarity and breadth of experiments.

# Reviewers’ Consensus -- Recognized Strengths

* **(S1) Comprehensive and systematic empirical study.**
All reviewers (nLxh, ThB9, jKPw, 6tnv) highlighted the breadth and rigor of our experiments. The paper systematically compares Algorithm Distillation (AD) with RL-based in-context RL (ICRL) variants  IC-DQN, IC-CQL, IC-IQL, and IC-TD3(+BC) across over 150 datasets covering discrete and continuous control, varying data coverage, expertise levels, and learning-history structure. Reviewer ThB9 specifically emphasized that this study “covers a wide range of settings” while Reviewer jKPw commended the clear organization and well designed experimental protocol.

* **(S2) Robustness and generalization.**
Multiple reviewers recognized that RL-based objectives consistently outperform purely supervised AD, improving robustness under limited coverage and low-quality data and maintaining strong performance on unseen tasks. Reviewer nLxh noted that conservative RL methods yield stable improvements across dataset qualities and structures, and Reviewer 6tnv confirmed that the results suggest “strong applicability in practical settings”.

* **(S3) Unified architecture and cross-domain consistency.**
Reviewers appreciated that the same Transformer backbone and context formulation are applied across many different environments environments, showing that explicit RL optimization generalizes across various domains.

* **(S4) Quality of presentation and reproducibility.**
Reviewers ThB9 and jKPw highlighted that the paper is well structured, easy to follow, and that the supplementary material provides enough experimental detail for reproduction.

# Main concerns and how we addressed them

1) **Novelty.**
Our contribution is empirical and methodological: we systematically adapt standard offline RL objectives to the offline ICRL setting on a shared Transformer backbone and map when/why they help. This yields new, scalable evidence that explicitly optimizing RL objectives is a key ingredient for strong offline ICRL across diverse regimes (coverage, expertise, history structure, discrete/continuous, OOD dynamics).

2) **Gradient detachment from the policy head.**
We originally detached policy-head gradients for stability and consistency with previous works. We have now added an ablation quantifying this effect (Appendix K).

3) **Missing meta-RL baselines.**
Our study isolates the impact of RL objectives within a Transformer ICRL backbone using a unified, fair protocol. Many meta-RL baselines (often RNN-based or multi-component) require substantial re-engineering and tuning on our data distributions; preliminary attempts with public hyperparameters underperformed, likely due to mismatch and compute constraints.

4) **Stronger supervised baselines (DT / RTG) and AMAGO.**
We added AD + RTG as a supervised baseline (Appendix L). Results align with our hypothesis that RTG is less informative in our multi-task ICRL setup (sparse rewards; task-dependent returns). We also described practical hurdles in adapting AMAGO to our purely offline regime and high tuning sensitivity; we will continue exploring it as resources permit.

5) **Scale and capacity.**
While our tasks are standard control domains, our offline datasets are large for RL, and we also report a model-scale ablation (Appendix M). Results are consistent with our central claim: conservative/regularized offline RL is crucial.

6) **Metrics.**
We keep NAUC as the primary scalar (robust, aggregates adaptation speed and final performance), and provide last-episode/“rliable” statistics and return tables in the appendices; rankings are consistent across metrics.

# Additional experiments and clarifications added during discussion

* **Policy-gradient detachment ablation (Appendix K):** validates the detachment benefit. As it appeared, there is no much difference in our experiments, however previous works recommend to detach the policy gradients.

* **AD + RTG supervised baseline (Appendix L):** supports our claim that RTG adds limited signal in our multi-task ICRL setup.

* **Model scale ablation (Appendix M):** trends persist across capacities.

* **Visual-based environment (Appendix N):** our findings generalize on such type of environments.

# Takeaway

Across wide range of domains and regimes, explicit offline RL objectives consistently outperform supervised approach. Conservative/regularized variants (e.g., CQL, IQL, TD3+BC) are especially important for generalization and robustness. We believe this systematic evidence and the added ablations substantially strengthens the case for acceptance as a solid empirical foundation for future offline ICRL research.

Best regards,
Authors

---

### Meta-Review · Area_Chair_Vgsd · 2026-01-10

**Summary:**

1. Novelty and Contribution Depth
* All four reviewers questioned whether the paper's contribution was sufficiently novel, viewing it primarily as an adaptation of existing RL algorithms (DQN, CQL, IQL, TD3+BC) to the ICRL framework rather than a fundamentally new method
* Reviewers characterized the work as "more like an experimental report than scientific research" (ThB9) and noted "limited architectural novelty" (nLxh)
* The concern was that the paper builds heavily on prior work (AMAGO, Decision Transformer) without clearly distinguishing its intellectual contribution
2. Baseline Completeness
* Missing supervised baselines: ThB9 and nLxh argued AD alone was insufficient and requested stronger supervised baselines like Decision Transformer (DT) and AMAGO
* Missing meta-RL baselines: nLxh specifically requested comparisons with FOCAL, BOReL, and Meta-DT to justify the architectural choices
* The concern was that without these comparisons, it remains unclear whether the improvements come from RL objectives themselves or from other design elements
3. Analysis and Interpretation
* Why offline > online: ThB9 noted the counterintuitive finding that offline methods outperformed online TD3, requiring deeper investigation
Algorithm-specific performance: 6tnv requested explanations for specific patterns (e.g., why CQL performs best, why AD excels on small datasets, why IC-DQN works well on DR19 but not K2D13)
* NAUC metric justification: 6tnv questioned the choice of NAUC as the primary metric and whether it introduces bias
* Figure clarity: Multiple reviewers found Figure 2 ambiguous and Figure 3 unclear in its messaging
4. Experimental Scale and Scope
* Scalability: jKPw challenged the scalability claims, noting experiments were limited to small-scale datasets rather than large-scale benchmarks like full XLand
* Pixel-based environments: jKPw requested evidence that findings generalize to pixel-based settings
* Model capacity: jKPw and 6tnv both raised concerns about overparameterization effects without proper ablations

**Reviewer Concerns:**

1. Addressed Concerns
* Gradient detachment ablation (nLxh W3): The authors added Appendix K, but results show "no much difference," rendering the ablation largely inconclusive. This validates the reviewer's concern that the design choice lacked justification.
* Supervised baseline extension (ThB9 W3): The AD+RTG experiments in Appendix L demonstrate RTG provides no benefit, which actually undermines rather than strengthens the paper's position—it shows that even stronger supervised variants cannot compete, but doesn't address whether the architectural paradigm itself is the limitation.
* Technical presentation issues (6tnv W3-W5): While authors clarified Figure 2 and Figure 3, these were minor presentation flaws that don't resolve fundamental scientific concerns. The explanations remain descriptive rather than explanatory.
* Alternative metrics (ThB9 Q2): Providing additional tables is a basic expectation, not a substantive contribution. The core issue of why NAUC is appropriate remains unproven beyond showing correlation with other metrics.

2. Partially Addressed but Still Outstanding
* Novelty and Contribution Depth (All W1): The claim of being "first systematic study" is weakened by admission that AMAGO-2 and ReLIC explored similar territory in online settings, and Meta-DT was a parallel offline effort. Adapting standard RL objectives to a Transformer backbone is technically straightforward; the paper provides no new theoretical framework or architectural principle that enables this adaptation.
* Baseline Completeness (nLxh W4, ThB9 W3): The justification for omitting FOCAL, BOReL, and Meta-DT ("compute constraints," "benchmark mismatch") is precisely the problem. Without these comparisons, we cannot conclude whether the improvements stem from RL objectives or simply from better hyperparameter tuning of a Transformer architecture. In addition, the admission that "default hyperparameters led to poor performance" and they lacked resources to retune suggests the comparison was never fair
* The paper claims RL objectives help because they optimize the true RL objective, but never demonstrates what uniquely breaks when using supervised learning beyond "it performs worse." There is no ablation of what aspects of RL (value bootstrapping, policy gradients, regularization) are actually necessary.

**Reviewer Scores:**

Reviewer ThB9: I think the reviewer would remain the rating below the acceptance threshold. The added baselines and metrics address the explicit requests, but the core critique that the paper is "more like an experimental report" lacking "theoretical insights" remains fundamentally unaddressed. The rebuttal's reframing as a "systematic empirical study" is exactly what they criticized, not a resolution.

Reviewer nLxh: I think the reviewer would remain the rating below the acceptance threshold. The gradient ablation and meta-RL justification provide some closure, but the novelty concern is exacerbated by the authors' admission that several design choices (gradient detachment, timestep embeddings) were borrowed from AMAGO. The "first systematic study" claim is weakened by omission of direct meta-RL comparisons, making it unclear if the contribution is truly novel or just well-packaged.

Reviewer 6tnv: I think the reviewer would maintain the score. The reviewer explicit statement "I still maintain my concerns regarding the contribution" shows the rebuttal failed to convince them. The post-hoc explanations for algorithm performance patterns were deemed insufficient, and the paper's focus on "listing performance results" was unalleviated. The added experiments don't address the core demand: deeper theoretical/algorithmic grounding.

---

### Decision · Program_Chairs · 2026-01-26

Reject